# ISMIP6-based projections of ocean-forced Antarctic Ice Sheet evolution using the Community Ice Sheet Model

William H. Lipscomb[1], Gunter R. Leguy[1], Nicolas C. Jourdain[2], Xylar Asay-Davis[3], Hélène Seroussi[4], and Sophie Nowicki[5,6]

[1]Climate and Global Dynamics Laboratory, National Center for Atmospheric Research, Boulder, CO, USA
[2]Univ. Grenoble Alpes/CNRS/IRD/G-INP, IGE, Grenoble, France
[3]Los Alamos National Latoratory, Los Alamos, NM, USA
[4]Jet Propulsion Laboratory, California Institute of Technology, Pasadena, CA, USA
[5]NASA Goddard Space Flight Center, Greenbelt, MD, USA
[6]University at Buffalo, Buffalo, NY, USA

**Correspondence:** William H. Lipscomb (lipscomb@ucar.edu)

**Abstract.** The future retreat rate for marine-based regions of the Antarctic Ice Sheet is one of the largest uncertainties in sea-level projections. The Ice Sheet Model Intercomparison Project for CMIP6 (ISMIP6) aims to improve projections and quantify uncertainties by running an ensemble of ice sheet models with atmosphere and ocean forcing derived from global climate models. Here, the Community Ice Sheet Model (CISM) is used to run ISMIP6-based projections of ocean-forced Antarctic Ice Sheet evolution. Using multiple combinations of sub-ice-shelf melt parameterizations and calibrations, CISM is spun up to steady state over many millennia. During the spin-up, basal friction parameters and basin-scale thermal forcing corrections are adjusted to optimize agreement with the observed ice thickness. The model is then run forward for 550 years, from 1950–2500, applying ocean thermal forcing anomalies from six climate models. In all simulations, the ocean forcing triggers long-term retreat of the West Antarctic Ice Sheet, especially in the Filchner-Ronne and Ross sectors. Mass loss accelerates late in the 21st century and then rises steadily for several centuries without leveling off. The resulting ocean-forced sea-level rise at year 2500 varies from about 150 mm to 1300 mm, depending on the melt scheme and ocean forcing. Further experiments show relatively high sensitivity to the basal friction law, moderate sensitivity to grid resolution and the prescribed collapse of small ice shelves, and low sensitivity to the stress-balance approximation. The Amundsen sector exhibits threshold behavior, with modest retreat under many parameter settings, but complete collapse under some combinations of low basal friction and high thermal forcing anomalies. Large uncertainties remain, as a result of parameterized sub-shelf melt rates, simplified treatments of calving and basal friction, and the lack of ice–ocean coupling.

## 1 Introduction

The Antarctic Ice Sheet has been losing mass at an increasing rate for the past several decades (Shepherd et al., 2018; Rignot et al., 2019). Much of the ice loss has been driven by increased access of warm Circumpolar Deep Water (CDW) to marine-based parts of the West Antarctic Ice Sheet (WAIS), likely caused in part by radiatively forced changes in wind patterns (Thomas et al., 2004; Jenkins et al., 2010; Rignot et al., 2013; Holland et al., 2019; Gudmundsson et al., 2020). Paleoclimate

records show that the WAIS retreated in past climates not much warmer than the present, including the Last Interglacial (Dutton et al., 2015). Many WAIS glaciers lie on reverse-sloping beds (i.e., with the seafloor sloping upward in the direction of ice flow), making these glaciers vulnerable to the marine ice sheet instability (MISI; Weertman, 1974; Mercer, 1978; Schoof, 2007). Models suggest that ice in the Amundsen Sea sector, including Thwaites and Pine Island Glaciers, may already be in the early stages of collapse (Joughin et al., 2014; Favier et al., 2014). Despite recent advances in ice sheet modeling (Pattyn, 2018), projections of 21st century Antarctic Ice Sheet retreat and resulting sea-level rise (SLR) are highly uncertain, ranging from modest ($\sim 10$ cm; Ritz et al., 2015) to large and abrupt ($> 1$ m; Pollard and DeConto, 2016). Although recent work (Edwards et al., 2019) suggests that the most extreme projections may overestimate the rate of SLR, much of the WAIS is likely vulnerable to long-term, self-sustaining retreat.

To improve projections and better understand and quantify uncertainties, ice sheet modelers have organized several community intercomparisons, most recently the Ice Sheet Model Intercomparison Project for CMIP6 (ISMIP6; Nowicki et al., 2016). ISMIP6 has been endorsed by the Coupled Model Intercomparison Project – Phase 6 (CMIP6; Eyring et al., 2016) and is providing process-based ice-sheet and sea-level projections linked to the CMIP ensemble of climate projections from global climate models. Here, we will refer to these global models as Earth system models (ESMs); they are also commonly known as atmosphere–ocean general circulation models (AOGCMs). Nowicki et al. (2020) have summarized the ISMIP6 projection protocols for standalone ice sheet model (ISM) experiments. The general strategy is to use output from CMIP5 and CMIP6 global models to derive atmosphere and ocean fields for forcing ISMs over the period 2015–2100. Goelzer et al. (2020) and Seroussi et al. (2020) evaluated the multi-model ensembles of projections for the Greenland and Antarctic Ice Sheets, respectively.

Seroussi et al. (2020) analyzed 16 sets of Antarctic simulations from 13 international groups, with forcing derived from six CMIP5 ESMs and representing a spread of climate model results. The Antarctic contribution to sea level during 2015–2100 varies from sea-level fall of 7.8 cm to sea-level rise of 30.0 cm under the RCP (Representative Concentration Pathway) 8.5 scenario. The main contributor to falling sea level is a more positive surface mass balance (SMB), with most ESMs simulating more snowfall in a warming climate. Antarctic-sourced SLR, on the other hand, is driven by ocean warming leading to marine ice-sheet retreat and dynamic mass loss, especially for the WAIS. The amount of ice loss varies widely across simulations because of differences in the strength and spatial patterns of ESM ocean warming, and in ISM physics and numerics.

This paper complements the study of Seroussi et al. (2020) by evaluating the Antarctic response to ocean forcing in a single model, the Community Ice Sheet Model (CISM; Lipscomb et al., 2019). A novel feature is the tuning of a single parameter in each of 16 sectors to adjust the ocean forcing, thus optimizing model agreement with thickness observations near grounding lines. We do not simulate atmospheric forcing changes, since the ice-sheet response to SMB anomalies is relatively consistent across models for both Greenland (Goelzer et al., 2020) and Antarctica (Seroussi et al., 2019, 2020). This is not to say that SMB changes are unimportant for future Antarctic Ice Sheet evolution, but rather that the ocean contribution is more uncertain than the SMB contribution. Instead, we consider the Antarctic Ice Sheet response to increased sub-ice-shelf melting as a function of basal melting parameterizations (Jourdain et al., 2020) and ESM ocean warming. To study long-term ice sheet evolution, we extend the simulations to year 2500, with the forcing after 2100 based on late-21st-century forcing from high-end emissions scenarios. In this way, we explore the following questions:

- If the ocean warming projected for the late 21$^{st}$ century were to continue unabated for several more centuries, what parts of the Antarctic Ice Sheet would be most vulnerable to retreat?

- Given this vulnerability, and assuming that emissions remain on a high-end trajectory until 2100, what will be the committed long-term, ocean-forced sea-level rise from Antarctic ice loss? By "committed", we refer to SLR resulting from ice-sheet retreat that has been set in motion and is likely irreversible.

- How sensitive is the simulated ice-sheet retreat to poorly constrained model parameters and forcing?

Because of modeling and forcing uncertainties that grow larger on time scales beyond a century, our long-term results (e.g., beyond 2100), should not be viewed as predictions. Rather, we aim to simulate retreat processes, including MISI, that could unfold over several centuries, and to explore parameter space, using the ISMIP6 framework to build on previous multi-century Antarctic simulations (e.g., Pollard and Deconto, 2009; Cornford et al., 2015; Pollard and DeConto, 2016; Larour et al., 2019). The ice-sheet physics is conventional in the sense that it includes well-understood retreat mechanisms such as MISI, but not fast (and more speculative) feedbacks such as cliff collapse (Pollard et al., 2015; Pollard and DeConto, 2016). Some feedbacks such as solid-Earth rebound and relative-sea-level changes (Gomez et al., 2010; Larour et al., 2019) are omitted, and there is no ice–ocean coupling (e.g., De Rydt and Gudmundsson, 2016; Seroussi et al., 2017; Favier et al., 2019). Thus, the timing and magnitude of simulated ice sheet retreat are imprecise, but we can identify responses that are robust across simulations, while drawing attention to the largest sources of uncertainty.

Section 2 gives an overview of CISM and summarizes the protocol and ocean data sets for ISMIP6 Antarctic projections. We describe the model initialization technique and evaluate the spun-up state in Sect. 3. In Section 4, we present the results of ocean-forced simulations, including standard configurations and several sensitivity experiments. Section 5 gives conclusions and suggests directions for future research.

## 2 Model and experimental description

### 2.1 The Community Ice Sheet Model

CISM is a parallel, higher-order ice sheet model designed to perform both idealized and whole-ice-sheet simulations on timescales of decades to millennia. It is a descendant of the Glimmer model (Rutt et al., 2009) and is now an open-source code developed mainly at the National Center for Atmospheric Research, where it serves as the dynamic ice sheet component of the Community Earth System Model (CESM). The most recent documented release was CISM v2.1 (Lipscomb et al., 2019), coinciding with the 2018 release of CESM2 (Danabasoglu et al., 2020). The model performs well for community benchmark experiments, including the ISMIP-HOM experiments for higher-order models (Pattyn et al., 2008) and several stages of the Marine Ice Sheet Model Intercomparison Project: the original MISMIP (Pattyn et al., 2012), MISMIP3d (Pattyn et al., 2013), and MISMIP+ (Asay-Davis et al., 2016; Cornford et al., 2020). CISM participated in two earlier ISMIP6 projects focused on ice sheet model initialization: initMIP-Greenland (Goelzer et al., 2018) and initMIP-Antarctica (Seroussi et al., 2019). More

recently, CISM results were submitted for the ISMIP6 projections (Goelzer et al., 2020; Seroussi et al., 2020), the LARMIP-2 experiments (Levermann et al., 2019), and the Antarctic BUttressing Model Intercomparison Project (ABUMIP; Sun et al., 2020).

CISM runs on a structured rectangular grid with a terrain-following vertical coordinate. Most simulations in this paper were run on a 4-km grid, as for the CISM contributions to the ISMIP6 Antarctic projections (Seroussi et al., 2020). At 4-km resolution, grounding lines are under-resolved, but this is the finest resolution that permits a large suite of whole-Antarctic simulations at reasonable computational cost. To test sensitivity to grid resolution, we repeated some experiments on 8 km and 2 km grids (Sect. 4.2.1). All simulations were run with 5 vertical levels. Increasing the number of vertical levels would not substantially change the results, especially in regions dominated by basal sliding rather than vertical shear (i.e., the regions critical for grounding-line retreat and SLR). Scalars (e.g., ice thickness $H$ and temperature $T$) are located at grid cell centers, with horizontal velocity $\boldsymbol{u} = (u, v)$ computed at vertices. The dynamical core has parallel solvers for a hierarchy of approximations of the Stokes stress-balance equations, including the shallow-shelf approximation (MacAyeal, 1989), a depth-integrated higher-order approximation (Goldberg, 2011), and the 3D Blatter-Pattyn (BP) higher-order approximation (Blatter, 1995; Pattyn, 2003). The latter two approximations are classified as L1L2 and LMLa, respectively, in the terminology of Hindmarsh (2004). The simulations for this paper use the depth-integrated solver, known as DIVA (Depth-Integrated Viscosity Approximation), which solves a 2D elliptic equation for the mean horizontal velocity, followed by a vertical integration at each vertex to obtain the full 3D velocity. This solver gives a good balance between accuracy and efficiency in idealized settings and whole-ice-sheet (Leguy et al., 2020; Lipscomb et al., 2019). In Sect. 4.2.2, we compare DIVA to the more expensive BP solver in selected runs.

CISM supports several basal friction laws. Most simulations in this study use a power law:

$$\boldsymbol{\tau}_b \approx C_p |\boldsymbol{u_b}|^{\frac{1}{m}-1} \boldsymbol{u_b}, \tag{1}$$

where $\boldsymbol{\tau_b}$ is the basal shear stress, $\boldsymbol{u_b}$ is the basal ice velocity, $m = 3$ is a power-law exponent, and $C_p$ is an empirical coefficient for power-law behavior. For a subset of simulations (Sect. 4.2.4), we use a friction law based on Schoof (2005), with a functional form suggested by Asay-Davis et al. (2016):

$$\boldsymbol{\tau}_{\mathbf{b}} = \frac{C_p C_c N}{\left[ C_p^m |\boldsymbol{u_b}| + (C_c N)^m \right]^{\frac{1}{m}}} |\boldsymbol{u_b}|^{\frac{1}{m}-1} \boldsymbol{u_b}, \tag{2}$$

where $N$ is the effective pressure and $C_c$ is an empirical coefficient for Coulomb behavior. In the ice sheet interior, where the ice is relatively slow-moving with high effective pressure, this law asymptotes to power-law behavior, Eq. (1). Where the ice is fast-moving with low effective pressure, we have Coulomb behavior:

$$\boldsymbol{\tau}_b \approx C_c N \frac{\boldsymbol{u_b}}{|\boldsymbol{u_b}|}. \tag{3}$$

Since the power-law coefficient is spatially variable and poorly constrained, we use it as a tuning factor, adjusting $C_p(x, y)$ for grounded ice to minimize the difference between the simulated and observed ice thickness. This tuning is described in

Section 3.1. Following Asay-Davis et al. (2016), the dimensionless Coulomb coefficient is set to $C_c = 0.5$. As in Leguy et al. (2014), the effective pressure is given by

$$N(p) = \rho_i g H \left(1 - \frac{H_f}{H}\right)^p, \tag{4}$$

where $\rho_i$ is ice density, $g$ is gravitational acceleration, $H_f = \max(0, -\frac{\rho_w}{\rho_i} b)$ is the flotation thickness, $\rho_w$ is seawater density,

and $b$ is the bed elevation, defined as negative below sea level. The parameter $p$ represents the hydrological connectivity of the subglacial drainage system to the ocean and varies between zero (no connectivity, with $N$ equal to overburden pressure) and one (strong connectivity, with $N$ approaching zero near the grounding line).

CISM also supports several calving laws, but none was found to give calving fronts in good agreement with observations for both large and small Antarctic ice shelves. Instead, we use a no-advance calving mask, removing all ice that flows beyond the

observed calving front. The calving front can retreat where there is more surface and basal melting than advective inflow, but more often the calving front remains in its initial position. For a subset of experiments (Sect. 4.2.3), we apply additional masks to force the collapse of selected ice shelves.

Many studies (e.g., Pattyn, 2006; Schoof, 2007) have emphasized the challenges of simulating ice dynamics in the transition zone between grounded and floating ice, especially when the grid resolution is ∼1 km or coarser, as is the case for most

simulations of whole ice sheets. Grounding line parameterizations (GLPs), which give a smooth transition in basal shear stress across the transition zone, have been shown to improve numerical accuracy in models with relatively coarse grids (e.g., Gladstone et al., 2010; Leguy et al., 2014; Seroussi et al., 2014). Our experiments use a GLP described by Leguy et al. (2020). For staggered grid cells (i.e., cells centered on the velocity $\boldsymbol{u}$) that contain the grounding line, the basal friction is weighted by the area fraction $\phi_g$ of grounded ice, as determined from a flotation function defined at adjacent cell centers:

$$f_{\text{float}} = -b - (\rho_i/\rho_w)H. \tag{5}$$

Values of $f_{\text{float}}$ are negative, positive, and zero, respectively, for grounded ice, floating ice, and along the grounding line. In Sect. 4.2.1, we test the sensitivity of results to changes in grid resolution.

Finally, CISM supports several parameterizations of sub-ice-shelf melting. For Antarctic projections, the melt parameterizations are based on ISMIP6 protocols (Sect. 2.2). As with basal friction, CISM uses a GLP for basal melting; melt in grid cells

containing the grounding line is weighted by the floating area fraction, $1 - \phi_g$. Some studies (e.g., Seroussi and Morlighem, 2018) have argued that melt should not be allowed in partly grounded cells. For CISM, however, Leguy et al. (2020) found that applying some melt in these cells improves numerical convergence as a function of resolution, at least for idealized configurations with strong buttressing (Asay-Davis et al., 2016).

## 2.2   Protocols based on ISMIP6 for Antarctic projections

Protocols for the ISMIP6 Antarctic projections are described in detail by Nowicki et al. (2020) and Jourdain et al. (2020). Here, we give a brief summary, noting where the simulations in this study differ from the protocols. The ISMIP6 projection experiments are run with standalone ISMs, forced by time-varying, annual-mean atmosphere and ocean fields derived from

the output of CMIP5 and CMIP6 ESMs. In our study, the projection experiments use ocean forcing that evolves during the simulation, but the atmospheric forcing (specifically, the SMB) is held to the values used when spinning up the model. The CMIP5 models were selected by a procedure described by Barthel et al. (2020) to represent present-day Antarctic conditions well and to sample the diversity in climate evolution. For CMIP6 models, there was not time for a formal selection, but output from selected models was processed as it became available. Each experiment runs for 86 years, from the start of 2015 to the end of 2100. Model initialization methods are left to the discretion of each group, and are detailed for specific models in Seroussi et al. (2019, 2020). If the initialization date is before the start of the projections, a short historical run is needed to advance the ISM to the end of 2014.

For projections, ISMIP6 provides data sets of annual-mean atmosphere and ocean forcing on standard grids. The ocean forcing consists of 3D fields of thermal forcing (i.e., the difference between the in situ ocean temperature and the in situ freezing temperature), as described by Jourdain et al. (2020). It is not possible to use ocean temperature and salinity directly from ESMs, because current CMIP models do not simulate ocean properties in ice shelf cavities. Moreover, global ocean models are usually run at a resolution of $\sim 1°$, too coarse to give an accurate mean ocean state near Antarctic ice shelves. Instead, Jourdain et al. (2020) combined recent data sets (Locarnini et al., 2019; Zweng et al., 2019; Good et al., 2013; Treasure et al., 2017), spanning the years 1995–2018, to construct a 3D gridded climatology of ocean temperature and salinity. This climatology was interpolated to fill gaps and extrapolated into ice-shelf cavities. To obtain forcing fields for projections, temperature and salinity from the various ESM ocean models were extrapolated to cavities, and their anomalies were added to the observationally derived climatology. The result is a time-varying 3D product that can be vertically interpolated to give the thermal forcing at the base of a dynamic ice shelf at any time and location in the Antarctic domain. The extrapolation procedure does not directly resolve ocean circulation in cavities, but was seen as a necessary simplification in the absence of output from high-resolution regional ocean models.

To compute basal melt rates beneath ice shelves, ISMIP6 models can use a standard approach, an open approach, or both. The open approach is chosen independently by each modeling group; the only requirement is to use the ocean data provided by Jourdain et al. (2020). For the standard approach, basal melt rates beneath ice shelves are computed as a quadratic function of thermal forcing as described by Favier et al. (2019), with a thermal forcing correction suggested by Jourdain et al. (2020):

$$m(x,y) = \gamma_0 \times \left( \frac{\rho_w c_{pw}}{\rho_i L_f} \right)^2 \times (TF(x,y,z_{\text{draft}}) + \delta T_{\text{sector}}) \times |\langle TF \rangle_{\text{draft} \in \text{sector}} + \delta T_{\text{sector}}|, \tag{6}$$

where $m$ is the melt rate, $TF$ is the thermal forcing, $\gamma_0$ is an empirical coefficient, $\rho_w$ and $c_{pw}$ are the density and specific heat of seawater, and $\rho_i$ and $L_f$ are the density and latent heat of melting of ice. The brackets in $< TF >$ denote the average over a drainage basin or sector, and $\delta T_{\text{sector}}$ is a thermal forcing correction with units of temperature, with one value per sector. The Antarctic Ice Sheet is divided into 16 sectors as shown in Fig. 1. Since this method uses sector-average thermal forcing, it is known as the *nonlocal* parameterization. For the simulations in this paper, the last term in Eq. 6 is modified to $\max(\langle TF \rangle_{\text{draft} \in \text{sector}} + \delta T_{\text{sector}}, 0)$, to prevent spurious melting and freezing in sectors where $\langle TF \rangle_{\text{draft} \in \text{sector}} + \delta T_{\text{sector}} < 0$. The nonlocal quadratic dependence on TF in Eq. (6) is based on the idea that the melt rate is proportional to the local TF and also to the speed of the sub-shelf flow, which in turn is proportional to the regional mean TF.

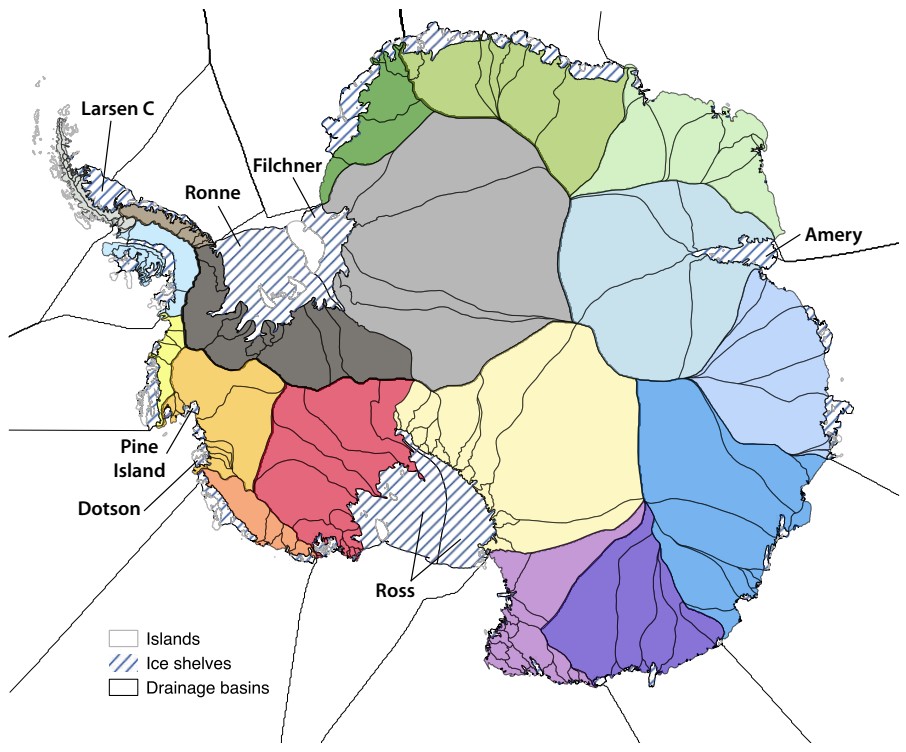

**Figure 1.** Antarctic sectors (colors) defined by Mouginot et al. (2017) and Rignot et al. (2019). Eighteen sectors are shown, but 16 sectors are used in this study, with sectors combined in the Ross and Filchner-Ronne basins. Reprinted with permission from Jourdain et al. (2020, their Fig. 2).

An alternative *local* parameterization is obtained by replacing $\langle TF \rangle_{\text{draft} \in \text{sector}}$ with $TF(x, y, z_{\text{draft}})$ in the last term:

$$m(x,y) = \gamma_0 \times \left( \frac{\rho_w c_{pw}}{\rho_i L_f} \right)^2 \times \left\{ \max \left[ TF(x, y, z_{\text{draft}}) + \delta T_{\text{sector}}, 0 \right] \right\}^2, \tag{7}$$

giving a quadratic dependence on the local thermal forcing. A third parameterization, which we call *nonlocal-slope*, is obtained by multiplying Eq. (6) by $\sin(\theta)$, where $\theta$ is the local angle between the ice-shelf base and the horizontal. At the same time,
5    $\gamma_0$ is increased by a factor of $\sim100$, since $\sin(\theta)$ typically is $\sim10^{-2}$ near grounding lines. This scheme, suggested by Little et al. (2009) and Jenkins et al. (2018), generally gives larger melt rates near grounding lines and lower melt rates near calving fronts, in agreement with observational estimates and ocean simulations (e.g., Rignot et al., 2013; Favier et al., 2019). Total melt over the ice sheet is the same, if $\gamma_0$ is tuned based on a total-melt criterion as in Jourdain et al. (2020). CISM simulations in Seroussi et al. (2020) used this nonlocal-slope parameterization as an open approach.
10    Sub-ice-shelf thermal forcing and melt rates are uncertain, as is the functional relationship between thermal forcing and melt rates. For this reason, $\gamma_0$ and $\delta T_{\text{sector}}$ are not well constrained and can be viewed as tuning parameters. To calibrate Eqs. (6) and (7), Jourdain et al. (2020) used two methods. In the *MeanAnt* method, $\gamma_0$ is chosen so that the total Antarctic melt rate given by the parameterization matches the observational estimates of Depoorter et al. (2013) and Rignot et al. (2013), before

applying basin-scale thermal forcing corrections. Then the $\delta T_{\text{sector}}$ values are chosen to reproduce the estimated mean melt rate in each sector. The *PIGL* method is similar, except that the melt rate targets are given by observations in the Amundsen sector near the grounding line of Pine Island Glacier (Rignot et al., 2013). The $\gamma_0$ values for the PIGL calibration are an order of magnitude larger than those obtained by the MeanAnt method, but the present-day average melt rates in each basin are

similar with the two calibrations. See Jourdain et al. (2020) for more details on these parameterizations and calibrations.

The simulations in our paper differ from the ISMIP6 protocols in the treatment of $\delta T_{\text{sector}}$. Although we use calibrated values of $\gamma_0$, we tune $\delta T_{\text{sector}}$ to better match the thickness of grounded and floating ice near the grounding line, as described in Sect. 3.1. This procedure is motivated by the fact that ice thickness is better constrained than melt rates, and future projections are sensitive to the initial grounding-line position (Seroussi et al., 2019). Thus, $\delta T_{\text{sector}}$ is calibrated to obtain melt rates that

will drive the ice toward the observed extent and thickness, but the basin-average melt rates will differ from observational estimates.

In summary, we have presented three basal melt parameterizations (local, nonlocal, and nonlocal-slope) and two calibration methods (MeanAnt and PIGL), giving six possible combinations (local-MeanAnt, nonlocal-PIGL, etc.). The values of $\gamma_0$ for each combination, shown in Tab. 1, correspond to the median values in Jourdain et al. (2020). These combinations form the

basis for the spin-up and forcing experiments described in Sects. 3 and 4.

**Table 1.** Calibrated $\gamma_0$ values for the three quadratic parameterizations and two calibration methods (in $\text{m y}^{-1}$).

| Parameterization | Calibration | $\gamma_0$ |
|---|---|---|
| local | MeanAnt | $1.11 \times 10^4$ |
| nonlocal | MeanAnt | $1.44 \times 10^4$ |
| nonlocal-slope | MeanAnt | $2.06 \times 10^6$ |
| local | PIGL | $4.95 \times 10^4$ |
| nonlocal | PIGL | $1.59 \times 10^5$ |
| nonlocal-slope | PIGL | $5.37 \times 10^6$ |

## 3 Model spin-up

### 3.1 Spin-up procedure

When spinning up the model, we aim to reach a stable state with minimal drift under modern forcing, while also simulating ice sheet properties that agree with observations. It is challenging to achieve both goals at once. Long spin-ups typically yield a

steady-state ice sheet with large biases in thickness, velocity, and/or ice extent, while initialization methods that assimilate data to match present-day conditions often have a large initial transient. We compromised by using a hybrid method similar to that of Pollard and DeConto (2012), running a long spin-up while continually nudging the ice sheet toward the observed thickness.

Beneath grounded ice, we adjust $C_p(x,y)$, a poorly-constrained, spatially-varying coefficient in the basal friction law, Eq. (1) or (2). This coefficient controls the power-law behavior, with higher $C_p$ giving more friction and slower sliding. Wherever grounded ice is present, $C_p$ is initialized to $20{,}000\,\mathrm{Pa\,m^{-1/3}\,y^{-1}}$. During the spin-up, $C_p$ is decreased where $H > H_\mathrm{obs}$ and increased where $H < H_\mathrm{obs}$, based on the idea that lower friction will accelerate the ice and lower the surface, while higher friction will slow the ice and raise the surface. The rate of change of $C_p$ is given by

$$\frac{dC_p}{dt} = -\frac{C_p}{H_0}\left[\frac{(H - H_\mathrm{obs})}{\tau_c} + 2\frac{dH}{dt}\right], \tag{8}$$

where $H_\mathrm{obs}$ is an observational target, $H_0 = 100$ m is a thickness scale, and $\tau_c = 500$ y is a time scale for adjusting $C_p$. The first term in brackets nudges $H$ toward $H_\mathrm{obs}$, and the second term damps the nudging to prevent overshoots. We hold $C_p$ within a range between $10^2$ and $10^5\,\mathrm{Pa\,m^{-1/3}\,y^{-1}}$. Smaller values can lead to excessive sliding speeds when the basal friction approaches zero. With $C_p$ at its maximum value, basal sliding is close to zero, and there is little benefit in raising $C_p$ further.

This method works well at keeping most of the grounded ice near the observed thickness. Also, since $C_p$ is independent of the ice thermal state, we remove low-frequency oscillations associated with slow changes in basal temperature, resulting in a better-defined steady state. In forward runs, however, $C_p(x,y)$ is held fixed and cannot evolve in response to changes in basal temperature or hydrology. As a result, we can have unphysical basal velocities when the ice dynamics differs from the spun-up state. Also, the tuning of $C_p$ can compensate for other errors. For example, if the prescribed topography is missing pinning points near the grounding line, the ice will be biased thin, and $C_p$ can be driven to high values to make up for the lack of buttressing.

A different method is needed to initialize floating ice shelves, where $C_p = 0$. In CISM simulations for initMIP-Antarctica (Seroussi et al., 2019) and the ISMIP6 projections (Seroussi et al., 2020), basal melt rates were obtained by local nudging. That is, an equation similar to Eq. (8) was used to nudge $H$ toward $H_\mathrm{obs}$ by adjusting the melt rate $m(x,y)$ in each floating grid cell. In climate change experiments, the spun-up melt rates were added to melt rate anomalies in each basin. This method yields ice-shelf thicknesses and grounding-line locations that agree well with observations, but it overfits the observations, giving noisy melt rates that compensate for other errors without being tied to ocean temperatures. Other complications arise when applying this spin-up method to the ISMIP6 projections, which prescribe thermal forcing anomalies instead of melt rate anomalies. In climate change experiments, a melt rate anomaly computed from the thermal forcing anomaly must be added to the spun-up melt rate, instead of computing the evolving melt rate directly from the evolving thermal forcing.

For the simulations described here (which were done too late to be included in Seroussi et al., 2020), we take a different approach. During the spin-up, basal melt rates are computed directly from the thermal forcing, using the climatological data set and melt parameterizations described in Sect. 2.2. When we use the calibrated values of both $\gamma_0$ and $\delta T_\mathrm{sector}$, many grounding lines drift far from their observed locations. The drift can be reduced (but not eliminated) by continually adjusting $\delta T_\mathrm{sector}$ in each of 16 sectors (see Fig. 1), nudging toward an ice thickness target in a region near the grounding line. Here, "near the grounding line" is defined as having $f_\mathrm{float}$ (see Eq. (5)) with a magnitude less than a prescribed value:

$$|f_\mathrm{float}| = \left|-b - \frac{\rho_i}{\rho_w}H\right| < H_\mathrm{thresh}, \tag{9}$$

where $H_{\mathrm{thresh}}$ is a prescribed threshold thickness. We set $H_{\mathrm{thresh}} = 500$ m so that the target region includes most of the ice likely to switch between floating and grounded during a spin-up, without extending too far upstream.

During the spin-up, $\delta T_{\mathrm{sector}}$ is adjusted as follows:

$$\frac{d(\delta T_{\mathrm{sector}})}{dt} = -\frac{1}{\tau_m m_T}\left[\frac{\bar{H} - \bar{H}_{\mathrm{obs}}}{\tau_m} + 2\frac{d\bar{H}}{dt}\right], \tag{10}$$

where $\bar{H}$ is the mean ice thickness over the target region, $\bar{H}_{\mathrm{obs}}$ is the observational target for this region, $m_T = 10\,\mathrm{m\,y^{-1}\,^\circ C^{-1}}$ is a scale for the rate of change of basal melt rate with temperature, and $\tau_m = 100$ y is a timescale for adjusting $\delta T$. As in Eq. (8), the first-derivative term damps oscillations. In most basins, this adjustment keeps $\bar{H}$ close to $\bar{H}_{\mathrm{obs}}$ with $|\delta T_{\mathrm{sector}}| < 1\,^\circ\mathrm{C}$. In basins where $\bar{H} < \bar{H}_{\mathrm{obs}}$ no matter how much $\delta T_{\mathrm{sector}}$ is lowered, $\delta T_{\mathrm{sector}}$ is capped at $-2\,^\circ\mathrm{C}$.

Given the calibrated thermal forcing, the melt rate $m$ is computed for floating grid cells using one of the three parameteriza-
tions described in Sect. 2.2. In partly grounded cells, $m$ is weighted by the floating ice fraction. In shallow cavities, following Asay-Davis et al. (2016), $m$ is weighted by $\tanh(H_c/H_{c0})$, where $H_c$ is the cavity thickness and $H_{c0} = 50$ m is an empirical depth scale.

For each spin-up, the ice sheet is initialized to the present-day thickness using the BedMachineAntarctica data set (Morlighem et al., 2019). The internal ice temperature is initially set to an analytic vertical profile and then evolves freely under advection.
The surface mass balance and surface temperature are provided by a regional climate model, RACMO2.3 (van Wessem et al., 2018), and the geothermal heat flux is from Shapiro and Ritzwoller (2004). As described above, $C_p$ is adjusted in grounded cells to better match the observed local ice thickness, and $\delta T_{\mathrm{sector}}$ is adjusted in each of the 16 sectors to match the observed thickness of ice near the grounding line.

Since the spun-up ice sheet represents a quasi-equilibrium state before the mass loss of the past few decades, we assigned
a date of 1950 to the spin-up. In sectors where the thermal forcing in the climatology exceeds the forcing that was typical in the mid 20[th] century and before, $\delta T_{\mathrm{sector}}$ can compensate by becoming more negative. The ice sheet is out of equilibrium today, especially in the Amundsen sector where glaciers are losing mass. We compensated by increasing the target thickness for this sector by the equivalent of 1000 Gt of ice, roughly the mass lost by Pine Island Glacier in recent decades (Rignot et al., 2019). With this adjustment, the advance of the Pine Island grounding line from the present location to a more stable position
downstream (which takes place in all spin-ups) does not have to be compensated by retreat elsewhere in the sector. Otherwise, we assumed that the recent mass loss is small enough that the present-day ice thickness is an appropriate target.

The spin-ups are run for 20,000 model years, allowing the ice sheet to approach steady state; at this time the total mass is changing by $< 1\,\mathrm{Gt\,y^{-1}}$. We evaluate the spun-up state below.

## 3.2 Spun-up model state

To initialize the standard projection experiments described in Sect. 4.1, we carried out six Antarctic spin-ups, each with a different pairing of the three basal melt parameterizations (local, nonlocal, and nonlocal-slope) and the two calibrations (MeanAnt and PIGL). Although $\gamma_0$ varies widely among the different spin-ups (Tab. 1), the spun-up states are similar across parameterizations and calibrations, because of the freedom to adjust $C_p$ and $\delta T_{\mathrm{sector}}$ independently for each run to match the observed

thickness. The simulated thickness, velocity, and ice extent are broadly in agreement with observations, but with some persistent biases. Some biases can be attributed to errors in ocean thermal forcing (which is treated simply by the basin-scale melt parameterizations) and seafloor topography (e.g., an absence of pinning points, resulting in grounding-line retreat that is compensated by spurious ocean cooling). Pinning points could either be missing in the high-resolution (0.5 km) topographic data set (Morlighem et al., 2019), or smoothed away when interpolating to the coarser CISM grid.

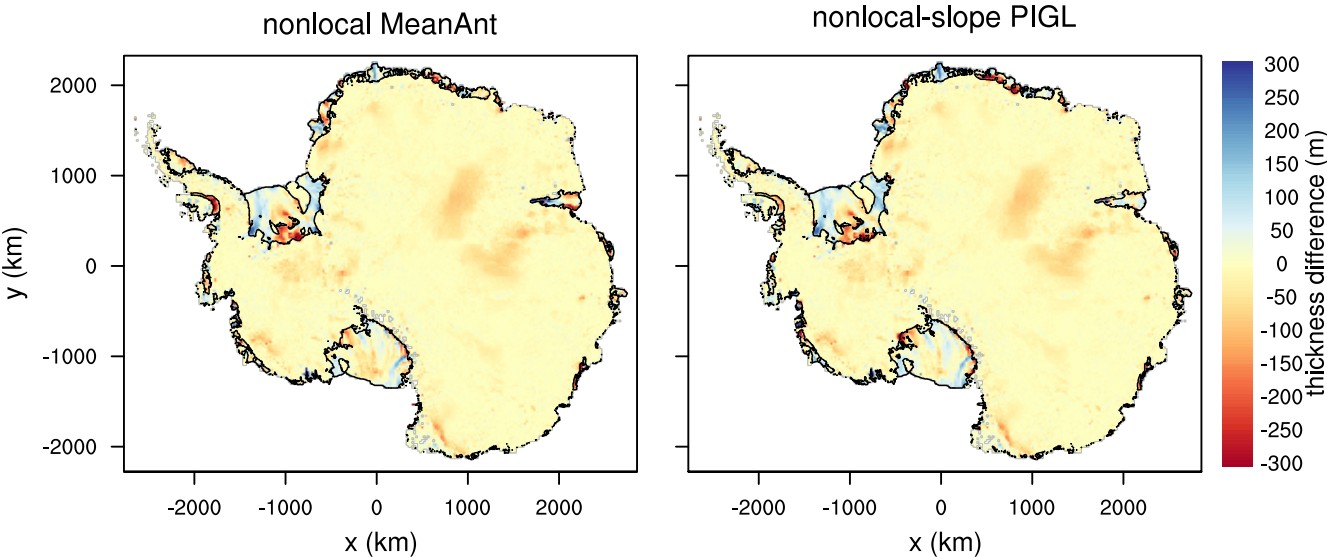

**Figure 2.** Change in ice thickness (m) between the start and end of two 20-ky CISM spin-ups at 4-km resolution, for two combinations of melt parameterization and calibration: (left) nonlocal-MeanAnt and (right) nonlocal-slope-PIGL. The initial state is based on observations (Morlighem et al., 2019), and positive values indicate where the spun-up ice state is thicker than observed. Black lines show boundaries of floating ice at the end of each spin-up.

Figure 2 shows the difference between the final ice thickness and the observed thickness for two spin-ups: nonlocal-MeanAnt and nonlocal-slope-PIGL. Figure 3 is like Fig. 2, but focused on the Amundsen sector. These two melt schemes are interesting to compare because, as shown in Sect. 4.1, the latter is much more sensitive to ocean warming. The spun-up states, however, are similar, with some shared biases. In both spin-ups, Thwaites Glacier is too thin near the grounding line, while Pine Island Glacier is too thick, as are the Crosson and Dotson Ice Shelves. The Filchner-Ronne Ice Shelf is thin near the central grounding line but thick elsewhere, and there are similar thick/thin patterns for the Ross Ice Shelf. There are also a few differences; for example, the George VI Ice Shelf and the seaward parts of the Amery and Larsen C shelves are thinner in the nonlocal-MeanAnt run.

Figure 4 compares ice surface speeds from the end of the nonlocal-MeanAnt spin-up to observed surface speeds (Rignot et al., 2011). Overall, the agreement with observations is very good for both grounded and floating ice, even though the model is nudged toward observed velocities only indirectly, via the thickness field. In general, good agreement in thickness implies agreement in velocity as well, at least when using BedMachineAntarctica thicknesses, which are obtained using the mass

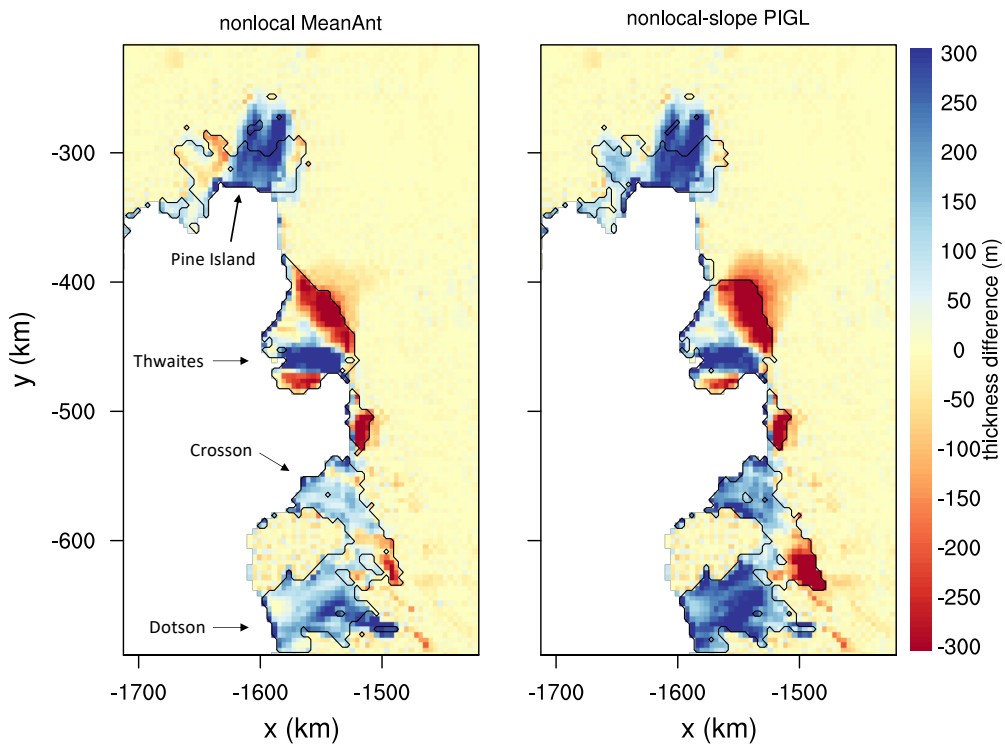

**Figure 3.** Same as Fig. 2, but focused on the Amundsen sector.

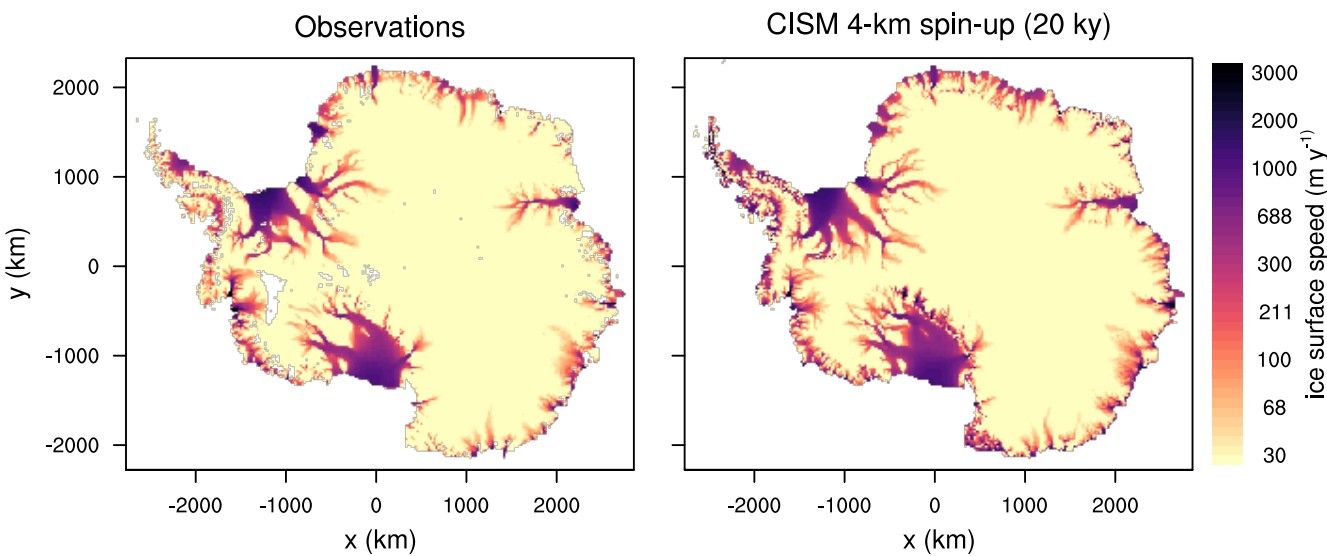

**Figure 4.** Antarctic ice surface speed ($\mathrm{m\,y^{-1}}$, log scale) from (left) observations (Rignot et al., 2011) and (right) the end of a 20-ky CISM spin-up at 4-km resolution using the nonlocal-MeanAnt melt parameterization and calibration.

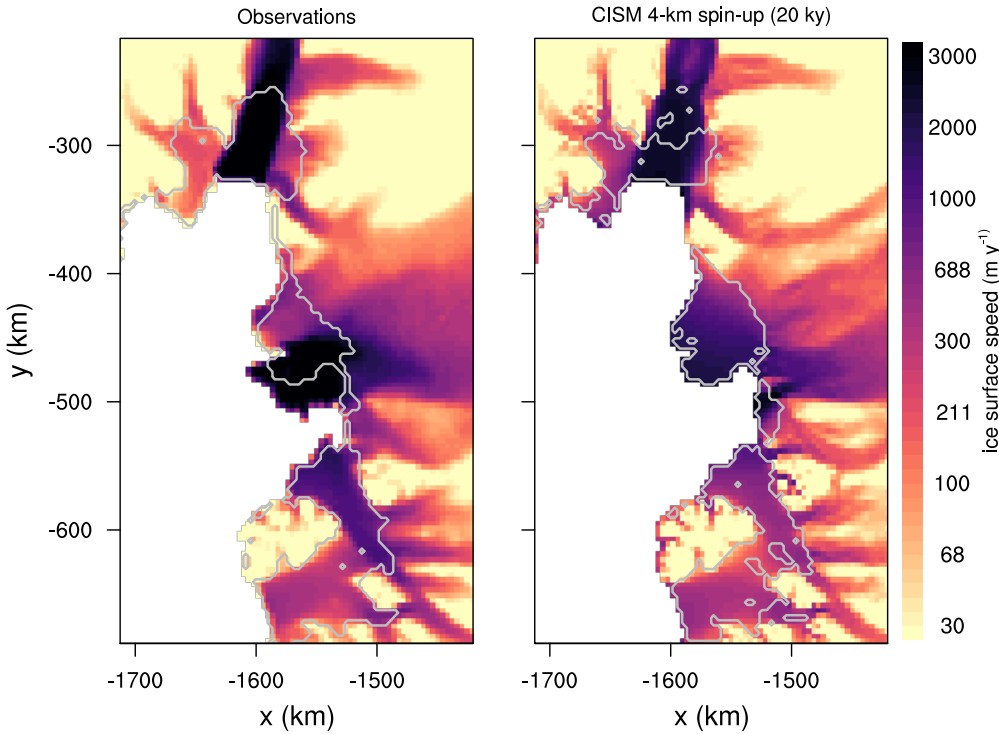

**Figure 5.** Same as Fig. 4, but focused on the Amundsen sector. Light gray lines show boundaries of floating ice.

conservation method of Morlighem et al. (2011). One place of disagreement is the Kamb Ice Stream (on the Siple Coast of the Ross Ice Shelf), which is clearly visible in the model but absent in the observations, having stagnated in the 1800s and left a signature in the thickness field (Ng and Conway, 2004).

For the same spin-up, Fig. 5 shows ice surface speeds in the Amundsen sector, including Pine Island and Thwaites Glaciers. The observations reveal a dual structure in the Thwaites velocity field, with a fast western core where the glacier flows into the Thwaites Ice Tongue, and slower speeds in the east where flow is impeded by an ice rise (see Fig. 1 in Rignot et al., 2014). The model, however, lacks a sharp division between east and west, and the Thwaites grounding line is retreated compared to observations, perhaps because interaction with seafloor topography is not captured accurately. At the same time, the Pine Island grounding line is advanced compared to observations. The grounding lines of both glaciers have been retreating since at least the 1990s (Rignot et al., 2014), and the spin-up method is not well suited to initializing dynamic grounding lines in their observed locations.

Figure 6 shows the values of the thermal forcing correction $\delta T_{\mathrm{sector}}$ for each sector at the end of the six spin-ups. In most sectors, the corrections are modest ($< 1°$C in magnitude). For the Aurora sector in East Antarctica, however, $\delta T$ is strongly negative, ranging from $-1.0°$C to $-1.7°$C depending on the melt scheme and calibration. Also, $\delta T_{\mathrm{sector}}$ is consistently negative for the Amundsen sector, with values from $-0.7°$C to $-1.8°$C. In both sectors, significant cooling is needed to curtail the

grounding-line retreat that occurs under climatological thermal forcing. As a result, the spun-up melt rates in these regions are lower than observed. Total sub-shelf melting for Antarctica at the end of the spin-up ranges from 625 to 744 $\mathrm{Gt\,y^{-1}}$, about half the values estimated by Depoorter et al. (2013) and Rignot et al. (2013). Negative values of $\delta T_{\mathrm{sector}}$ might be compensating for other errors, such as biases in the climatology or the failure of the melt scheme to deliver ocean heat to the right places. Another

5   possibility, considered in Section 4.2.5, is that for some sectors, the thermal forcing derived from the 1995–2018 climatology exceeds the forcing that was typical in the mid $20^{\mathrm{th}}$ century and before. In this case, $\delta T_{\mathrm{sector}} < 0$ would be correcting for the recent warming, to generate melt rates closer to preindustrial values.

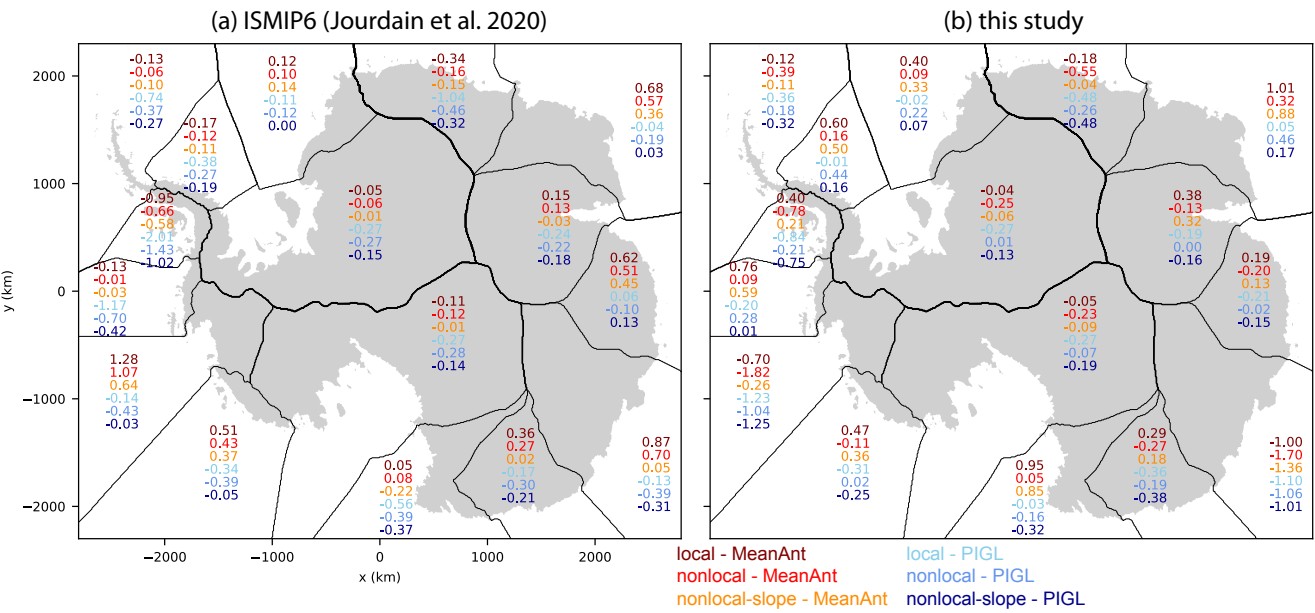

**Figure 6.** Values of the thermal forcing correction $\delta T_{\mathrm{sector}}$ (°C) in each of 16 Antarctic sectors for six combinations of melt parameterization (local, nonlocal, and nonlocal-slope) and calibration (MeanAnt and PIGL). (a) ISMIP6 values, calibrated based on sub-shelf melt rates. Values for the local and nonlocal parameterizations were reported by Jourdain et al. (2020); values for nonlocal-slope were computed for this study. (b) Values obtained during CISM spin-ups.

## 4   Results of projection experiments

After spinning up the model as described in Sect. 3, we ran a series of projection experiments. Section 4.1 gives the results

10  of what we call the standard experiments, which were run on a 4-km grid using the DIVA solver with a basal friction power law, and without prescribing ice-shelf collapse. We ran several projections with idealized thermal forcing, followed by many projections with TF derived from ESMs as in ISMIP6. Section 4.2 describes the results of sensitivity experiments in which the

grid resolution, stress-balance approximation, ice-shelf extent, and basal sliding law are varied. We also explore the sensitivity of the Amundsen sector to parameter changes.

## 4.1 Standard experiments

We first show results from experiments with idealized, spatially uniform thermal forcing anomalies. These experiments help
identify the regions that are most vulnerable to a given ocean warming. In the first set of experiments, the TF anomaly is ramped up linearly by $1°$C between 1950 and 2100 and then is held fixed at $1°$C until 2500. The second set is like the first, except that the TF anomaly is ramped up by $2°$C. From Eqs. (6) and (7), the melt rate is a quadratic function of thermal forcing, and therefore the incremental change in melt rate is proportional to the product of the initial TF and the TF anomaly. Thus, ice retreat is favored where TF and/or its anomaly are large. The ice-sheet response also is sensitive to the bed topography and basal friction, among other factors.

Figure 7 shows the difference in ice thickness between 2500 and 1950 for a TF anomaly of $1°$C, using the nonlocal and nonlocal-slope melt parameterizations with the MeanAnt and PIGL calibrations. Results with the local melt scheme (not shown) are similar to those with the nonlocal scheme. All four experiments show significant thinning of the four largest ice shelves (Ross, Filchner-Ronne, Amery, and Larsen C) as well as small shelves (e.g., in the Amundsen sector and East Antarctica). Thinning of grounded ice is concentrated in the Filchner-Ronne and Ross sectors of West Antarctica, and is modest for the Amundsen sector. The loss of ice mass above flotation, which contributes directly to SLR, ranges from 479 mm sea-level equivalent (s.l.e.) for nonlocal-MeanAnt to 1726 mm s.l.e. for nonlocal-slope-PIGL. For the Ross and western Ronne sectors, the PIGL runs have much more grounding-line retreat and mass loss than do the MeanAnt runs. The nonlocal runs, unlike the nonlocal-slope runs, have substantial basal melting far from the grounding line, leading to calving-front retreat. For nonlocal-PIGL, the four largest shelves disappear by 2500 with the increased basal melting. For the nonlocal-slope runs, in which the increased melting is concentrated near the grounding line, these shelves thin but remain intact.

Figure 8 shows the ice thickness change for a uniform TF anomaly of $2°$C. With a doubling of the anomaly, the SLR contribution more than doubles, reaching nearly 4 m for nonlocal-slope-PIGL. Much of the nonlinearity stems from marine-ice collapse in the Amundsen sector. For nonlocal-slope-MeanAnt, the Pine Island Glacier basin collapses, and for both PIGL runs, the Thwaites basin also collapses, raising sea level by an additional $\sim 1$ m compared to runs without Amundsen collapse. For nonlocal-slope-PIGL, the collapse creates a continuous ice shelf between the Ross and Amundsen Seas. These runs illustrate a threshold of instability for the Amundsen sector, which we discuss further in Sect. 4.2.5. A threshold in the range $1–2°$ is consistent with the modeling study of Rosier et al. (2020), who found that Pine Island Glacier collapses with ocean warming greater than $1.2°$ (using a local melt parameterization similar to Eq. (7)). A TF anomaly of $2°$C also drives significant ice loss in East Antarctica, including the Wilkes, Aurora, and Amery sectors, for all melt schemes except nonlocal-MeanAnt.

Next, we analyze projection experiments with ocean thermal forcing anomalies derived from six CMIP ESMs. The four CMIP5 models are CCSM4, HadGEM2-ES (hereafter HadGEM2), MIROC-ESM-CHEM (hereafter MIROC), and NorESM1-M (hereafter NorESM1). These were among the six top-ranking models chosen by Barthel et al. (2020). CCSM4, MIROC, and NorESM1 were used for the ISMIP6 Tier 1 experiments, and we added HadGEM2 (a Tier 2 selection) to sample a model

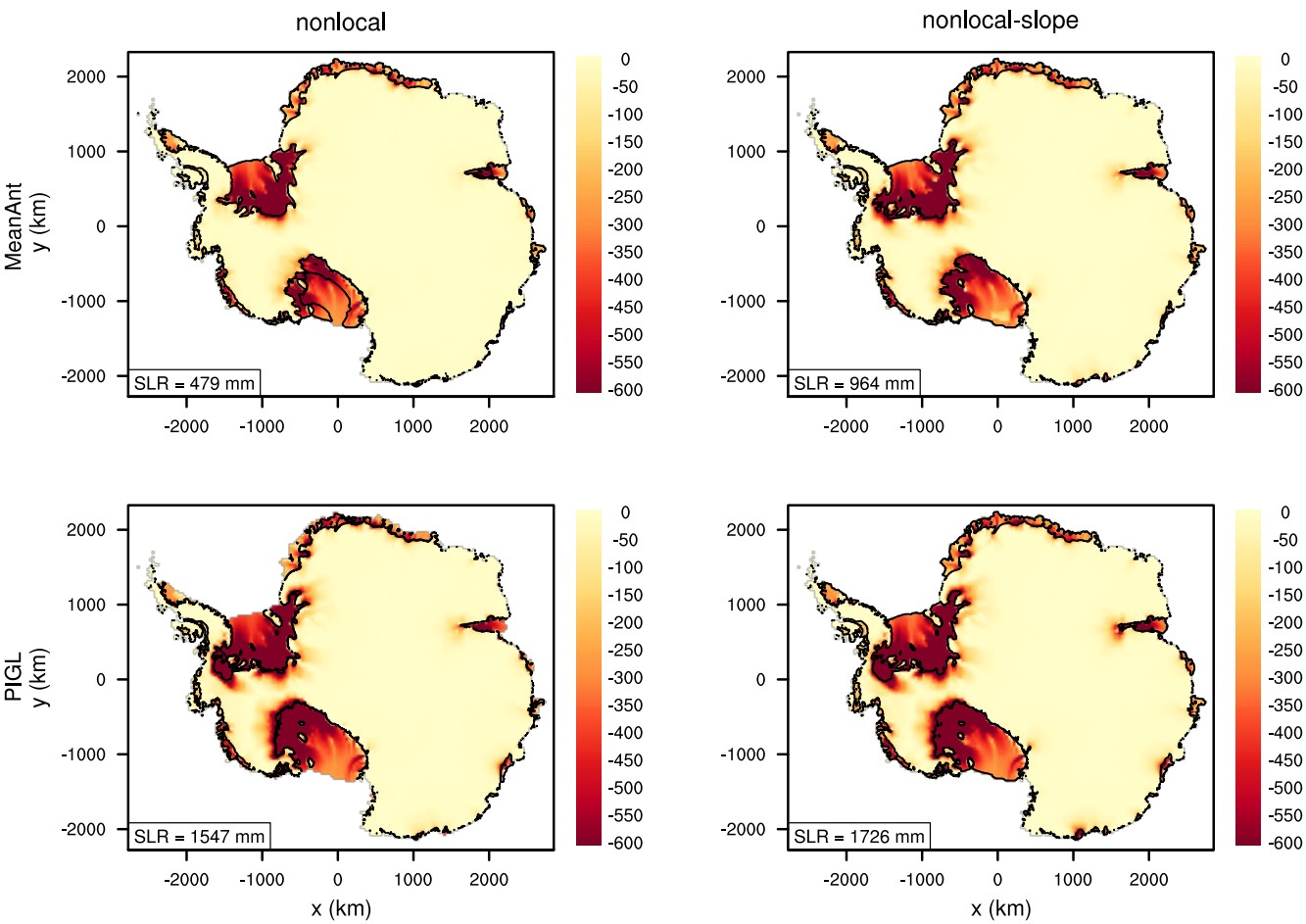

**Figure 7.** Difference in ice thickness (m) between years 2500 and 1950 in experiments with idealized ocean forcing. Starting from the spun-up state, thermal forcing increases by 1°C between 1950 and 2100 and is held fixed thereafter. The two columns correspond to the nonlocal and nonlocal-slope melt parameterizations, and the two rows to the MeanAnt and PIGL calibrations. Black lines show boundaries of floating ice at the end of each run. In the nonlocal-PIGL run (lower left), black lines at the present-day calving fronts of several large ice shelves are absent, indicating shelf collapse. In the nonlocal-MeanAnt run (upper left), the Ross Ice Shelf partly collapses. Boxes in the lower left of each panel show the SLR contribution from the loss of grounded ice.

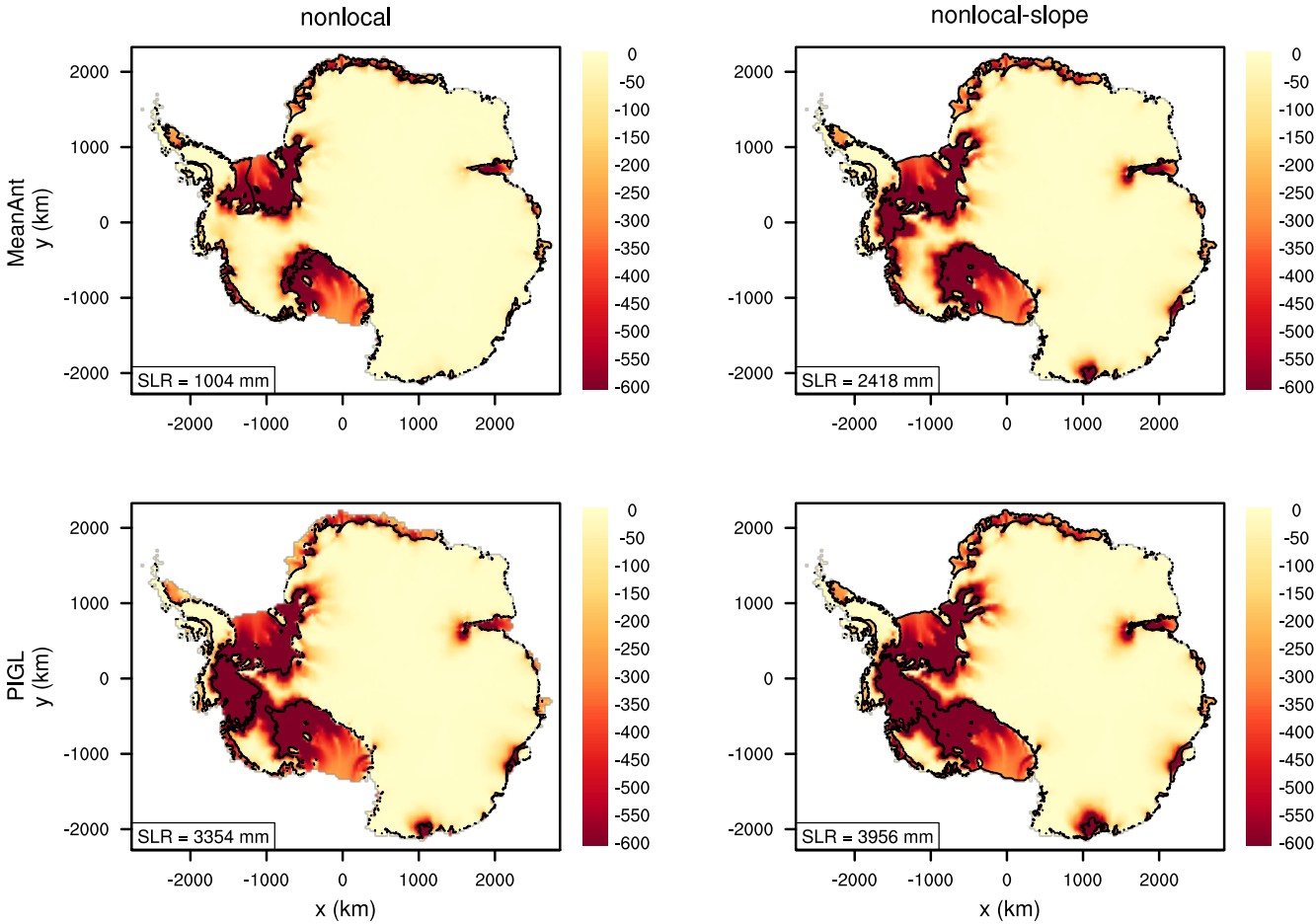

**Figure 8.** Same as Fig. 7, but with thermal forcing increasing by $2°C$ between 1950 and 2100 and held fixed thereafter. In the nonlocal runs (upper and lower left), black lines are absent at the present-day calving fronts of several ice shelves, indicating shelf collapse.

with relatively high ocean warming. The two CMIP6 models are CESM2 and UKESM, which are successors of CCSM4 and HadGEM2, respectively. All the ESMs ran high-end emissions scenarios: RCP8.5 for the CMIP5 models and SSP5-85 for the CMIP6 models. Each spun-up ice sheet state is run forward with ocean forcing from each of the six ESMs.

As described in Sect. 2.2, the thermal forcing is derived by adding ESM temperature and salinity anomalies to a background climatology, and then extrapolating into sub-shelf cavities. Figure 9 shows the thermal forcing at ocean level 9 ($z = -510$ m) for each ESM, averaged over 2081-2100. For each model, the forcing is relatively uniform across a given basin. The Amundsen sector is the warmest of the three large WAIS basins, as a result of warm CDW penetrating into sub-shelf cavities (Holland et al., 2020). HadGEM2 and UKESM are both relatively warm in the Filchner-Ronne basin, and less warm in the Ross basin. NorESM1 is moderately warm for Filchner-Ronne and especially warm for Ross. CCSM4 and (to a lesser extent) CESM2 are

moderately warm for Filchner-Ronne, but both are fairly cold for Ross, while MIROC is moderately warm for Ross and cold for Filchner-Ronne. These patterns are reflected in the ice loss discussed below.

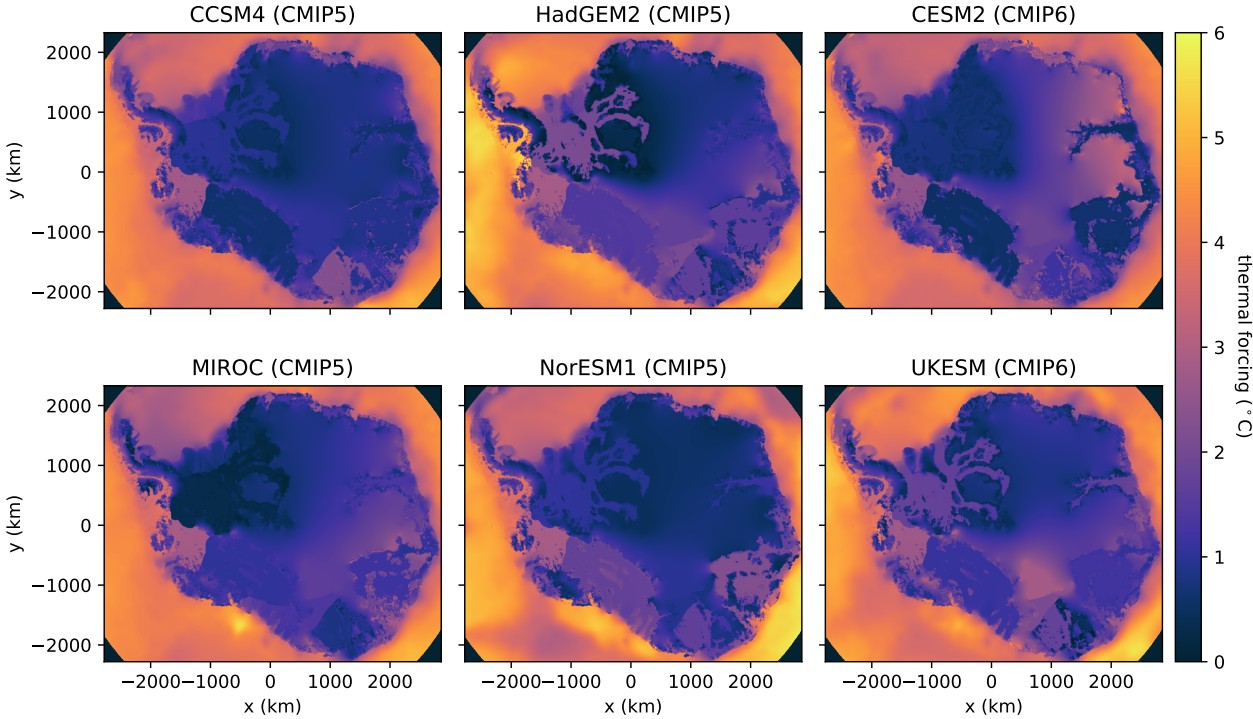

**Figure 9.** Ocean thermal forcing (°C) at $z = -510$ m, averaged over 2081–2100, for the four CMIP5 models and two CMIP6 models used in ocean-forced projection experiments. Thermal forcing is used to compute basal melt rates in grid cells with floating ice.

Figure 10 shows the TF anomaly at $z = -510$ m for each ESM, averaged over 2081-2100 and with respect to the 1941–1960 average. For the Filchner-Ronne basin, the anomalies are largest ($\sim 2°$C) for HadGEM2 and UKESM, while NorESM1 has the largest anomalies for the Ross basin. The anomalies are $\sim 1°$C or less for the Amundsen sector. Thus, most of the Amundsen thermal forcing in Fig. 9 is already present in the contemporary (1995–2018) background climatology, to which ESM anomalies are added. To the extent that the Amundsen sector has warmed in recent decades, the ESMs appear not to fully capture the warming, perhaps because they are too coarse to simulate CDW-driven warming. Also, an ESM might already have a warm Amundsen sector (with CDW having access to the sub-shelf cavity) in its mid-20[th]-century climate, with the result that subsequent warming is small.

Each projection experiment is run for 550 years starting at the end of 1950, the nominal date of the spin-up. In the ISMIP6 protocols, the first 64 years (to the start of 2015) constitute a historical run and the remainder a projection run, but for our simulations the historical and projection periods are forced in the same way. From 1951–2100, we apply the annual-mean thermal forcing provided by ISMIP6, consisting of ESM anomalies added to the background climatology. After 2100, we

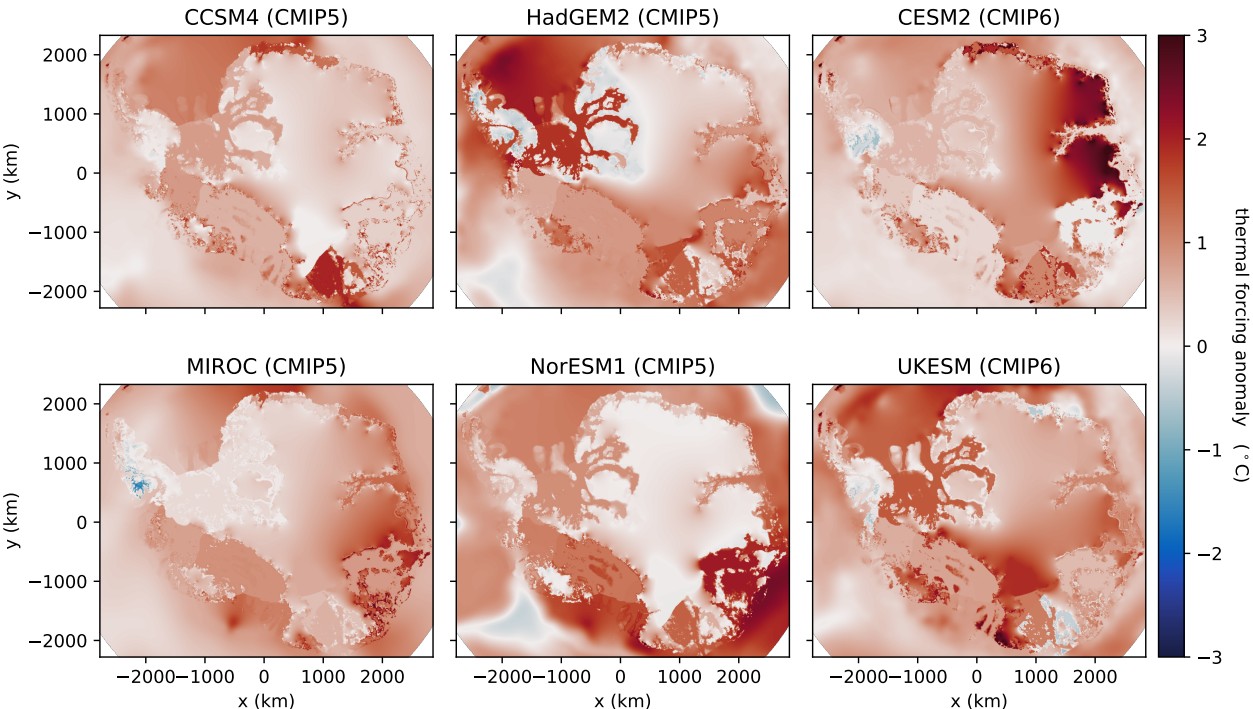

**Figure 10.** Ocean thermal forcing anomaly ($^\circ$C) at $z = -510$ m for the four CMIP5 models and two CMIP6 models used in ocean-forced projection experiments. The anomaly is averaged over 2081–2100 and is computed with respect to the base period 1941–1960.

cycle repeatedly through the last 20 years of forcing, 2081–2100, to evaluate the committed SLR associated with a late-21st-century climate. For comparison, we ran a subset of forcing experiments using a fixed climatology, computed as the mean of the 2081–2100 thermal forcing. Cycling through the annual forcing drives greater mass loss (by $\sim 15\%$) than does the fixed climatology, suggesting that years with high thermal forcing have a disproportionate influence on long-term mass loss, because
of the quadratic relationship between thermal forcing and melt rates.

Figure 11 shows time series of ocean-forced SLR for 36 projection experiments, with one panel per parameterization-calibration pair. Along with the ESM-forced runs, each plot shows the SLR from a control run with no change in thermal forcing compared to the spin-up. In this figure (and in figures below, unless otherwise specified), results from control runs are not subtracted from projection runs. SLR during the control runs is minimal, confirming that the spin-ups are close to steady
state.

Several patterns emerge. First, SLR starts slowly for all experiments, then accelerates near the end of the 21st century, suggesting that a threshold has been crossed based on some magnitude or duration of thermal forcing. After 2100, SLR is fairly linear and shows no sign of leveling off after 500 years. This is consistent with retreat driven by MISI in large reverse-sloping basins. Once the retreat is under way, it continues until reaching a stable seafloor configuration, up to several hundreds
of km upstream.

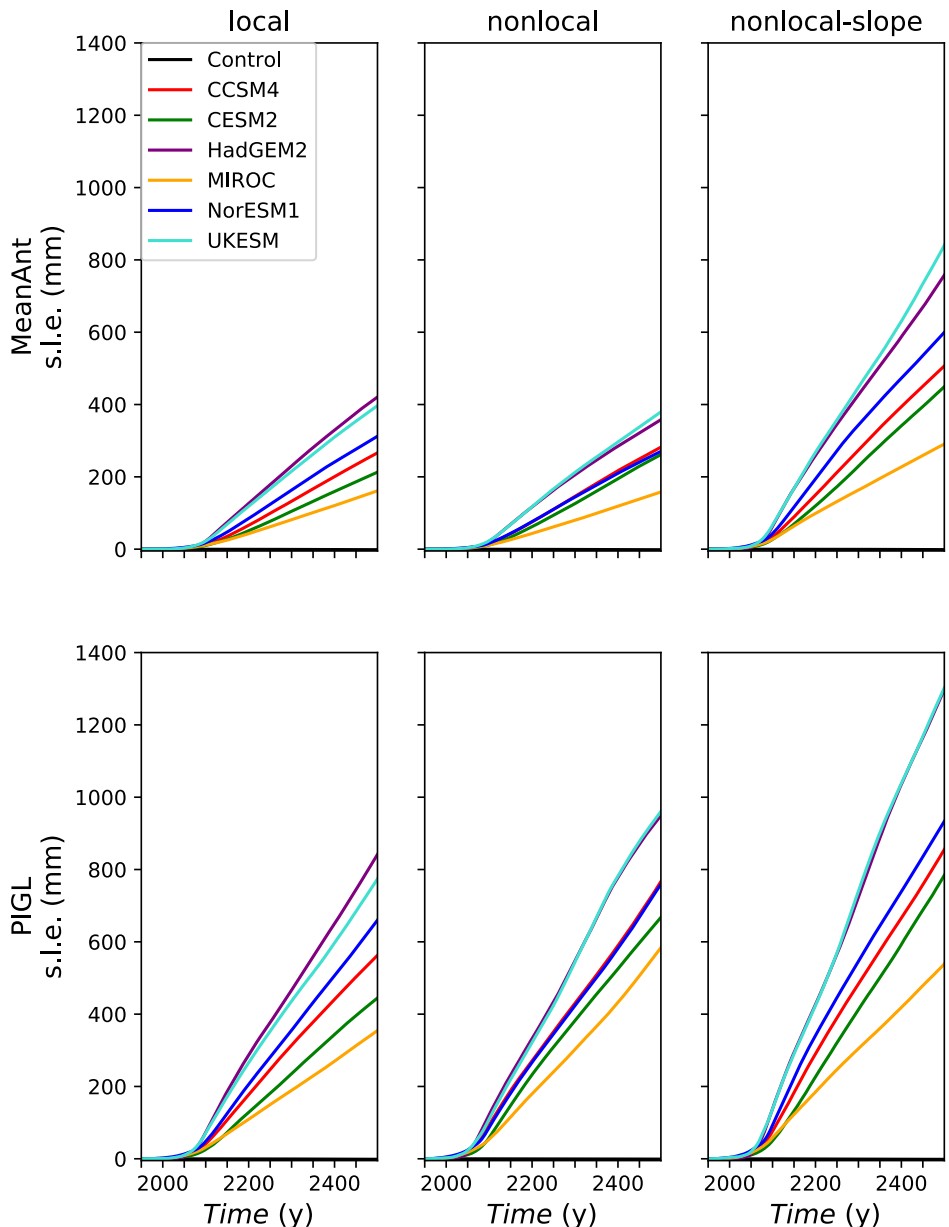

**Figure 11.** Sea-level rise (mm s.l.e.) from Antarctic ice loss in 550-year ocean-forced projection experiments. The three columns correspond to the three melt parameterizations, and the two rows to the two calibrations. Each panel shows the SLR response to ocean forcing from six AOGCMs, along with an unforced control run.

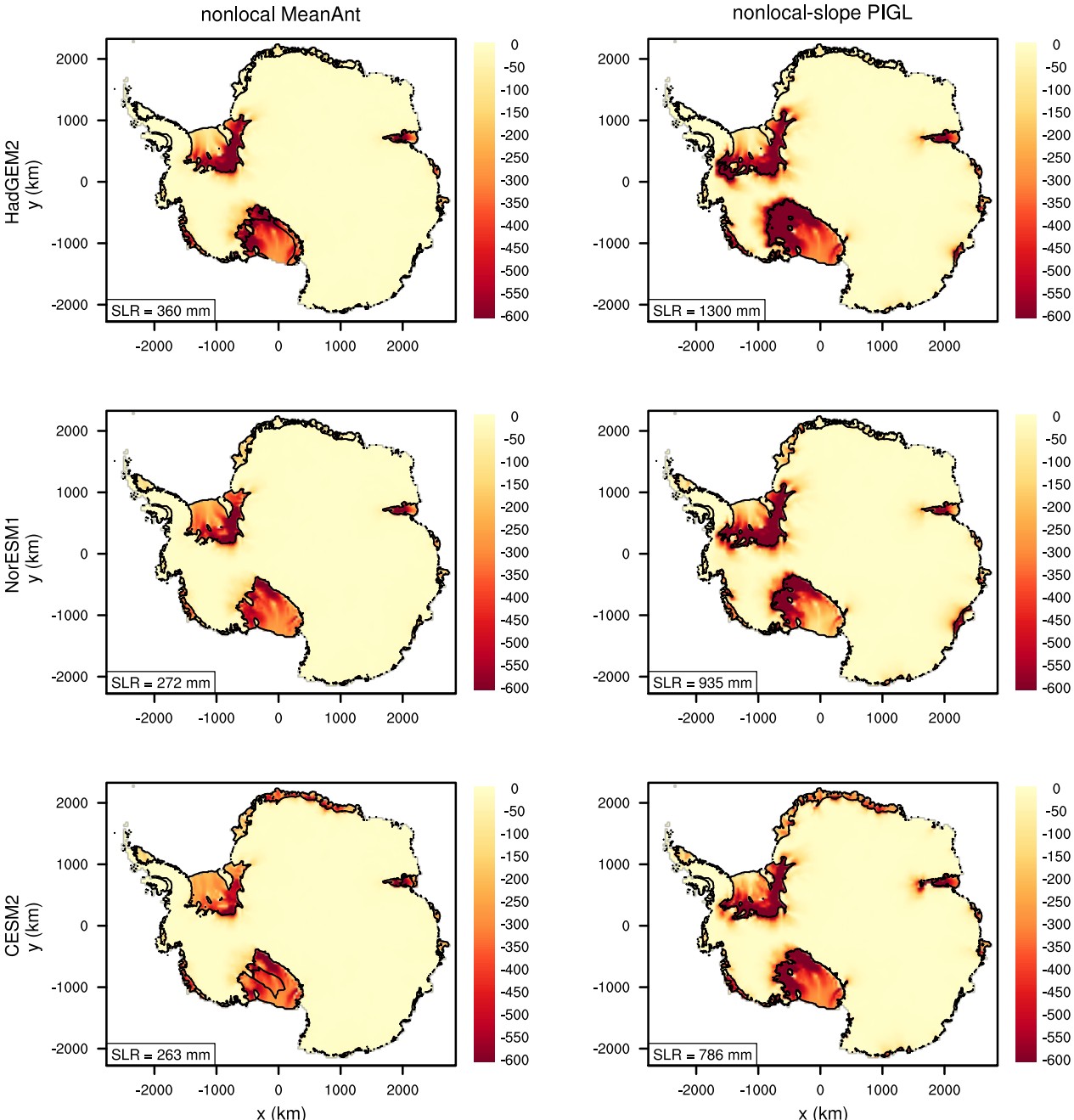

**Figure 12.** Difference in ice thickness (m) between years 2500 and 1950 in ocean-forced projection experiments. The left and right columns show results from the nonlocal-MeanAnt and nonlocal-slope-PIGL melt combinations, respectively. The three rows show results with ocean forcing from HadGEM2, NorESM1, and CESM2. Black lines show boundaries of floating ice at the end of each run. In the nonlocal HadGEM2 run (upper left), retreat of the black line from the Ross calving front indicates shelf collapse. In the nonlocal CESM2 run (lower left), the enclosed black contour shows that part of the Ross Ice Shelf interior has collapsed, without calving-front retreat. Boxes in the lower left of each panel show the SLR contribution from the loss of grounded ice.

Second, the ice sheet is more sensitive to some melt combinations than others. The results from local and nonlocal parameterizations are similar for the MeanAnt runs, but the nonlocal scheme gives greater SLR (by $\sim$100–200 mm) for the PIGL runs. The nonlocal-slope parameterization yields more SLR than local and nonlocal, and the PIGL calibration gives more SLR than MeanAnt. The greater sensitivity for the nonlocal-slope runs can be attributed to larger melt rates at steep slopes near

grounding lines, where melting is most effective in driving ice retreat. The greater sensitivity of PIGL compared to MeanAnt follows from the larger $\gamma_0$. During the spin-ups, PIGL runs acquire more negative values of $\delta T_{\text{sector}}$ than MeanAnt (Fig. 6), to compensate for greater $\gamma_0$. Then, for a given TF anomaly during the forward runs, the increase in $m$ is proportional to $\gamma_0$ and thus is larger for PIGL.

Third, the CMIP model rankings are consistent across melt schemes. HadGEM2 and UKESM are the warmest models and

drive the most SLR, followed in most cases by NorESM1. CCSM4 and CESM2 are next, and MIROC yields the least SLR. This ranking reflects the magnitude of the thermal forcing and its anomaly in the various WAIS basins (Figs. 9 and 10). For a given ESM, the most sensitive melt scheme (nonlocal-slope-PIGL) yields 3 to 4 times as much SLR as the least sensitive schemes (local- and nonlocal-MeanAnt), and the warmest models (HadGEM2 and UKESM) drive 2 to 3 times as much SLR as the coolest (MIROC). Thus, the total SLR over 550 years varies by an order of magnitude, from $\sim$150 mm to 1300 mm.

Figure 12 shows spatial plots of thickness changes from projection runs with forcing from three ESMs (HadGEM2, NorESM1, and CESM2) with nonlocal-MeanAnt and nonlocal-slope-PIGL, the least and most sensitive melt combinations. Ice loss for HadGEM2 is similar to that for a uniform TF anomaly of $1°$C (Fig. 7), and NorESM1 and CESM2 have somewhat smaller losses. As in the experiments with a uniform TF anomaly, most of the SLR contribution comes from the Filchner-Ronne and Ross sectors, with only a small Amundsen contribution. Compared to the WAIS, the SLR contribution from East Antarctica is

small, apart from the Amery sector in the nonlocal-slope-PIGL runs. As seen in Figs. 7 and 8, the nonlocal-slope scheme drives substantial grounding-line retreat without calving-front retreat, while the nonlocal scheme can trigger Ross Ice Shelf collapse, with less grounding-line retreat.

It is possible that the melt parameterizations underestimate the delivery of heat to Amundsen grounding lines, in part because of the negative $\delta T_{\text{sector}}$ corrections (Sect. 3.2) and the modest TF anomalies in the ESMs (Fig. 10). Conversely, the extrapo-

lation procedure and melt parameterizations might overestimate heat delivery to the Filchner-Ronne and Ross cavities, which currently have little basal melting. Although some ocean models with CMIP3 atmospheric forcing have projected Weddell Sea warming (e.g., Hellmer et al., 2012; Timmermann and Hellmer, 2013), it is not clear that these large shelves will readily transition from cold, low-melt regimes to warm, high-melt regimes (Naughten et al., 2018).

## 4.2   Sensitivity experiments

In this section we explore the sensitivity of projected ice loss to changes in model settings and forcing. We vary the grid resolution between 2 and 8 km; we replace the DIVA solver with the Blatter-Pattyn solver; we prescribe the collapse of selected ice shelves; and we replace the basal friction power law with the Schoof friction law, Eq. (2). We also study the sensitivity of the Amundsen sector under different parameter choices.

### 4.2.1 Grid resolution

Previous studies have found grounding-line retreat to be sensitive to grid resolution, with finer resolution typically leading to greater Antarctic retreat (Cornford et al., 2013, 2015). In benchmark experiments (Pattyn et al., 2013; Asay-Davis et al., 2016) for marine ice sheets, Leguy et al. (2020) found that with GLPs for basal friction and sub-shelf melting, CISM results at resolutions of 2 to 4 km are close to the converged results at 0.5 to 1.0 km. To test CISM's dependence on resolution for Antarctic projections, we ran a subset of experiments on 2-km and 8-km grids, with the same forcing and physics as in the 4-km runs. At each resolution we ran four spin-ups (using the nonlocal and nonlocal-slope parameterizations and the MeanAnt and PIGL calibrations), followed by six ESM-forced projections and a control run for each spin-up. Since CISM is expensive to run at continental scale on a 2-km grid, the 2-km spin-ups were run for 10 ky (instead of 20 ky for the standard experiments), resulting in slightly more drift during the projection runs. The drift in ice mass at the end of the 2-km spin-ups is 1 to 5 Gt/y (toward greater mass and lower sea level), compared to drift of < 1 Gt/y at the end of the 20-ky standard spin-ups.

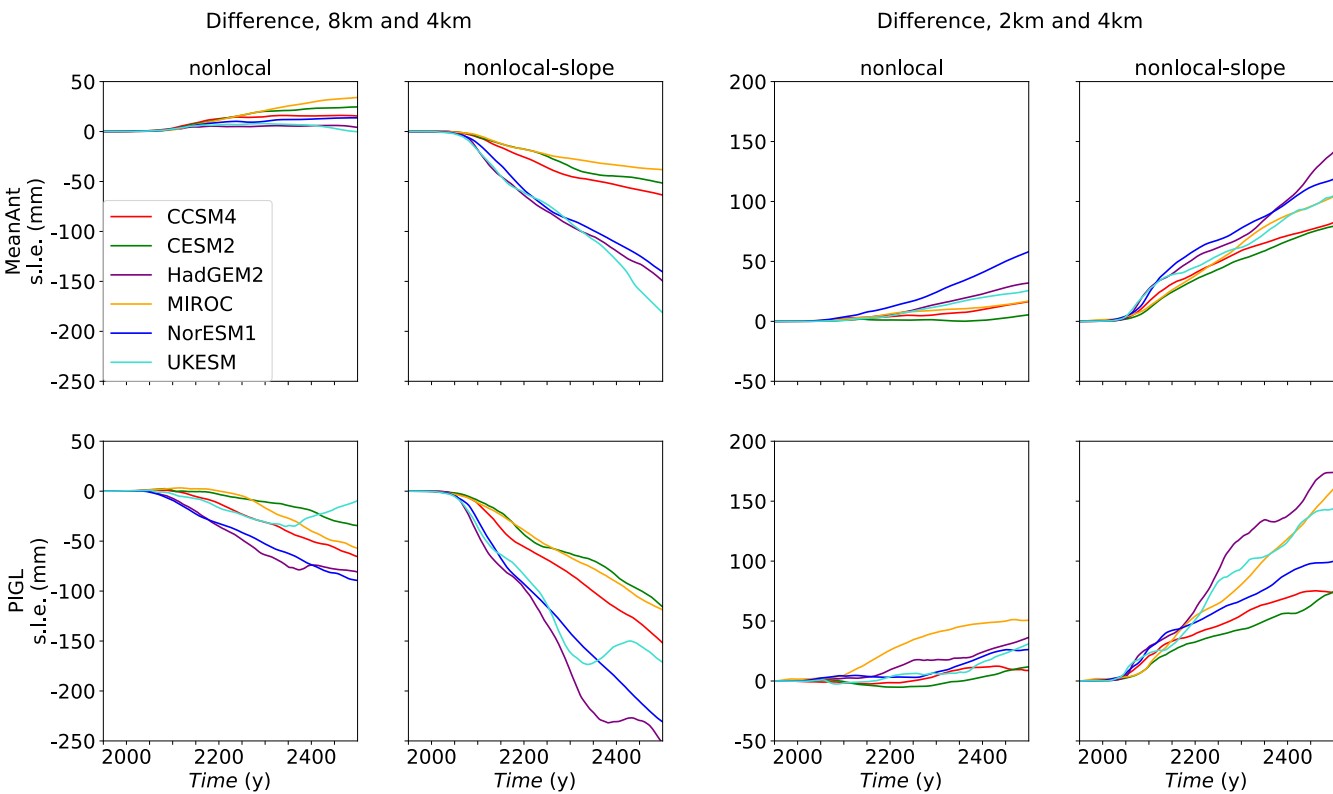

**Figure 13.** Difference in SLR contribution (mm s.l.e.) during 550-year projection experiments, as a function of grid resolution. The four left-hand panels show differences between 8-km and 4-km runs, and the four right-hand panels show differences between 2-km and 4-km runs. Positive values indicate a greater SLR contribution on the non-standard grid (8 km or 2 km). The four panels at each resolution correspond to the nonlocal and nonlocal-slope melt parameterizations with the MeanAnt and PIGL calibrations.

The spun-up ice states at 2 km and 8 km (not shown) are similar to the 4-km spin-ups (Sect. 3.2). For the projection experiments, the 8-km runs lose less mass than the 4-km runs (except for nonlocal-MeanAnt, which has slightly greater ice loss), while the 2-km runs lose more mass than the 4-km runs. Figure 13 shows time series of the difference in cumulative SLR, relative to the respective control runs, between the 8-km and 4-km runs (in the left-hand panels), and between the 2-km and 4-km runs (in the right-hand panels). For reference, Fig. 11 shows the 4-km time series. Generally, the SLR differences are largest for the runs with the most total ice loss. The 2-km SLR exceeds the 4-km SLR by up to ~15%, and the 4-km SLR exceeds the 8-km SLR by as much as 20%. The largest differences, and the greatest overall ice loss, are for the nonlocal-slope-PIGL runs forced by HadGEM2, for which the SLR contributions at 8, 4, and 2 km are 1047 mm, 1300 mm, and 1473 mm, respectively. Fig. 14 shows the difference between the thinning in the 2-km run and the 4-km run, where thinning is computed as the difference between the final state in 2500 and the initial state in 1950. At finer resolution, thinning is greater in the Ross and western Ronne sectors of the WAIS and in the Aurora sector (including Moscow University and Totten Glaciers) of East Antarctica. The 4-km run, however, has more thinning in the central Ronne sector. Thinning in the 4-km versus 8-km runs (not shown) has a similar pattern, with the largest differences in the Ross, Ronne, and Aurora sectors.

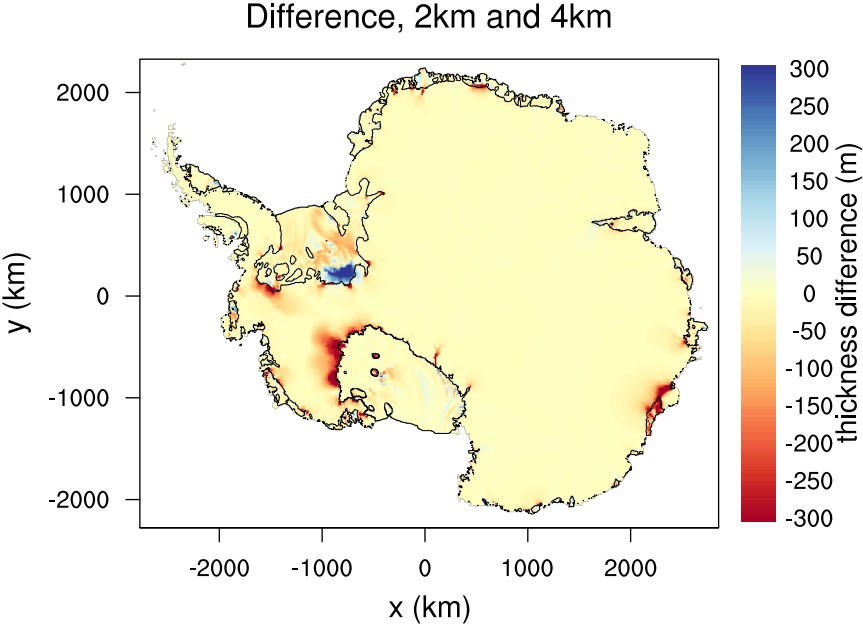

**Figure 14.** Difference in the ice thickness change (m) from two 550-year projection experiments, one on a 2-km grid and the other on a 4-km grid. The 2-km results were interpolated to the 4-km grid. Both runs use the nonlocal-slope-PIGL melt scheme with ocean forcing from HadGEM2. The ice thins in both runs. Negative values indicate greater thinning in the 2-km run than the 4-km run. Black lines show boundaries of floating ice at the end of the 4-km run.

Thus, refining the grid to 2 km leads to greater ice loss, but not dramatically so. The differences due to changing the grid resolution are smaller than those associated with varying the melt scheme for a given ESM, or varying the ESM for a given melt scheme (Fig. 11). We are not yet able to estimate the additional ice loss that would occur with refinement to 1 km or less.

### 4.2.2 Stress-balance approximation

5 Next, we compare standard experiments with the DIVA solver to experiments using the Blatter-Pattyn solver, which retains more terms in the Stokes equations and is formally more accurate (Schoof and Hindmarsh, 2010). Since CISM's BP solver is much slower than DIVA, it was not feasible to run long BP spin-ups at 4-km resolution, and therefore we compare BP to DIVA in 8-km runs. The model was spun up for 10 ky with four melt combinations (nonlocal and nonlocal-slope, each paired with MeanAnt and PIGL), and then run forward to 2500 with ESM ocean forcing.

10 Figure 15 shows time series of the difference in SLR contribution between BP and DIVA for each melt scheme and ESM. At year 2500 the differences are 30 to 50 mm in magnitude for the nonlocal-PIGL runs, and less than 20 mm for the other three melt schemes. The SLR contribution is greater for DIVA for all melt schemes except nonlocal-slope-MeanAnt. Figure 16 shows the difference in thinning between BP and DIVA for the runs with the largest difference: nonlocal-PIGL with UKESM forcing. The Ross Ice Shelf collapses in both runs, because of substantial melt rates distributed across the entire shelf. The 15 DIVA run loses more ice overall, but regional differences can be of either sign. For example, the Filchner Ice Shelf and Pine Island Glacier thin more with BP, but the central Ronne Ice Shelf and Thwaites Glacier thin more with DIVA. The differences are small compared to the total thinning.

Thus, the differences in SLR contributions between DIVA and BP runs are modest. These results are consistent with Lipscomb et al. (2019) and Leguy et al. (2020), who found close agreement between BP and DIVA for Greenland Ice Sheet 20 simulations and idealized marine-ice sheet simulations, respectively.

### 4.2.3 Ice-shelf collapse

The standard projections use a no-advance calving scheme that holds the calving front at its present-day location unless the shelf melts entirely from above or below. In reality, surface melting and hydrofracture from atmospheric warming could weaken shelves and in some cases trigger shelf collapse, which would enhance grounding-line retreat. For this reason, ISMIP6 provided 25 time-dependent masks for ice-shelf collapse (Nowicki et al., 2020), using the method of Trusel et al. (2015) to estimate annual surface melt from near-surface temperatures in CMIP models. The extent of hydrofracture and subsequent shelf collapse varies among ESMs, but the collapse accelerates after 2050 and is concentrated around the Antarctic Peninsula and the adjacent Bellingshausen sector (see Fig. 11 of Nowicki et al. (2020)). For this study, we repeated the 550-year projection runs with shelf collapse prescribed from the same ESMs providing ocean forcing. Masks are held fixed after 2100.

30 To complement the ISMIP6 collapse experiments, we also ran "expanded-collapse" experiments in which all small Antarctic shelves—not just those in the ISMIP6 masks—collapse abruptly in 2100, after which any ice flowing into these regions is calved immediately. There is no shelf collapse before 2100. Here, "small" denotes all shelves outside the Ross, Filchner-

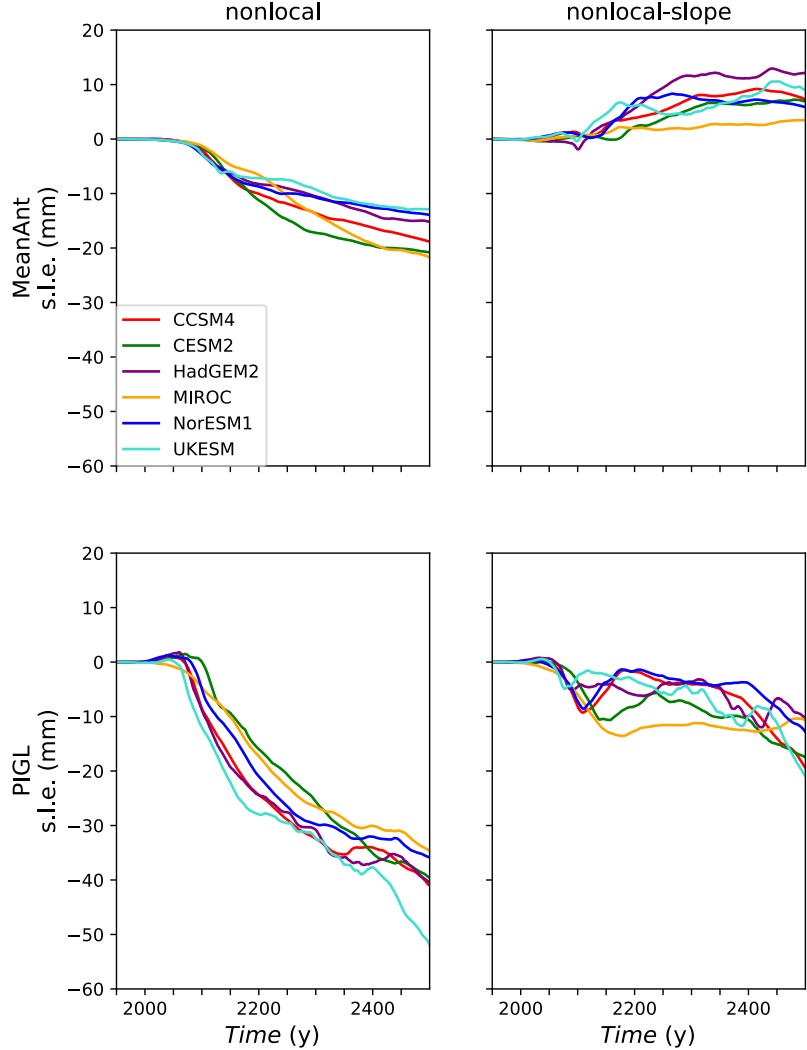

**Figure 15.** Difference in SLR contribution (mm s.l.e.) during 550-year projection experiments on an 8-km grid, comparing the Blatter-Pattyn (BP) and DIVA velocity solvers. Positive values indicate a greater SLR contribution in runs using the BP solver. The four panels correspond to the nonlocal and nonlocal-slope melt parameterizations with the MeanAnt and PIGL calibrations.

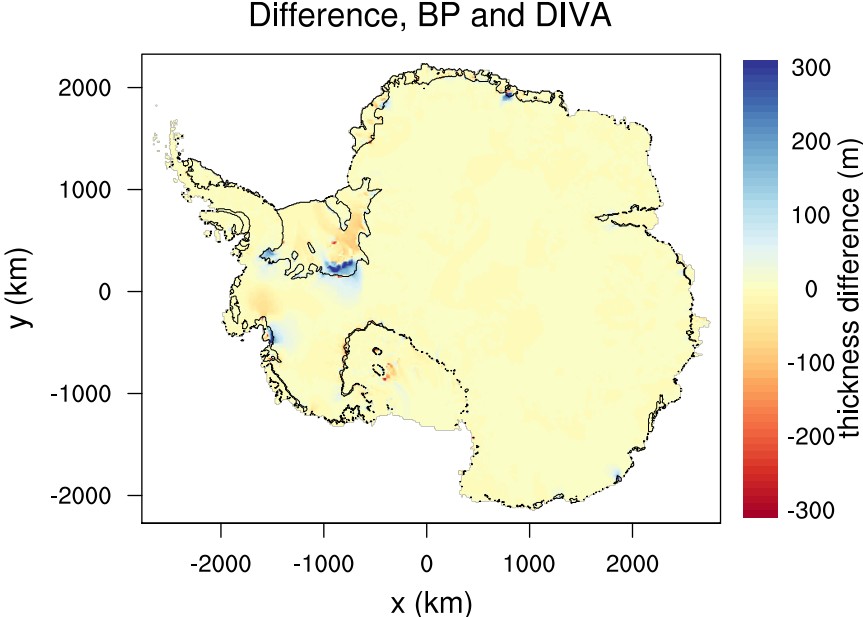

**Figure 16.** Difference in the ice thickness change (m) from two 550-year projection experiments on an 8-km grid, comparing the Blatter-Pattyn (BP) and DIVA velocity solvers. Both runs use the nonlocal-PIGL melt scheme with ocean forcing from UKESM. The ice thins in both runs. Negative values indicate more thinning in the BP run than the DIVA run. Black lines show boundaries of floating ice at the end of the DIVA run.

Ronne, and Amery sectors. We do not prescribe the collapse of the three largest shelves, because none of the six ESMs warms enough by 2100 to trigger collapse in these sectors.

The four left-hand panels of Fig. 17 show time series of the difference in cumulative SLR between the experiments with and without ISMIP6-based shelf collapse. The largest difference is about 200 mm, for the nonlocal-PIGL run with CESM2 forcing. Otherwise, the differences are mostly in a range of 50–150 mm, but with smaller values for NorESM1, the model with the least shelf collapse. For the ISMIP6 Antarctic projections, Seroussi et al. (2020) found that shelf collapse contributes a mean of 28 mm of SLR by 2100 compared to control experiments. Our runs have a comparable response in the $21^{st}$ century, but the majority of collapse-driven SLR takes place after 2100.

The four right-hand panels of Fig. 17 show the corresponding time series for the expanded-collapse experiments. The SLR differences at year 2500 range from ∼150 mm to 300 mm for the nonlocal-slope-PIGL runs, but are clustered near 200 mm for the other melt schemes. Figure 18 shows the change in ice thickness for the run with the greatest added SLR from shelf collapse: 296 mm, using the nonlocal-slope-PIGL scheme with CCSM4. The additional grounded ice loss is concentrated in the Aurora and Wilkes sectors of East Antarctica and in the Amundsen sector.

These differences, while significant, are small compared to the multi-meter SLR seen in the ABUMIP experiments (Sun et al., 2020) with the abrupt and sustained loss of all floating ice. In ABUMIP, the Ross, Filchner-Ronne, and Amundsen

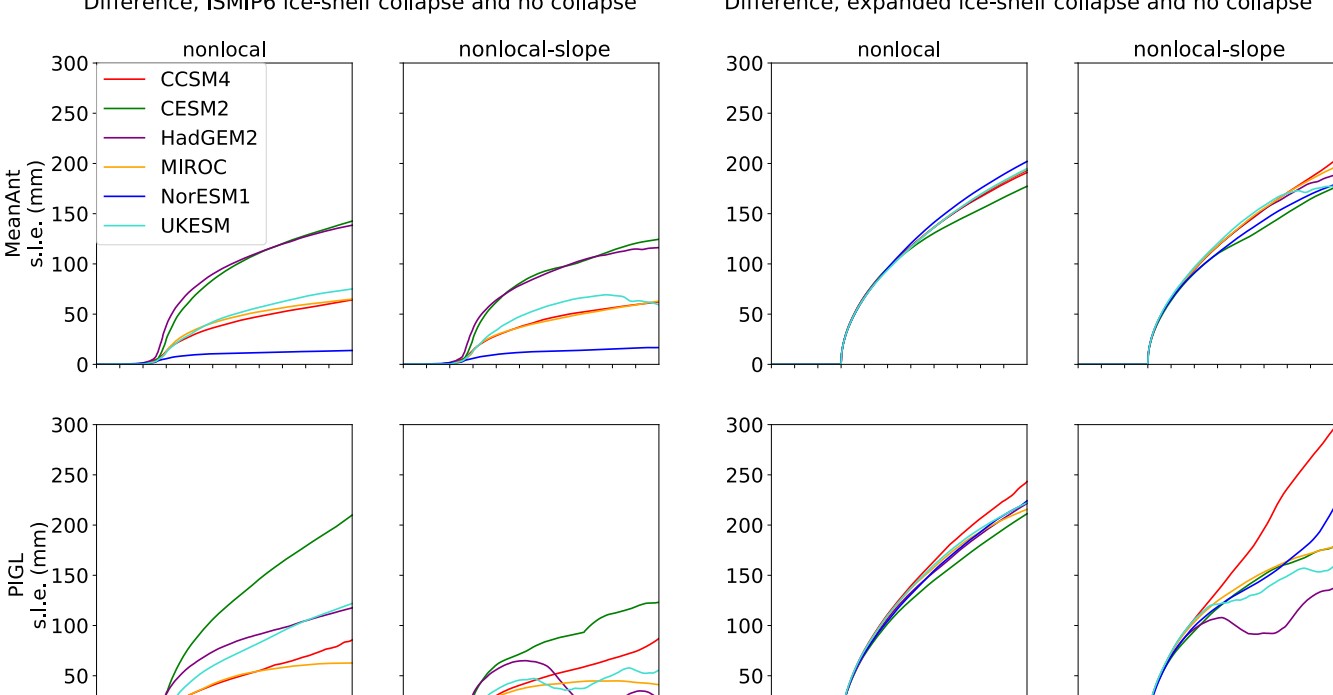

**Figure 17.** Difference in SLR contribution (mm s.l.e.) during 550-year projection experiments, comparing runs with and without ice-shelf collapse. The four left-hand panels show differences between runs with shelf collapse prescribed by ISMIP6 (Nowicki et al., 2020) and runs without collapse (standard experiments). The four right-hand panels show differences between runs with prescribed collapse of most ice shelves (excluding Ross, Filchner-Ronne, and Amery) in year 2100, and runs without collapse. Positive values indicate a greater SLR contribution in runs with collapse. The four panels for each collapse pattern correspond to the nonlocal and nonlocal-slope melt parameterizations with the MeanAnt and PIGL calibrations.

sectors each contribute ∼1 m or more of mean SLR when ice shelves are removed (see Tab. 3 of Sun et al. (2020)). Our less dramatic results confirm that the loss of small shelves would be less catastrophic than Ross or Filchner-Ronne collapse. The small Amundsen response in our runs suggests that while the sustained removal of all floating ice (as in ABUMIP) is sufficient to drive Amundsen collapse, the removal of several small shelves is not.

5    **4.2.4    Basal friction law**

The standard experiments in Sect. 4.1 assume that basal friction is described by a power law, Eq. (1), in which the effective pressure $N$ changes abruptly from a finite value to zero at the grounding line. Coulomb friction laws such as Eq. (3) imply a smooth transition to $N = 0$, with the bed providing little resistance to sliding near the grounding line. As a result, Coulomb friction renders the ice more sensitive to the loss of ice-shelf buttressing (Sun et al., 2020). To study the sensitivity of our

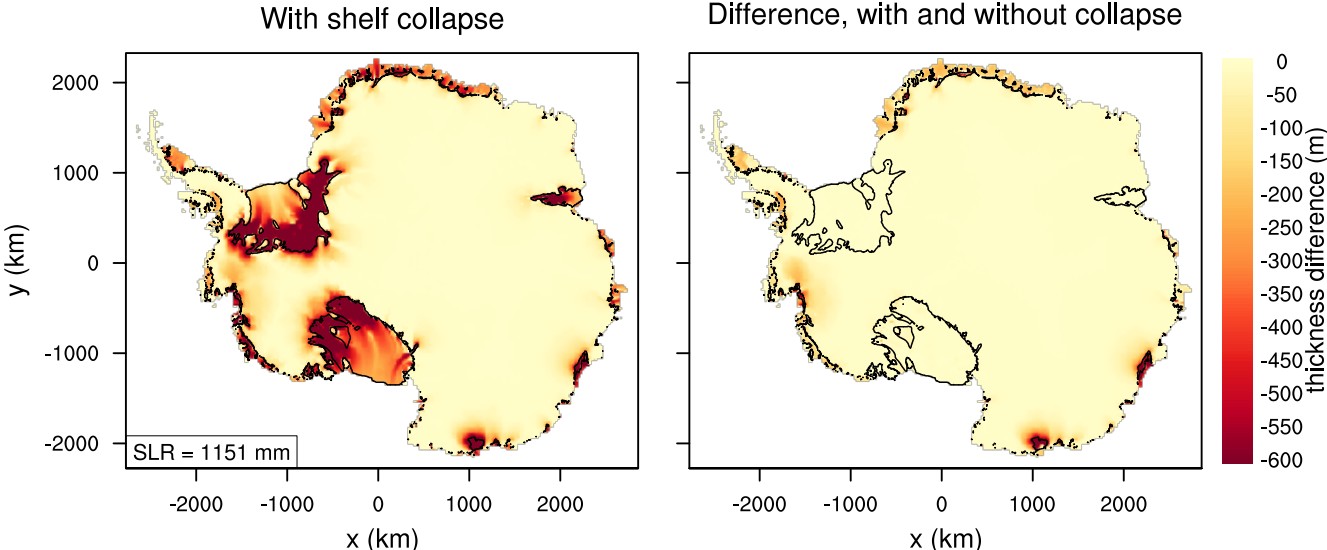

**Figure 18.** *Left:* Difference in ice thickness (m) between years 2500 and 1950 in an experiment with prescribed collapse of small ice shelves. The box in the lower left gives the SLR contribution in this run. *Right:* Difference in the ice thickness change (m) during 550-year experiments with and without shelf collapse. Both runs use the nonlocal-slope-PIGL melt scheme with CCSM4 ocean forcing. Negative values indicate greater thinning in the run with shelf collapse. Black lines show boundaries of floating ice at the end of the run with shelf collapse.

results to the basal sliding law, we ran additional spin-ups and projections using Eq. (2), which asymptotes to Eq. (3) near grounding lines. Effective pressure is given by Eq. (4) with $p = 0.5$ (i.e., partial support of the ice overburden by subglacial water pressure) or $p = 1$ (full support as $H$ approaches the flotation thickness at the grounding line). Although $p$ is not well constrained by data, recent work suggests that a Coulomb friction law with $p > 0$ gives better model–data agreement for Pine
Island Glacier than does a power law (Joughin et al., 2019).

For both $p$ values, CISM was spun up for 20 ky at 4-km resolution for each of four melt combinations: nonlocal and nonlocal-slope combined with MeanAnt and PIGL. With $p = 0.5$, the spun-up state is similar to that with $p = 0$. Reduced basal friction near the grounding line is compensated by higher values of the basal friction parameter, $C_p$, and by more negative thermal forcing corrections, $\delta T_{\text{sector}}$. With $p = 1$, initial spin-up attempts were unsuccessful, with irreversible collapse in
the Amundsen sector. To obtain a more realistic ice state, we modified the spin-up procedure in two ways: (1) The calving mask for Thwaites Ice Shelf was extended by several grid cells, allowing the shelf to ground on a ridge downstream of the present-day calving front, and (2) the model was spun up for 10 ky with $p = 0$ before switching to $p = 1$ for the remaining 10 ky. With these changes, the spun-up ice state with $p = 1$ is similar to the observed state. For all four spin-ups with $p = 0.5$, the maximum thermal forcing correction, $\delta T_{\text{sector}} = -2^\circ\text{C}$, is applied to the Aurora sector. For the spin-ups with $p = 1$, the
maximum correction is applied to both the Amundsen and Aurora sectors. These two sectors already have large negative corrections in standard spin-ups (Fig. 10).

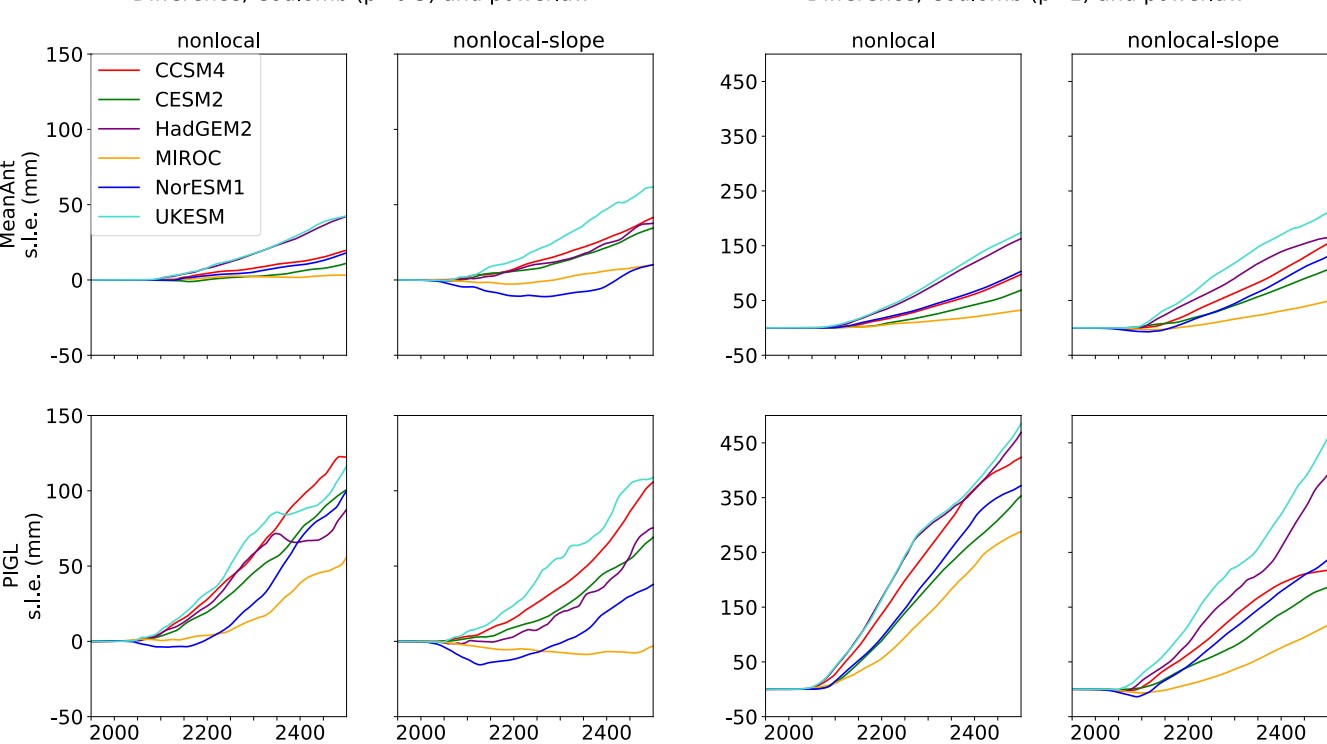

**Figure 19.** Difference in SLR contribution (mm s.l.e.) during 550-year projection experiments, comparing runs with power-law friction and Coulomb friction near the grounding line. The four left-hand panels show the difference between Coulomb friction with $p = 0.5$ and power-law friction, and the four right-hand panels show the difference between Coulomb friction with $p = 1$ and power-law friction. The four panels at each resolution correspond to the nonlocal and nonlocal-slope melt parameterizations with the MeanAnt and PIGL calibrations. Note the different vertical scales in the left and right panels.

Each spin-up was followed by 550-year projections for each of six ESMs. Figure 19 shows the differences in SLR contributions between Coulomb runs and power-law runs for $p = 0.5$ (left-hand panels) and $p = 1$ (right-hand panels). In nearly all cases, the Coulomb runs lose more ice than the power-law runs. With $p = 0.5$, the differences are modest, ∼60 mm or less for MeanAnt and ∼120 mm or less for PIGL. For the MeanAnt runs with $p = 1$, the differences are larger, approaching 200 mm for UKESM and HadGEM2 thermal forcing. The largest differences, up to ∼500 mm, are seen in the PIGL runs with $p = 1$, showing that the combination of weak basal friction and sensitive melt rates makes the ice much more vulnerable. Figure 20 shows the difference in the ice thickness change between Coulomb friction (with $p = 1$) and power-law friction for one of the most sensitive projections: nonlocal-slope-PIGL with UKESM forcing. The SLR contribution is 1302 mm for the standard run, increasing to 1760 mm for $p = 1$. Most of the additional ice loss is in the weakly grounded Ross sector.

**Figure 20.** Difference in the ice thickness change (m) from two 550-year projection experiments, one with Coulomb basal friction ($p = 1$) near the grounding line and one with power-law friction. Both runs use the nonlocal-slope-PIGL melt scheme with ocean forcing from UKESM. The ice thins in both runs. Negative values indicate greater thinning in the Coulomb run than the power-law run. Black lines show boundaries of floating ice at the end of the run with Coulomb friction.

### 4.2.5 Amundsen sector sensitivity

Finally, we explore the sensitivity of the Amundsen sector, including the Thwaites Glacier basin. Grounding lines in this sector have retreated rapidly in the past few decades (Rignot et al., 2019), and model simulations suggest that Thwaites Glacier collapse might already be under way (Joughin et al., 2014). Rignot et al. (2014) predicted that the Thwaites grounding line will
5  continue retreating for hundreds of km along its reverse-sloping bed, with only a few shallow ridges to slow the retreat. In most of our CISM simulations, however, Thwaites Glacier does not collapse. The grounding line starts to retreat but is stabilized by a large underwater ridge to the south and east of the present-day grounding line, between the Thwaites and Pine Island basins. Only with a large increase in melt rates (e.g., the PIGL runs with a TF anomaly of 2°C, shown in Fig. 8), does the grounding-line detach from this ridge, leading to collapse over several centuries.
10  The small Amundsen response in the standard projections can be attributed, in part, to the modest thermal forcing anomalies (∼1°C or less in the Amundsen sector) in the ESMs. Melt rate increases are further moderated by the negative feedback between basal melting and thinning; as the shelf thins, its base is exposed to reduced thermal forcing at shallower depths. Also, the use of a power law for basal friction can inhibit retreat. As a result, the grounding line stabilizes without much ice loss.

We recall that the Amundsen sector has a large, negative thermal forcing correction factor in many spin-ups, especially those with high melt rates near the grounding line (nonlocal-slope and/or PIGL; see Fig. 6) and weak basal friction. This correction factor, $\delta T_{\mathrm{sector}}$, is intended to correct for biases in sparse ocean observations and in the melt parameterization itself. It is likely, however, that thermal forcing in the Amundsen sector has increased in recent decades, due to some combination of natural variability and forced changes that have drawn warm CDW into cavities. Thus, the TF climatology used for spin-ups could be warmer than the conditions that prevailed when the ice sheet was evolving to its modern boundaries. In this case, the spin-up would generate $\delta T_{\mathrm{sector}} < 0$ to lower the Amundsen melt rates, and we would need a relatively large TF anomaly (perhaps $\sim 2^\circ$C) to raise melt rates to observed present-day values.

Motivated by the hypothesis that the recent warming which is driving Amundsen retreat is not reflected in ESMs, we ran another set of projection experiments. We start from the spin-ups described in Sect. 4.1 (for power-law friction) and Sect. 4.2.4 (for Coulomb friction), which yield optimized values of $\delta T_{\mathrm{sector}}$. Then, for projections, we set $\delta T_{\mathrm{sector}} = 0$ for the Amundsen sector only, on the assumption that the TF climatology in this region is accurate for the present day but warmer than conditions in the mid $20^{\mathrm{th}}$ century and earlier. Compared to other regions, the Amundsen sector has a large number of recent observations, including some observations in sub-shelf cavities.

We find that the Thwaites basin collapses when $\delta T_{\mathrm{Amundsen}} = 0$ is combined with Coulomb friction (with $p = 1$) and the PIGL calibration. With $p \leq 0.5$ and/or the MeanAnt calibration, there is no collapse. Figure 21 shows time series of the SLR contribution from seven projections for each spin-up with $p = 1$: six runs with $\delta T_{\mathrm{Amundsen}} = 0$ combined with ESM ocean forcing, and one run with $\delta T_{\mathrm{Amundsen}} = 0$ and no other changes in ocean forcing. For the PIGL runs, Amundsen collapse adds $\sim 1$ m of SLR, with or without TF anomaly forcing from the ESMs. Figure 22 shows the change in ice thickness between 1950 and 2500 for the four runs with $\delta T_{\mathrm{Amundsen}} = 0$ and no added ESM forcing (hence, no forcing changes in other sectors). The differences between MeanAnt and PIGL are striking.

For the nonlocal-PIGL run (corresponding to the black line in the lower left panel of Fig. 21), the collapse proceeds in several stages, as shown in Fig. 23. In 2100, the grounding lines of several Amundsen glaciers (Pine Island, Thwaites, Crosson, and Dotson) have joined and retreated modestly, with further retreat impeded by a ridge between the Pine Island and Thwaites basins. The ice is grounded on the ridge, which rises to a depth of about 100 m near location $x = -1540$ km, $y = -330$ km. By 2200, the grounding line has moved $\sim 100$ km upstream on either side of the ridge, but still has not retreated past the ridge. During the next century, the grounding line breaks free of the ridge and the ice shelf vastly expands, leading to an acceleration of ice loss and SLR. By 2400, the central part of the grounding line stabilizes against higher topography, several hundred km south of the present-day grounding line, while retreat continues to the east and west. For the nonlocal-slope-PIGL run, the retreat begins several decades later, but the retreat pattern is similar. In both MeanAnt runs, the grounding line stabilizes when it reaches the ridge.

Thus, the Amundsen sector exhibits threshold behavior in CISM. With the combination of a large thermal forcing anomaly (relative to the end of the spin-up), high sensitivity of melt rates to thermal forcing, and low basal friction near the grounding line, the grounding line retreats upstream of stabilizing topography, and the grounded ice in the basin collapses. Otherwise, Amundsen retreat is small, and Antarctic ice loss is dominated by thinning and retreat in the Ross and Filchner-Ronne sectors.

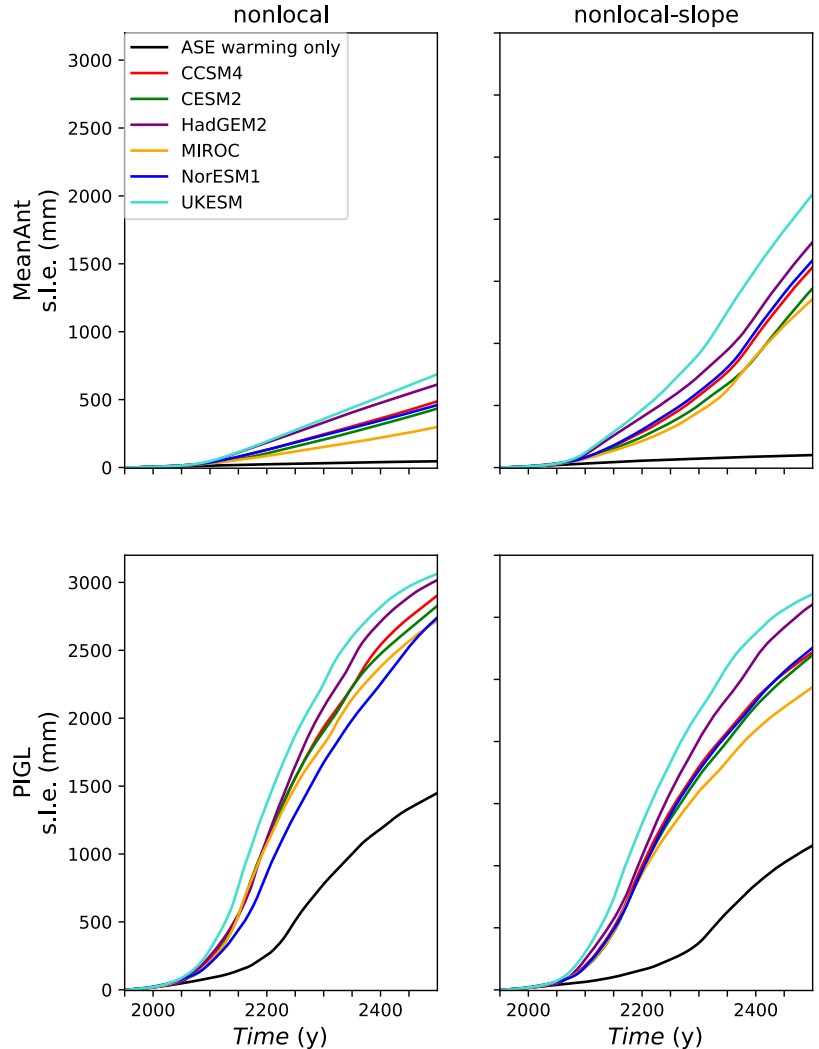

**Figure 21.** Sea-level rise (mm s.l.e.) from Antarctic ice loss in 550-year experiments with a Coulomb friction law ($p = 1$), and with $\delta T_{\mathrm{sector}} = 0$ for the Amundsen sector to represent a $2°$C anomaly relative to the spin-up. Black lines show SLR for experiments without ESM ocean forcing anomalies, so that the response is limited to the Amundsen sector. The other lines show SLR for experiments with ESM ocean forcing anomalies in addition to the $2°$C Amundsen anomaly. The four panels at each resolution correspond to the nonlocal and nonlocal-slope melt parameterizations with the MeanAnt and PIGL calibrations.

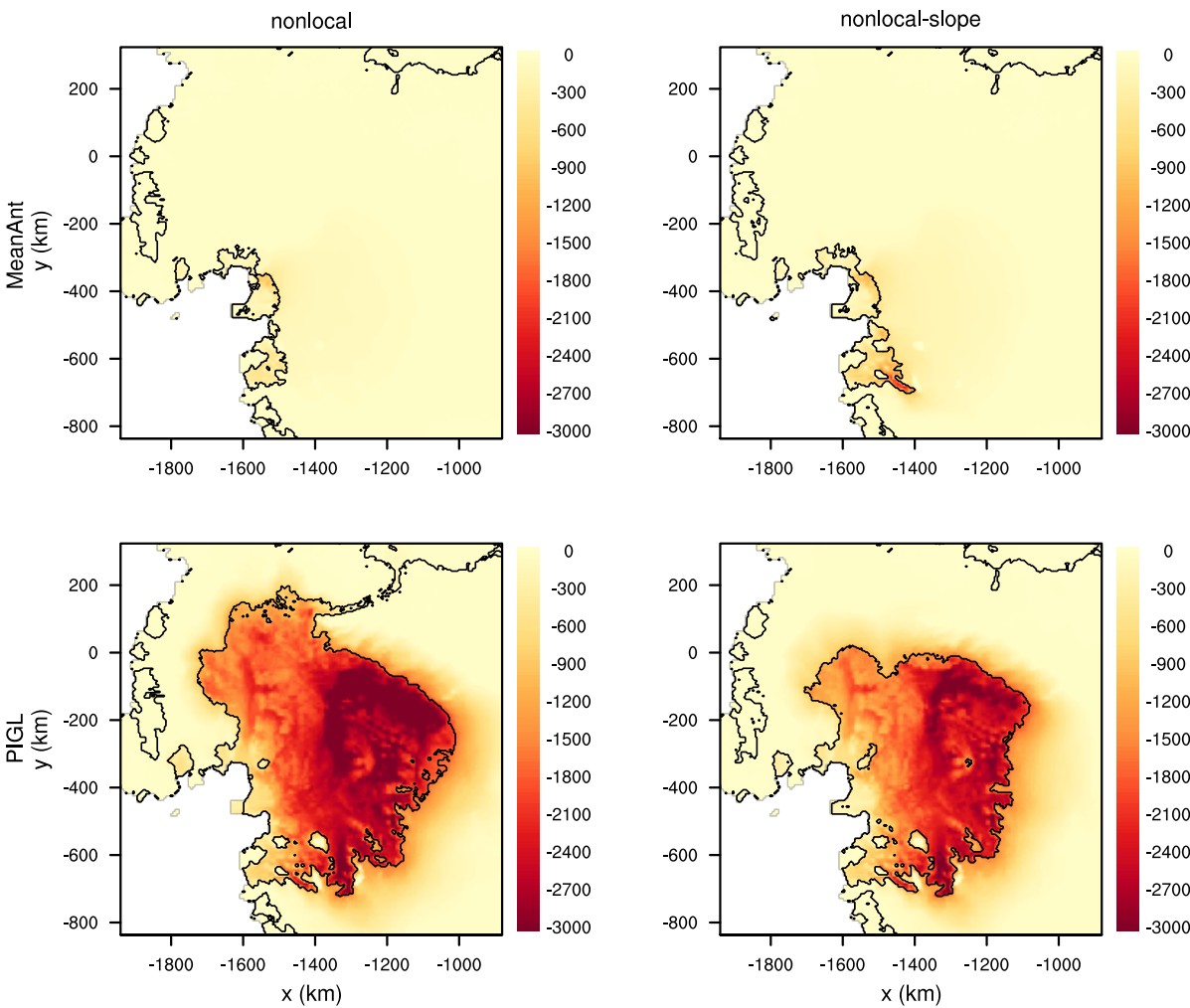

**Figure 22.** Difference in ice thickness (m) between years 2500 and 1950 in projection experiments with a Coulomb friction law ($p = 1$), and with $\delta T_{\text{sector}} = 0$ for the Amundsen sector to provide a $2°C$ anomaly relative to the spin-up. No ESM ocean forcing anomaly is applied, so the response is limited to the Amundsen sector. The four panels at each resolution correspond to the nonlocal and nonlocal-slope melt parameterizations with the MeanAnt and PIGL calibrations. Black lines show boundaries of floating ice at the end of each run.

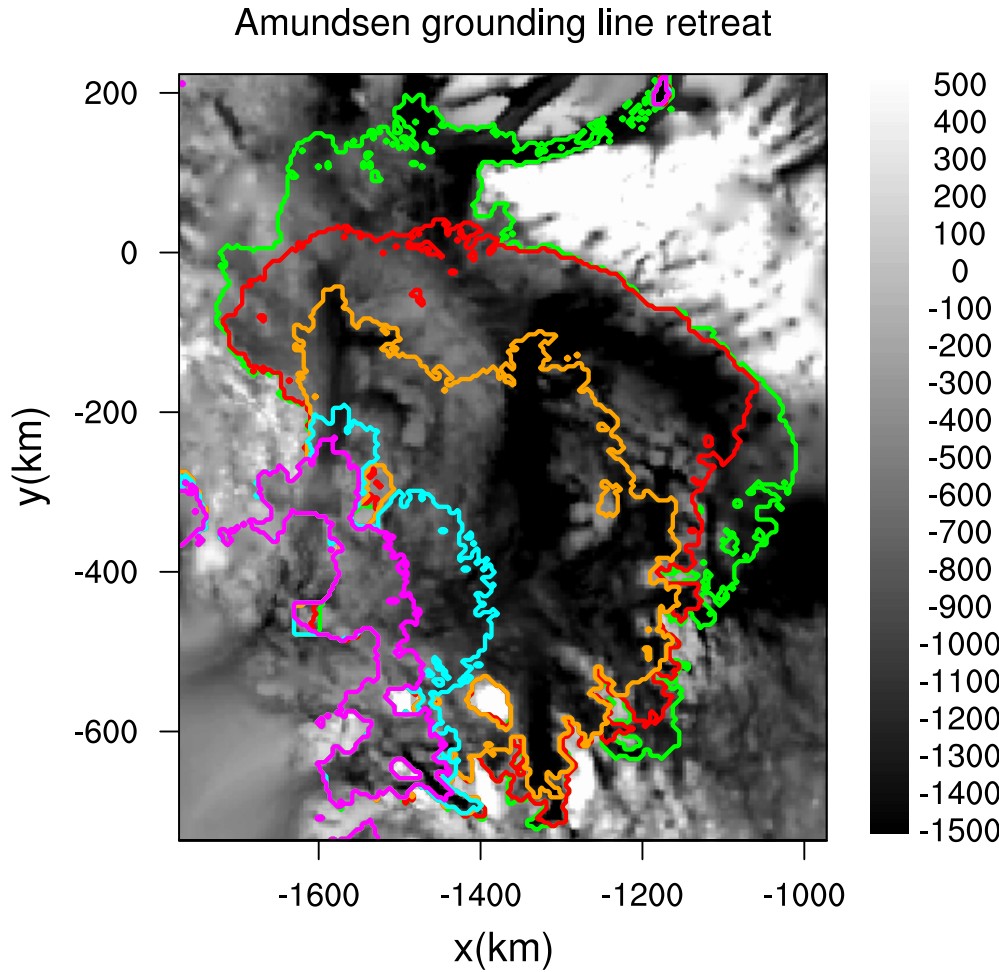

**Figure 23.** Grounding-line retreat in the Amundsen sector during a 550-year projection experiment using the nonlocal-PIGL melt scheme, corresponding to the black line in the lower left panel of Fig. 21. Seafloor topography (m) is shaded on a gray scale. The solid lines show boundaries of floating ice in years 2100 (mauve), 2200 (cyan), 2300 (orange), 2400 (red), and 2500 (green).

## 5   Conclusions

Using the Community Ice Sheet Model, we ran an ensemble of ice sheet simulations based on the protocols for the ISMIP6 Antarctic projections (Nowicki et al., 2020). For standard experiments, CISM was run on a 4-km grid using a depth-integrated higher-order solver (DIVA) with a power law for basal friction. We carried out six spin-ups to test combinations of three basal melt parameterizations (local, nonlocal, and nonlocal-slope) and two calibrations (MeanAnt and PIGL), as described by Jourdain et al. (2020). In each spin-up, the model was nudged toward present-day ice thickness by adjusting friction parameters in the basal sliding law and adjusting basin-scale thermal forcing corrections in the melt parameterizations. The resulting spun-up states are similar across melt schemes, with minimal drift. The ice thickness, velocity, and shelf extent are generally in good agreement with observations, but there are persistent errors in grounding line locations, such as a Thwaites grounding line that is too far retreated.

From the spun-up states, we ran each ensemble member forward for 550 years, applying ISMIP6 ocean thermal forcing for 1950–2100 and cycling repeatedly through the 2081–2100 forcing for the rest of the simulation. Using forcing from four CMIP5 models and two CMIP6 models, we ran 36 projection experiments. For each simulation we analyzed the ice mass loss and associated sea level rise. In all cases, the Antarctic Ice Sheet loses mass, with total grounded ice losses ranging from about 150 mm to 1300 mm s.l.e. Mass loss begins slowly, accelerates in the late $21^{st}$ century, and continues steadily for the next several centuries. The thinning of grounded ice (i.e., ice that contributes to SLR) is concentrated in the Filchner-Ronne and Ross basins of WAIS, which have reverse-sloping beds and are vulnerable to marine ice sheet instability. There is less ice loss in the Amundsen sector, where grounding lines typically stabilize after a modest retreat. East Antarctica loses relatively little grounded ice.

We ran several sets of sensitivity experiments to study the effects of changes in grid resolution, stress-balance approximation, ice-shelf extent, and basal friction law. Changes are small ($< 50$ mm s.l.e. during 550-year projections) when DIVA is replaced with a Blatter-Pattyn solver, and moderate (with additional SLR of up to $\sim$200 mm) when CISM is run on a 2-km grid or when small ice shelves are removed by the end of the $21^{st}$ century. Replacing a power law with a Coulomb friction law with $p = 1$ (i.e., low effective pressure near the grounding line) increases the total SLR by up to $\sim$500 mm. Thus, the sensitivity of SLR to the basal friction law can be comparable to the sensitivity to the basal melt parameterization and calibration.

We also explored the conditions required for major retreat in the Amundsen sector. The Amundsen basin exhibits threshold behavior, collapsing and adding $\sim$1 m of SLR when Coulomb friction and high basal melt-rate sensitivity are combined with a thermal forcing anomaly of $\sim$2°C relative to the spin-up. The source of this threshold behavior is a ridge to the south and east of the present-day Thwaites grounding line. Once the grounding line passes this ridge, further retreat is inexorable.

These results are consistent with recent studies (e.g., Cornford et al., 2015; Pollard and DeConto, 2016; Larour et al., 2019) showing potential WAIS collapse, driven by ocean warming in regions with reverse-sloping beds. The novelty of this study lies in the application of the ISMIP6 Antarctic forcing protocols (with melt rates derived from the thermal forcing data of Jourdain et al., 2020) to an ensemble of multi-century simulations, along with the tuning of sector-wide correction factors to optimize the agreement with observed ice thickness near grounding lines. The results suggest that, by the end of this century

(assuming high-end emissions scenarios), sub-ice-shelf melt rates could be large enough, if sustained, to drive the steady and possibly irreversible retreat of much of the WAIS. The Filchner-Ronne and Ross sectors are sensitive to modest increases ($\sim$1°C) in thermal forcing, while the Amundsen response varies from minor retreat to complete collapse, depending on the parameterization of basal friction and basal melting near grounding lines.

5    Ice loss is greater with the nonlocal-slope parameterization than with the local and nonlocal parameterizations, and greater with the PIGL calibration (which has higher values of the melt parameter $\gamma_0$, and thus is more sensitive to changes in thermal forcing) than with MeanAnt. Mass loss varies by a factor of about 4 between the most sensitive melt scheme (nonlocal-slope-PIGL) and the least sensitive (local- and nonlocal-MeanAnt), even when the ocean forcing comes from the same ESM. All the melt schemes leave out important physics, and it is not known whether one parameterization or calibration is more accurate than the others. Similarly, ice loss varies by a factor of 2 to 3 between the warmest ESMs (HadGEM2 and UKESM) and the coolest (MIROC), even when using the same melt scheme. While suggesting that much of the WAIS is vulnerable to projected ocean warming, these results do not place firm bounds on the rate or magnitude of future SLR.

Ice sheet retreat in these simulations is modest for the next several decades. It is not clear to what extent this inertia is real, rather than an artifact of the spin-up procedure. The model starts close to steady state, nominally in 1950, and is run through the recent historical period without assimilating retreat that is already under way (e.g., Rignot et al., 2019). Several ISMs in the ISMIP6 Antarctic ensemble (Seroussi et al., 2020) use similar spin-up methods that could underestimate near-term ice loss, while others use data assimilation techniques that are better able to capture ongoing trends. Given the inertia in some models, including CISM, Antarctic projections ending in 2100 (largely for practical reasons, since CMIP forcing often is unavailable after 2100) do not give a full picture of sea-level commitment from ice sheet retreat. It is possible that ocean warming during this century could result in a commitment of several meters of SLR from the WAIS, but with most of the SLR taking place in future centuries.

To these conclusions, we add several caveats. First, the inversion procedure gives large negative temperature corrections ($\sim$1°C or greater in magnitude) for the Amundsen sector. If the lower thermal forcing in the spin-up is representative of preindustrial conditions, and the higher thermal forcing in observations reflects recent warming that is not captured by ESMs, then the ESM-derived thermal forcing anomalies could be too low. Amundsen sensitivity would then be underestimated. If Amundsen glaciers are best described by a Coulomb basal friction law, then the use of power-law friction would also understate the sensitivity.

Second, the ocean data extrapolation transfers ESM heat anomalies from the open ocean to the distant grounding lines of the Filchner-Ronne and Ross shelves, where present-day melting is minimal. If the simple melt schemes deliver too much heat to grounding lines in what are now cold-water cavities, the Filchner-Ronne and Ross retreat would be overestimated.

More generally, these simulations leave out many physical processes and feedbacks. Changes in atmospheric forcing are omitted by design, to simplify the analysis; increased snowfall in the ISMIP6 multi-model ensemble can mitigate the mass loss from ocean forcing (Seroussi et al., 2020). Although we ran some simulations with ice-shelf removal, we did not consider fast feedback processes such as cliff collapse (Pollard and DeConto, 2016). Solid-Earth and sea-level feedbacks are missing; these have been found to delay (but not prevent) long-term ice retreat in the Thwaites basin (Larour et al., 2019). At 4-km grid

resolution, processes near the grounding line are under-resolved. Moreover, the depth-based melt parameterizations ignore important processes such as eddy heat transfer onto the continental shelf (Stewart and Thompson, 2015) and topographic steering. Without these processes, thermal forcing in cavities could be missing critical spatial structure. Finally, there is no interaction between the ice sheet and the cavity circulation or open ocean, so that freshwater fluxes from increased melting are
unable to modify the thermal forcing.

These uncertainties suggest several lines of research to improve ice-sheet and sea-level projections:

– Running similar long-term simulations with multiple ice sheet models, to quantify structural ISM uncertainties.

– Extending more CMIP ESM simulations beyond 2100, to generate forcing for multi-century ice sheet simulations.

– Forcing ISMs with a greater variety of scenarios, including overshoot scenarios to study whether WAIS retreat, once
begun, could be stopped or reversed.

– Adding more realistic physical processes and feedbacks, such as hydrofracture, subglacial hydrology, and solid-Earth/sea-level effects. Some ISMs already simulate these processes more realistically than was done in this study.

– Further refining the accuracy of high-resolution data sets of Antarctic bed topography (e.g., Morlighem et al. 2019).

– Continuing to develop comprehensive data sets of sub-ice-shelf melt rates and Southern Ocean temperature and salinity,
to better force models and validate parameterizations.

– Running regional, high-resolution simulations of the coupled ice shelf–ocean system. Such simulations are planned for the Weddell Sea and Amundsen Sea sectors during the second phase of the Marine Ice Sheet–Ocean Model Intercomparison Project (MISOMIP2).

– Developing simple models of ice-shelf cavities and sub-shelf melting, which can emulate the results of high-resolution
coupled models but are efficient enough to run for long timescales in global models.

– Incorporating interactive ice sheet–ocean coupling in the next generation of ESMs.

Many of these efforts are under way and could contribute to future intercomparison projects, including the anticipated ISMIP7.

*Code availability.* CISM is an open-source code developed on the Earth System Community Model Portal (ESCOMP) git repository at *https://github.com/ESCOMP/CISM*. The version used for these runs is tagged as *ISMIP6_Antarctica_TC2020*.

*Data availability.* The model configuration and output files for the spin-ups described in Sect. 3 and the projection experiments in Sect. 4 will be archived on the NCAR Climate Data Gateway at *https://www.earthsystemgrid.org/*.

*Author contributions.* WHL conceived the study, designed the spin-up procedure, and developed CISM as needed to run the ISMIP6 experiments. WHL and GRL staged, ran, and analyzed the simulations. NCJ and XAD created the thermal forcing data sets and calibrated the melt parameterizations. HS coordinated Antarctic projections for ISMIP6, and SN led the overall ISMIP6 project. WHL and GRL wrote the paper with contributions from all authors.

*Competing interests.* William Lipscomb, Sophie Nowicki, and Hélène Seroussi are editors of the ISMIP6 special issue of *The Cryosphere*. The authors declare no other competing interests.

*Acknowledgements.* We thank the Climate and Cryosphere (CliC) project, which provided support for ISMIP6 through sponsoring of workshops, hosting the ISMIP6 website and wiki, and promoting ISMIP6. We acknowledge the World Climate Research Programme, which, through its Working Group on Coupled Modelling, coordinated and promoted CMIP5 and CMIP6. We thank the climate modeling groups
for producing and making available their model output, the Earth System Grid Federation (ESGF) for archiving the CMIP data and providing access, the University at Buffalo for ISMIP6 data distribution and upload, and the multiple funding agencies who support CMIP5, CMIP6, and ESGF. We thank the ISMIP6 steering committee, the ISMIP6 model selection group, and the ISMIP6 dataset preparation group for their continuous engagement in defining ISMIP6. We thank Mathieu Morlighem for early access to the BedMachineAntarctica data set. We are grateful to Bill Sacks and Kate Thayer-Calder for software engineering support, and to the entire ISMIP6 community for stimulating
discussions. We also thank Thomas Zwinger and one anonymous reviewer for their detailed and thoughtful comments.

This material is based upon work supported by the National Center for Atmospheric Research, which is a major facility sponsored by the National Science Foundation under Cooperative Agreement No. 1852977. Computing and data storage resources, including the Cheyenne supercomputer (doi:10.5065/D6RX99HX), were provided by the Computational and Information Systems Laboratory (CISL) at NCAR. NJ is funded by the French National Research Agency (ANR) through the TROIS-AS project (ANR-15-CE01-0005-01) and by the
European Commission through the TiPACCs project (grant 820575, call H2020-LC-CLA-2018-2). Support for XAD was provided through the Scientific Discovery through Advanced Computing (SciDAC) program funded by the US Department of Energy (DOE), Office of Science, Advanced Scientific Computing Research and Biological and Environmental Research Programs. HS and SN were supported by grants from the NASA Cryospheric Science and Modeling, Analysis and Prediction Programs, and the NASA Sea Level Change Team. For analysis and figure preparation, we used the NCAR Command Language (Version 6.6.2) [Software], 2019, Boulder, Colorado: UCAR/NCAR/CISL/TDD,
http://dx.doi.org/10.5065/D6WD3XH5.

This is ISMIP6 contribution no. X.

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
