# Peer review of "ISMIP6-based projections of ocean-forced Antarctic Ice Sheet evolution using the Community Ice Sheet Model"

_The Cryosphere, 2019_

## Referee Comment (RC1) · Thomas Zwinger (Referee) · 25 Feb 2020

**1   General Impression**

This is a manuscript that complements the currently reviewed ISMIP 6 inter-comparison on the Antarctic ice sheet. Beyond the contribution that was already accounted for in the paper by Seroussi et al. (Seroussi et al., 2020), the authors show new ways of spinup and present computations beyond the time-frame of ISMIP 6.

The text describes in detail the methods used and the assumptions applied within the contribution from the ice-sheet model CISM to this inter-comparison. From my

point of view, the paper to a large extent analyses the impacts of assumptions and approximations on the results, except for two open points which I will raise in the section below. The article is well written and has a clear line to follow. I got the impression of a few inaccuracies in the notation, which in my view is easily to be fixed. I also have some suggestions on how to improve or augment the figures.

Modelling techniques presented here are mainly variations of existing approaches, but the size and the context of the applications make for the scientific novelty of this paper. I also see the value in documenting the techniques leading to these results, which of course cannot be presented in such detail in an inter-comparison paper. In that sense, I see it as a useful contribution to the literature, also to demonstrate what efforts have to be taken in order to realise such an implementation.

I mainly placed a few technical suggestions to (according to my subjective impression) improve the readability of the text, in particular for readers not being that familiar with ISMIP 6. Besides this, the only more substantial criticism I would place is the in my view not complete analysis of the impact on the results of some of the assumptions made, in particular the over-simplification of calving and the impact of the (actually not clearly revealed) approximation to the Stokes equation and the resolution applied in the ice-sheet model. If these points are addressed, I would recommend this manuscript for publication in TC.

**2   Main points**

As mentioned above, I would see it necessary to have two in my view not completely clear issues to be elaborated:

1. The influence of the in my view oversimplified calving law on the results, in particular its impact on grounding line migration and retreat

2. The choice of the approximation to the Stokes equation and its consequences on the accuracy and sensitivity of the results

Concerning the first point, i.e. calving, you explain (page 4, line 20): *Instead, we use a no-advance calving mask, removing all ice that flows beyond the observed calving front. The calving front can retreat where there is more surface and basal melting than advective inflow, but more often the calving front remains in place.* Here you might mention the point in time of the observation of the prescribed front (I presume it somehow refers to your initial state in 1995). In the Conclusions, you mention it in just one sentence: *In terms of ice sheet physics, these simulations do not include hydrofracture or calving-front retreat, and thus are missing positive feedbacks associated with reduced buttressing of grounded ice by ice shelves (Sun et al., in review).* In my opinion, the reader would benefit from some clarification on how much impact neglecting the possibility of ice-shelf collapse (as it would be provided by ISMIP 6 forcing) on the resulting sea level has. In my view, the current approach significantly might lead to an underestimation of sea-level rise, in particular at times beyond the year 2050. One additional experiment could be to, for instance, just remove the shelf in front of Thwaites and see what short-time impact this has for this region – maybe in combination with the upper end combination of AOGM and ocean forcing. Or to apply the shelf-collapse scenario from ISMIP 6. If this is for some technical reason not possible at this stage, please mention that in the paper.

On the second point, model approximation, you present (Page 3, line 30) that the CISM code includes options for the following approximations to the Stokes equations: *SSA, L1L2, LMLa* – mainly following Hindmarsh' nomenclature (Hindmarsh, 2004). On page 4, line 1, you write: *The simulations for this paper use the depth-integrated solver, which gives a good balance between accuracy and efficiency for continental-scale simulations (Lipscomb et al., 2019).* I consider SSA as well as L1L2 as depth integrated (the latter of course with some corrections from a vertical profile). Could you please clearly identify which of the previously listed models you deploy in your study? I am

further confused how a depth integrated model can use 5 vertical layers (as described in the paragraph before), but assume that you use this for some aspects different from depth-integrated parts, such as temperature evolution or velocity corrections. Is this vertical resolution sufficient to resolve the physics? Can you include some lines on how much you estimate the influence of the choice of the approximation and the chosen resolutions on the results of your study. For instance, would one expect to reach the same conclusions if using a LMLa model or perhaps the current model with a finer resolution?

**3  Suggested corrections, additions and typos**

Please, find suggestions for corrections, changes and typos in the order of their occurrence in the text:

- Page 1, Author list: Typo: Nicolos → Nicolas

- Page 2, line 28: *To study long-term ice sheet evolution, we extend the simulations to 2500, ...* Minor issue, but for better readability, please mention that this is a date (it could be iterations or any other time unit).

- Page 4, line 6, 9 and 11: Again, minor issue, but in order to stay consistent with the nomenclature in equation (1), I would use the subscript for bedrock in all the following occurrences of the basal velocity, i.e., $u \rightarrow u_b$.

- Page 4, line 13: Can you elaborate on the choice of exactly this value for $C_c$? Is this representing some tested optimum? Or (see next question) does it not really matter what to put there?

- Page 4, line 17: *Setting $N$ to overburden pressure implies power-law behavior in nearly all of the ice sheet.* To my understanding, this means that power-law

sliding then most likely prevails down to the grounding line - or am I missing something? If this is the case, I would question the benefit of explaining sliding law (1) and the Coulomb sliding parameter therein in such a detail. Being aware that in lack of a detailed representation of pinning points, any sliding law that reaches zero resistance at the grounding line inherently causes issues (confirmed also by our experience with Full-Stokes models at similar resolution), I do not criticise the approach as such, but would welcome to include some information on how high the friction parameters at the grounding line actually are (or have to be in order to be stable) and how – also by keeping them constant relative to a moving grounding line – that impacts the prognostic runs.

- Page 5, line 5: *(The simulations described here use only the ocean forcing.)* I do not understand the meaning of this sentence. Does it mean, you do not apply any atmospheric, but only ocean forcing? Or does this only apply in connection to CMIP? Please, elaborate. On a side-note, in my personal opinion, I find it strange to put whole separate sentences in brackets. I would write it without. I mention this, as it occurs a few times in the text.

- Page 8, line 4: *Eq. 6 is based on the equation for a critically damped harmonic oscillator, where the first term in brackets nudges H toward Hobs, and the second term damps the nudging to prevent overshoots. (The damping is not exactly critical, however, because $dC_p/dt$ is not exactly proportional to $d^2H/dt^2$.)* I am slightly confused by this statement, since from my point of view, if $dC_p/dt$ would be somehow proportional to $d^2H/dt^2$, the $C_p$ on the right-hand-side would introduce some functional relation to $H$ and I would not immediately see the equation of a damped harmonic oscillator recovered by (6). Could you clarify or insert a reference where this is explained in detail in order to help the reader to follow that up. Minor issue: *Eq. 6 →* Eq. (6).

- Page 8, line 6: *We hold $C_p$ within a range between $10^2$ and $10^5$ Pa m$^{-1/3}$ y$^{-1}$, since smaller values can lead to numerical instability, and larger values do not significantly lower the sliding speed.* Can you explain the nature of the instabilities? I wonder if this is somehow linked to the fact that you had to fix the effective to the overburden pressure in the sliding law.

- Page 9, line 7 and Eq. (8): Based on results shown in Fig. 5, I conclude that the symbol $\delta T$ stands for the nudged values of the by the ISMIP 6 protocol given $\delta T_{sector}$. Simply, the missing subscript might also lead to the conclusion that it is the difference to this reference value, i.e., the correction to the correction. I would consistently add the subscript *sector* to $\delta T$.

- Page 9, line 5: *After some experimentation, we set $m_T = 10$ m y$^{-1}$ K$^{-1}$ and $\tau_m = 10$ y.* Do these values link in some way to physics? If not, please explain the procedure behind the term *some experimentation*.

- Page 9, line 6: *..., with modest values of $\delta T$ ($\approx$ 1 K or less).* *Modest* with respect to what? I guess to melt-rates.

- Page 9, line 9: Typo: To me it appears that there is an orphan *A* between two sentences.

- Page 9, line 24: *Some biases can likely be attributed to errors in ocean thermal forcing (which is treated simply by the basin-scale melt parameterizations) and seafloor topography (e.g., an absence of pinning points, resulting in grounding-line retreat that can be compensated by spurious ocean cooling).* Could the errors by absence of pinning points also link to the relatively coarse resolution of the applied ice flow-model and the under-resolution of bedrock data in some regions? If so, please mention it.

- Page 9, line 30 and page 10, Fig. 2: *Thwaites Glacier is too thin and Pine Island Glacier is too thick;* For me, the contours in Fig. 2 make it virtually impossible to

spot thickness differences at the grounding lines or in smaller shelves. Perhaps, as you mention the Amundsen sea sector here in the text, some zoom-in Figure for that particular regions (just like Fig. 4 provides for speed) would be good. Neither is it possible to spot a clear outline of the Antarctic ice sheet. My suggestion to improve this, would be to have a neutral (e.g. white or blue) colour outside the ice-sheet/shelf area and only plot the grounding line. I, personally, have difficulties with the low contrast between the different shades of yellow and green in this figure. To me there seem to be jumps in ice thickness difference across many places at grounding lines (both with negative and positive signs). For instance, very prominently in the left panel for Amery. I would conclude – assuming that any evolved geometry is at least $C^0$ continuous across the grounding line – that also the difference has to be continuous. If you please could explain where this discrepancy arises from or else point me to my misconception of this figure.

• Page 11, Fig. 3: In this figure the colour scale actually works for me. I conclude that in Fig. 2 it is not about the colours themselves but the representation. Nevertheless, the black lines in Fig. 3 are not really visible, but in my view on that scale (different in Fig. 4) anyhow obsolete concerning the clear velocity jump between land-based ice and shelves.

• Page 11, Fig. 4: In large parts over Thwaites, the black grounding lines are really hard to spot over the dark green texture – perhaps a different colour (some grey?) would help. I mention that, as this is important since you refer to discrepancies of grounding line positions in this region in the text.

• Page 12, Fig 5: On my screen the underlying Antarctic ice-sheet shape is extremely difficult to spot. Please, enhance this. To me it comes clear that you are representing the nudged total values of $\delta T_{sector}$ (see earlier comment on Eq. (8)). Would it add value to the graph to include the calibrated values ISMIP 6 values for comparison?

- Page 14, Fig. 6: Since there are finite values of the ocean thermal forcing underneath land-ice, can you please explain their meanings?

- Page 16, Fig. 8: Please, within the caption of this figure, explain the red lines in the pictures, in particular the enclosed one within the Ross ice-shelf for *HadGem2/nonlocal MeanAnt* scenario. I guess they mark shelf areas, and I also guess they represent data for least/most sensitive melt scenario with the same red colour. If so, please distinguish them with lines of different colours or different pattern. Also, please, indicate the meaning of the black line – I guess it is the initial grounding line position.

- Page 17, line 3: Typo: orphan dot

- Page 18, line 1: *Ice sheet retreat in these simulations is modest for the next several decades.* In my view, this statement links to the calving simplification mentioned in the main point. You might address the following question: Could the initial slow response be linked to the fact that you are not able to fully capture effects of potential ice-shelf collapse? To some extent you mention this a few lines below (Page 18, line 17):*In terms of ice sheet physics, these simulations do not include hydrofracture or calving-front retreat, and thus are missing positive feedbacks associated with reduced buttressing of grounded ice by ice shelves (Sun et al., in review).* For me the connection with the timings of SLR appears to be important and is missing.

- Page 18, line 21: *At 4-km grid resolution, processes such as grounding-line retreat may be under-resolved.* I think it is fair to say that at 4 km resolution grounding line mechanics is underresolved, in particular if combined with the choice of sliding law in this study ().

- Page 18, line 25: *These uncertainties suggest several lines of research to further improve ice-sheet and sea-level projections:* What I am missing in this list here

is the aspect of changing the accuracy of the approximation applied in the ISM - in particular as I understood that CISM would have the possibility to solve LMLa (Blatter-Pattyn) equations. Would results improve if deploying higher order ISM approximations? Are they feasible given the time-intensive spinup process?

**References**

Gladstone, R. M., Warner, R. C., Galton-Fenzi, B. K., Gagliardini, O., Zwinger, T., and Greve, R.: Marine ice sheet model performance depends on basal sliding physics and sub-shelf melting, The Cryosphere, 11, 319–329, https://doi.org/10.5194/tc-11-319-2017, 2017.

Hindmarsh, R. C. A. (2004), A numerical comparison of approximations to the Stokes equations used in ice sheet and glacier modeling, J. Geophys. Res., 109, F01012, doi:10.1029/2003JF000065.

Seroussi, H., Nowicki, S., Payne, A. J., Goelzer, H., Lipscomb, W. H., Abe Ouchi, A., Agosta, C., Albrecht, T., Asay-Davis, X., Barthel, A., Calov, R., Cullather, R., Dumas, C., Gladstone, R., Golledge, N., Gregory, J. M., Greve, R., Hatterman, T., Hoffman, M. J., Humbert, A., Huybrechts, P., Jourdain, N. C., Kleiner, T., Larour, E., Leguy, G. R., Lowry, D. P., Little, C. M., Morlighem, M., Pattyn, F., Pelle, T., Price, S. F., Quiquet, A., Reese, R., Schlegel, N.-J., Shepherd, A., Simon, E., Smith, R. S., Straneo, F., Sun, S., Trusel, L. D., Van Breedam, J., van de Wal, R. S. W., Winkelmann, R., Zhao, C., Zhang, T., and Zwinger, T.: ISMIP6 Antarctica: a multi-model ensemble of the Antarctic ice sheet evolution over the 21st century, The Cryosphere Discuss., https://doi.org/10.5194/tc-2019-324, in review, 2020.

---

## Referee Comment (RC2) · Anonymous Referee #2 · 17 Mar 2020

This paper provides a detailed report on a high-profile Earth systems model's (Community Earth Systems Model, CESM2) contribution to the Ice Sheet Model Intercomparison Project (ISMIP6) for the Coupled Model Intercomparison Project - Phase 6 (CMIP6). Being an ice-sheet model intercomparison, the contribution from CESM2 is provided by the Community Ice Sheet Model (CISM2), a single component of CESM2. Atmospheric and oceanic forcings for the ice-sheet model are uncoupled, and provided by output of six atmosphere ocean general circulation models (AOCMs), including CESM2 (when run, called CCSM4.0). The results of the Antarctic portion of ISMIP6 is to be reported in Seroussi et al. (in review), and include results from CISM2.

The novelty of this paper, over Seroussi et al. (in review) is that it:

1. Concentrates on and expands consideration of the sub-shelf melting parameterization, which is the aspect of modeling that Seroussi et al. (in review) report as being *"The largest sources of uncertainty come from the ocean-induced melt rates, the calibration of these melt rates based on oceanic conditions taken outside of ice shelf cavities and the ice sheet dynamic response to these oceanic changes."*
2. Extends simulation to 2500 by maintaining forcing after 2100 based on late 21st century forcing from high emissions scenarios. These longer runs are intended to
   a. Identify the regions of Antarctica most vulnerable to retreat
   b. Determine the long-term ocean-forced sea-level rise from Antarctica.
3. Unstated by the authors, but important to me is that the paper lays out a number of heuristics and other modeling approaches that were used to bring CISM2 to the level of being a credible participant in ISMIP6. Such information is valuable to other modeling groups. While some of this information does appear in Seroussi et al. (in review), it is in the form of a single, terse paragraph.

In terms of novelty 1, I agree that the ocean-induced melt rates are critical to our projections of Antarctica's response to climate change. However, I see that the entire scheme is based on an unpublished work, Jourdain et al. (in review). I recognize that publication of such a large collaborative project as ISMIP6 is extremely complicated and dependencies such as this do arise. On the other hand, I hope the authors can appreciate that my role as a reviewer is to verify the assumptions made are sound. I can not do this when they are based on unpublished techniques. I am distressed by line 18 of section 2.2, which states that the "*climatology was interpolated to fill gaps and **extrapolated** into ice-shelf cavities*" (my emphasis on extrapolated). Extrapolation is dangerous business and I'm not willing to accept that everything is OK with that until it's been through peer-review. My reservations are even greater when considering this paper, which frames much of its novelty around the topic of sub-ice shelf melting.

Continuing with my assessment of novelty 1, I also see that the authors have deviated from the prescribed ocean temperature anomalies per sector, $\delta T_{\text{sector}}$ in order to assure the thickness in the region of the grounding line is similar to modern thicknesses. OK, but I'm not sure this is a tunable parameter. Jourdain apparently assured that the values of $\delta T_{\text{sector}}$ agreed with the estimated mean melt in each sector (no reference provided). No such assurances arise in the

author's method, making $\delta T_{\text{sector}}$ an unphysical parameter that is accounting for other model shortcomings. Page 12-13 acknowledge what has been done, and suggest that the lack of buttressing may be responsible for the mismatch. I'd liked to have seen more here. If the paper is valuable because it is taking a harder look at ocean-induced melt rates, and their coupling to ice dynamics, then it should investigate why unphysical parameterizations are needed for spin-up. Lack of buttressing? Maybe. What about using spin up targets that are in steady state? Plainy the Pine Island/Thwaites region isn't in steady state, why initialize the model as if it is? I believe these $\delta T_{\text{sector}}$ are the richest material coming from these simulations, but they aren't being investigated with enough depth. For example, consider figure 5. Why are many regions, for many simulations in a state that is so negative? The ice near the grounding line is too thin? Surely there is more at play. Also, why are some sectors reaching their constraints in terms of temperature forcing (-2.00). It's as if the model is trying to hold back a strong departure from the steady state. Why take such measures? It seems contrived.

Moving on to novelty 2, I just don't accept that these longer runs have value. The idea that ocean coupling, atmospheric coupling, and basal traction do not change on time scales of half a millennia is pure fantasy. Of course, it's also fantastical to imagine that one could do these couplings well, so the authors aren't to be faulted for not providing the couplings. Rather, I fault them for presenting these results at all. They simply are not credible.

Finally, novelty 3. There is considerable value in simply documenting the complex work that goes into modeling on this scale, at this level of complexity. The paper does that, and does so in a way that is well presented and understandable. I think that the paper is quite honest in that there were no major components of the exercise that were left out, obfuscated, or overstated. A sensible argument for publication is that the cost of publication is low, and the quality of search is high. While the paper is highly specialized, the authors are outstanding professionals in the field and this is a useful report on what they have been doing. I only see one problem with this point of view - there were 15 groups participating in Seroussi et al. (in review). Are all of them to have companion papers detailing the way they handled ISMIP6? That seems really inefficient. I wonder if there couldn't be a single companion paper?

In conclusion, I don't feel that the paper meets its own criteria for novelty because Jourdain is unpublished, because the need to alter the thermal forcing correction isn't explored in enough depth, and because the longer runs don't provide much value. The paper might provide novelty in that the details are useful, but that could be satisfied more efficiently with a companion paper that included more ISMIP6 participants.

Provided Jourdain is published, I could support publication of this manuscript with what I'd call 'minor' revisions, which I hope would include an expanded discussion of the thermal forcing correction, the need to push the model to steady state, and a consideration of thermal forcing that produces refreezing. All of these expanded discussions should come at the expense of the long term runs. That said, the option I favor most would be a companion paper that includes

more ISMIP6 participants, and that lays out the details of modeling for the experiments in ISMIP6.

I have a number of comments on individual passages of the paper as well.

Page 1 line 7, 'nudge' seems chatty and imprecise. "Constrain to observed values"?

Page 1, line 13: say what the missing physics are.

Page 2, line 34: say what is meant by 'committed'.

Page 3, line 1: Be more consistent with the purpose of the paper, the abstract makes quantitative claims ("10 cm to nearly 2 m"), I'm not sure this sentence really fits with the approach in much of the rest of the paper.

Page 2 Lines 5-9 it's as if the authors agree the exercise is pointless, but are rationalizing the CPU allotments. Just drop it.

Page 4, line 1-16, this is really a long road to just stating that the power-law sliding law is used. The 'nearly all' statement in line 17 is frustrating too. Where is the relation Coulomb? What's the point of having effective pressure in relations if one just assumes that it is ice overburden. This entire discussion could be collapsed to something much more concise.

Page 4, line 30 - I'm often confused by the (unpublished) ISMIP6 protocols. Here you say CISM runs are based on the protocols. Page 7, line 5 says your differ from protocols with regard to $\delta T_{\text{sector}}$. That's a fairly big deviation from protocols. So...be consistent in saying what you've done.

Page 5, line 11 - are these mean annual, monthly, daily atmospheric forcings?

Page 5, line 35 - This is odd - I guess you force the thickness near the grounding line to be close to observations by altering $\delta T_{\text{sector}}$, but then protect against the large negative values found for many sectors by zeroing out the contribution to melt? So, you have refreezing (I guess) if it aids thickness on initialization, but don't allow refreezing in forward runs? This really isn't clear. Is equation 4 used for spinup, and the modified version used for forward modeling? Or, is the modified eq. 4 used for both? If so, then the negative values of $\delta T_{\text{sector}}$ never have any impact, do they? Eqn 5 also eliminates refreezing.

Here is the big question: Aren't you introducing a big transient when (if) you jump between the refreezing allowed in the initialization and the melt only situation in the forward runs? Is this transient physical (I don't think it is)? Again, this is the novelty of the paper, but here is another case where it's not being treated with enough consideration.

Page 8, line 22: OK - here we learn that this is not what was done as part of CISM2's contribution to Seroussi et al. (in review). This should have been scoped out in the introduction, as part of the novelty, with a clear statement of what the primary difference is.

Page 9, line 17: I'm not aware of an analytical solution to the temperature equation that can accomodate advection of heat. Please provide more detail.

Page 9, line 19, specify the extent of the ice that is considered for each type of 'nudge'.

Figure 5 - for context provide the $\delta T_{\text{sector}}$ from Jourdain.

Page 12, line 6-7. This is important to me, but the treatment not satisfactory.

Page 17, line 5 not really based on the protocols, right? "Loosely based"?

Page 18, line 9-11 - this is frustratingly vague.

---

## Author Comment (AC1) · 2 Oct 2020

**Reviewer comments and responses**

To the Editor:

We have attached the reviewer comments with our responses.  We thank both reviewers for their careful analysis and constructive suggestions.  Before proceeding with the responses, we thought it would be useful to summarize some substantial revisions of the original manuscript.

Reviewer 1 suggested that we look into the sensitivity of results to the Stokes approximation (i.e., Blatter-Pattyn vs. DIVA) and the treatment of calving and shelf collapse.  Reviewer 2 emphasized the uncertainties introduced by our spin-up procedure, in which a thermal forcing correction (sometimes large and negative) is applied in each basin.  Both reviewers pointed out that with the assumption of effective pressure equal to overburden, we were not simulating Coulomb sliding.  Also, it is well known that the grounding line is under-resolved at our default grid resolution of 4 km.

In response, we have expanded Section 4, "Results of projection experiments."  In addition to our "standard" results using the DIVA solver at 4 km resolution with power-law sliding, we added a subsection (4.2) exploring the following sensitivities:
- Grid resolution; we added experiments at 8-km and 2-km resolution
- Stokes approximation; we ran some experiments with the Blatter-Pattyn solver
- Ice-shelf collapse; we applied the shelf-collapse maps provided by ISMIP6 and also tested a more extreme scenario in which all small shelves collapse
- Basal sliding law; we ran experiments with $p = 0.5$ and $p = 1$ in the expression for effective pressure (where $p = 0$ corresponds to power-law sliding)

In response to Reviewer 2, we also made some numerical and parameter changes that had the effect, in some sectors, of stabilizing grounding lines and reducing negative temperature corrections.  We repeated all the standard experiments with these changes included, in addition to the new sensitivity experiments.

For the most part, the sensitivities are modest, and our main conclusions have not changed.  We did, however, find some configurations for which the Thwaites Basin collapses, in contrast to the standard configuration in which Thwaites retreat is modest.  We added a subsection (4.3) exploring this threshold behavior.

For forward runs, we changed the nominal initialization date for the spin-up from 1995 to 1950.  This allowed us to incorporate ESM anomalies going as far back as 1950.  Each forward run is now 550 years, from 1950 to 2500.

We changed the title slightly.  It now begins with "ISMIP6-based projections" instead of "ISMIP6 projections", reflecting our experiments diverge in several ways from the ISMIP6 protocols.

Our detailed responses are below, with reviewer comments in black font and responses in blue. On request, we can provide a revised manuscript, either with or without changes tracked from the first version.

Thank you for your patience and consideration.

**Reviewer #1** (Thomas Zwinger):

**1 General Impression**

This is a manuscript that complements the currently reviewed ISMIP6 intercomparison on the Antarctic ice sheet. Beyond the contribution that was already accounted for in the paper by Seroussi et al. (Seroussi et al., 2020), the authors show new ways of spin-up and present computations beyond the time-frame of ISMIP 6.

The text describes in detail the methods used and the assumptions applied within the contribution from the ice-sheet model CISM to this inter-comparison. From my point of view, the paper to a large extent analyses the impacts of assumptions and approximations on the results, except for two open points which I will raise in the section below. The article is well written and has a clear line to follow. I got the impression of a few inaccuracies in the notation, which in my view is easily to be fixed. I also have some suggestions on how to improve or augment the figures.

Modelling techniques presented here are mainly variations of existing approaches, but the size and the context of the applications make for the scientific novelty of this paper. I also see the value in documenting the techniques leading to these results, which of course cannot be presented in such detail in an inter-comparison paper. In that sense, I see it as a useful contribution to the literature, also to demonstrate what efforts have to be taken in order to realise such an implementation.

I mainly placed a few technical suggestions to (according to my subjective impression) improve the readability of the text, in particular for readers not being that familiar with ISMIP6. Besides this, the only more substantial criticism I would place is the in my view not complete analysis of the impact on the results of some of the assumptions made, in particular the over-simplification of calving and the impact of the (actually not clearly revealed) approximation to the Stokes equation and the resolution applied in the ice-sheet model. If these points are addressed, I would recommend this manuscript for publication in TC.

**2 Main points**

As mentioned above, I would see it necessary to have two in my view not completely clear issues to be elaborated:

1. The influence of the in my view oversimplified calving law on the results, in particular its impact on grounding line migration and retreat

We agree that the no-advance calving law is very simple and that ice-shelf collapse (not explored in our original manuscript) could significantly affect future grounding-line retreat. We are unable to present results with a physically based calving law, because we were not able to obtain a satisfactory Antarctic spin-up with such a law. However, we added two calving sensitivity tests, as described below.

2. The choice of the approximation to the Stokes equation and its consequences on the accuracy and sensitivity of the results

Please see below.

Concerning the first point, i.e. calving, you explain (page 4, line 20): *Instead, we use a no-advance calving mask, removing all ice that flows beyond the observed calving front. The calving front can retreat where there is more surface and basal melting than advective inflow, but more often the calving front remains in place.* Here you might mention the point in time of the observation of the prescribed front (I presume it somehow refers to your initial state in 1995). In the Conclusions, you mention it in just one sentence: *In terms of ice sheet physics, these simulations do not include hydrofracture or calving-front retreat, and thus are missing positive feedbacks associated with reduced buttressing of grounded ice by ice shelves (Sun et al., in review).* In my opinion, the reader would benefit from some clarification on how much impact neglecting the possibility of ice-shelf collapse (as it would be provided by ISMIP 6 forcing) on the resulting sea level has. In my view, the current approach significantly might lead to an underestimation of sea-level rise, in particular at times beyond the year 2050. One additional experiment could be to, for instance, just remove the shelf in front of Thwaites and see what short-time impact this has for this region – maybe in combination with the upper end combination of AOGM and ocean forcing. Or to apply the shelf-collapse scenario from ISMIP 6. If this is for some technical reason not possible at this stage, please mention that in the paper.

We added two calving sensitivity tests:

1. For each melt parameterization and ESM, apply the ice-shelf collapse masks provided by ISMIP6 for the optional shelf-collapse scenarios (in combination with the ESM thermal forcing anomalies). This option was not available in CISM at the time of the original submission, but is now supported.
2. For each melt parameterization and ESM, force the abrupt collapse of all small ice shelves in year 2100. Here, "small" is defined as all ice shelves outside of sectors 2, 7 and 14, which correspond to the Amery, Ross, and Ronne-Filchner Shelves, respectively. We took this approach because (1) no ESM in our ensemble shows significant collapse in these three basins before 2100, and (2) the effects of large

ice-shelf collapse have recently been explored by Sun et al. (2020). Thwaites is among the shelves removed, so this test incorporates the suggestion above.

We found that the effects of shelf collapse are not negligible, but are smaller than the mass loss associated with ocean thermal forcing anomalies for most ESM and melt configurations. ISMIP6-prescribed shelf collapse typically adds 5 to 15 cm to the SLR due to ocean thermal forcing anomalies alone. The collapse of all small shelves has a larger effect, contributing additional SLR of 15 to 25 cm. Removal of the Thwaites shelf does not lead to much additional retreat in the Amundsen Sea basin, suggesting that the current buttressing effects are small.

Results of these shelf-collapse experiments are described in the new Section 4.2, "Sensitivity Experiments".

On the second point, model approximation, you present (Page 3, line 30) that the CISM code includes options for the following approximations to the Stokes equations: *SSA, L1L2, LMLa* – mainly following Hindmarsh' nomenclature (Hindmarsh, 2004). On page 4, line 1, you write: *The simulations for this paper use the depth-integrated solver, which gives a good balance between accuracy and efficiency for continental-scale simulations (Lipscomb et al., 2019)*. I consider SSA as well as L1L2 as depth integrated (the latter of course with some corrections from a vertical profile). Could you please clearly identify which of the previously listed models you deploy in your study? I am further confused how a depth integrated model can use 5 vertical layers (as described in the paragraph before), but assume that you use this for some aspects different from depth-integrated parts, such as temperature evolution or velocity corrections. Is this vertical resolution sufficient to resolve the physics?

We used CISM's DIVA solver, which falls in the L1L2 category of Stokes approximations. The solver is two-dimensional in the sense that it solves a global 2D elliptic equation for the mean values of u and v in each column. Given the mean values, it is possible to compute a full velocity profile following Goldberg (2011). The full 3D velocity profile is then used to advect thickness and temperature.

For Antarctica, we found five vertical levels to be sufficient; changes are small when we add more levels. More levels are preferable if the velocity is dominated by vertical shear, but in our regions of interest (near grounding lines), the flow is dominated by basal sliding.

We added some text in the Introduction to make these points more clear. The revised paragraph reads:

> CISM runs on a structured rectangular grid with a terrain-following vertical coordinate. The simulations in this paper were run on a 4-km grid with 5 vertical levels, as for the CISM contributions to ISMIP6 projections (Seroussi et al., 2020). At 4-km resolution, grounding lines are under-resolved, but this is the finest resolution that permits a large suite of whole-Antarctic CISM simulations at reasonable computational cost. Increasing the number of vertical levels would not substantially change the results, especially in

regions dominated by sliding rather than vertical shear.  Scalars (e.g., ice thickness H and temperature T) are located at grid cell centers, with horizontal velocity $\mathbf{u}$ = (u,v) computed at vertices. The dynamical core has parallel solvers for a hierarchy of approximations of the Stokes ice-flow equations, including the shallow-shelf approximation (MacAyeal, 1989), a depth-integrated higher-order approximation (Goldberg, 2011), and the 3D Blatter-Pattyn (BP) higher-order approximation (Blatter, 1995; Pattyn, 2003}. The latter two approximations are classified as L1L2 and LMLa, respectively, in the terminology of Hindmarsh (2004).  The simulations for this paper use the depth-integrated solver, known as DIVA (Depth-Integrated Viscosity Approximation), which solves a 2D elliptic equation for the mean velocity, followed by a vertical integration at each vertex to obtain the full 3D velocity.  This solver gives a good balance between accuracy and efficiency for continental-scale simulations (Lipscomb et al., 2019).  In section 4.2, we compare DIVA results to a small set of runs using the more expensive BP solver.

Can you include some lines on how much you estimate the influence of the choice of the approximation and the chosen resolutions on the results of your study. For instance, would one expect to reach the same conclusions if using a LMLa model or perhaps the current model with a finer resolution?

This is a good question that motivated some additional sensitivity experiments, described in the new Section 4.2. CISM's Blatter-Pattyn (LMLa) solver is more than an order of magnitude more expensive than DIVA.  Within our computing budget, we were unable to do long spin-ups with the BP solver at 4 km.  However, we were able to run four spin-ups (nonlocal and nonlocal-slope melt parameterizations, with MeanAnt and PIGL calibrations) using the BP solver at 8 km, and then carried out projection experiments using this solver.  We compared the results to those obtained with the DIVA solver with the same resolution and forcing.

Differences between DIVA and BP are small for all the experiments we ran.  Depending on the melt scheme and ESM, the difference in cumulative SLR between BP and DIVA varies from about -4 cm to +2 cm.  We have run other DIVA–BP comparisons at 4-km resolution (e.g., for Greenland in Lipscomb et al., 2019) or finer (in idealized tests such as MISMIP3d and MISMIP+, as described by Leguy in a paper to be submitted).  We do not think that replacing DIVA with BP would change our main conclusions.

As for the effect of finer resolution, we ran four additional spin-ups (one for each melt parameterization/calibration) at 2-km resolution. We then ran the usual suite of ESM experiments. To stay within our computing budget, we ran the spin-ups for 10,000 instead of 20,000 model years; as a result, the drift in the control run is larger than for the 4-km runs, but still small compared to the sea-level signal.  Generally, higher resolution leads to greater ice sheet retreat.  Compared to the 4-km runs, the total SLR in 2-km runs is about 10 to 20% larger, or ~10 cm for the high-end scenarios.  These differences are not large enough to change the main conclusions.

**3 Suggested corrections, additions and typos**

Please, find suggestions for corrections, changes and typos in the order of their occurrence in the text:

Page 1, Author list: Typo: Nicolos → Nicolas

We corrected the typo.

Page 2, line 28: *To study long-term ice sheet evolution, we extend the simulations to 2500, . . .* Minor issue, but for better readability, please mention that this is a date (it could be iterations or any other time unit).

We revised the text as suggested.

Page 4, line 6, 9 and 11: Again, minor issue, but in order to stay consistent with the nomenclature in equation (1), I would use the subscript for bedrock in all the following occurrences of the basal velocity, i.e., u → ub.

We made the suggested change.

Page 4, line 13: Can you elaborate on the choice of exactly this value for Cc? Is this representing some tested optimum? Or (see next question) does it not really matter what to put there?

This parameter is not well constrained by data, and we are not aware of formal tests that established it as an optimum. The value we chose is the recommended value in Asay-Davis et al. (2016), used by MISMIP+ participants. We revised the text to cite Asay-Davis et al. (2016) instead of Pimentel et al. (2010), who used a slightly different value.

Near grounding lines, the value of $C_c$ does matter. So does the value of p, since the Coulomb friction law contains the combination ($C_c/N$), where $N(p)$ decreases monotonically with increasing p. Thus, it is somewhat arbitrary whether to vary $C_c$ or p in a sensitivity study. We held $C_c$ fixed but varied p (with values of 0.5 and 1.0) in several sensitivity experiments.

Page 4, line 17: *Setting* N *to overburden pressure implies power-law behavior in nearly all of the ice sheet.* To my understanding, this means that power-law.

Sliding then most likely prevails down to the grounding line - or am I missing something? If this is the case, I would question the benefit of explaining sliding law (1) and the Coulomb sliding parameter therein in such a detail. Being aware that in lack of a detailed representation of pinning points, any sliding law that reaches zero resistance at the grounding line inherently causes issues (confirmed also by our experience with Full-Stokes models at similar resolution), I do not criticise the approach as such, but would welcome to include some information on how high the friction parameters at the grounding line actually are (or have to be in order to be

stable) and how – also by keeping them constant relative to a moving grounding line – that impacts the prognostic runs.

This is correct.  Using the Schoof law with $p = 0$, and thus setting $N$ to overburden pressure, is very similar to using a power law.  Our standard experiments now use a power law, which is another CISM option in addition to the Schoof law.

In response to this comment, and another comment from Reviewer 2, we decided to run a set of sensitivity experiments using the Schoof law (Eq. 1) with $p = 0.5$ and $p = 1$.  We revised the text in Section 2.1 accordingly, emphasizing that the power law is the default.  Running CISM with $p = 1$ (hence zero resistance at the grounding line) slows down the code significantly, but the velocity solver still converges.  We think the new results are of interest.  In particular, the Thwaites basin is more prone to collapse with $p = 1$ than with $p = 0$, as discussed in the revised Section 4.

Near the top of p. 8 in the original text, we state the maximum value of $C\_p$, which is high enough to effectively turn off sliding. In basins where the grounding line significantly retreats, we do introduce errors by using the tuned $C\_p$ values, either because these values are the wrong magnitude or because they should be changing in time. It is hard to estimate the size of the error.  This is a subject for a future study, perhaps with a physically based model of subglacial hydrology.

The paragraph on biases and limitations of $C\_p$ inversion now reads:

> This method works well in keeping most of the grounded ice near the observed thickness.  Also, since $C\_p$ is independent of the ice thermal state, we remove low-frequency oscillations associated with slow changes in basal temperature, resulting in a better-defined steady state.  In forward runs, however, $C\_p(x,y)$ is held fixed and cannot evolve in response to changes in basal temperature or hydrology. As a result, we can have unphysical basal velocities when the ice dynamics differs from the spun-up state. Also, the tuning of $C\_p$ can compensate for other errors. For example, if the topographic data set is missing pinning points near the grounding line, the ice will be biased thin, and $C\_p$ can be driven to high values to make up for the lack of buttressing. These biases should be kept in mind when interpreting results.

Page 5, line 5: *(The simulations described here use only the ocean forcing.)* I do not understand the meaning of this sentence. Does it mean, you do not apply any atmospheric, but only ocean forcing? Or does this only apply in connection to CMIP? Please, elaborate. On a side-note, in my personal opinion, I find it strange to put whole separate sentences in brackets. I would write it without. I mention this, as it occurs a few times in the text.

We apply the same atmospheric forcing (SMB from RACMO) as in the spin-up, but without adding SMB anomalies from the ESMs.  We revised the text as follows:

The ISMIP6 projection experiments are run with standalone ice sheet models, forced by time-varying, annual-mean atmosphere and ocean fields derived from the output of CMIP5 and CMIP6 ESMs. In our study, the projection experiments use ocean forcing that evolves during the simulation, but the atmospheric forcing (specifically, the surface mass balance) is fixed to the values used when spinning up the model.

We also removed parentheses around sentences in several places.

Page 8, line 4: *Eq. 6 is based on the equation for a critically damped harmonic oscillator, where the first term in brackets nudges H toward Hobs, and the second term damps the nudging to prevent overshoots. (The damping is not exactly critical, however, because* dCp/dt *is not exactly proportional to* d2H/dt2.*)* I am slightly confused by this statement, since from my point of view, if dCp/dt would be somehow proportional to d2H/dt2, the Cp on the right-hand-side would introduce some functional relation to H and I would not immediately see the equation of a damped harmonic oscillator recovered by (6). Could you clarify or insert a reference where this is explained in detail in order to help the reader to follow that up. Minor issue: *Eq. 6* → Eq. (6).

We removed the language about the critically damped harmonic oscillator, because the analogy with classical mechanics is perhaps not close enough to be helpful.  The text now simply describes the role of the two terms.

Throughout the text, we added parentheses around equation numbers.

Page 8, line 6: *We hold* Cp *within a range between* $10^2$ *and* $10^5$ *Pa m*−1/3 *y*−1*, since smaller values can lead to numerical instability, and larger values do not significantly lower the sliding speed.* Can you explain the nature of the instabilities? I wonder if this is somehow linked to the fact that you had to fix the effective pressure to the overburden pressure in the sliding law.

The instabilities appear in the form of CFL violations, i.e., unrealistically fast velocities that can crash the model.  We do not think this instability is related to fixing *N* to overburden pressure; rather, it comes from having close to zero friction in some lightly grounded cells near the grounding line.

We revised the text as follows:

We hold C_p within a range between 10^2 and  10^5 Pa m^{-1/3} y^{-1}.  Smaller values can lead to excessive sliding speeds when the basal friction approaches zero.  With C_p at its maximum value, basal sliding is close to zero, and there is little benefit in raising it further.

Page 9, line 7 and Eq. (8): Based on results shown in Fig. 5, I conclude that the symbol δT stands for the nudged values of the ISMIP 6 protocol given δTsector. Simply, the missing

subscript might also lead to the conclusion that it is the difference to this reference value, i.e., the correction to the correction. I would consistently add the subscript *sector* to δT .

We revised the notation as suggested.

Page 9, line 5: *After some experimentation, we set* mT = 10 *m y*−1 *K*−1 *and* τm = 10 *y.* Do these values link in some way to physics? If not, please explain the procedure behind the term *some experimentation*.

Briefly, we adjusted the values so as to achieve a relatively fast relaxation to equilibrium without oscillations.  This was a process of trial and error, with values linked only loosely to physics.

Rather than lengthen the text with an explanation, we decided to remove the phrase "after some experimentation" and simply state the values.

Page 9, line 6: *. . . , with modest values of* δT *(≈ 1 K or less). Modest* with respect to what? I guess to melt-rates.

We were thinking "modest with respect to the background thermal forcing", which typically is 1° or more in regions with significant melt.  But since "modest" is a relative term, we rewrote the text to state simply that |dT_sector| < 1 K or less in most basins.  The exceptions are the Aurora and Amundsen sectors, which are discussed later.

Page 9, line 9: Typo: To me it appears that there is an orphan *A* between two sentences.

We fixed the typo.

Page 9, line 24: *Some biases can likely be attributed to errors in ocean thermal forcing (which is treated simply by the basin-scale melt parameterizations) and seafloor topography (e.g., an absence of pinning points, resulting in grounding- line retreat that can be compensated by spurious ocean cooling).* Could the errors by absence of pinning points also link to the relatively coarse resolution of the applied ice flow-model and the under-resolution of bedrock data in some regions? If so, please mention it.

We agree.  We added the following sentence: "Pinning points could either be missing in the high-resolution (0.5 km) topographic data set (Morlighem et al., 2019b), or smoothed away when interpolating to the coarser CISM grid."   We hoped that pinning points would be better resolved on the 2-km CISM grid, but this was not generally the case, suggesting that at least some pinning points are missing (or are located at too great a depth) in the BedMachine data.

Page 9, line 30 and page 10, Fig. 2: *Thwaites Glacier is too thin and Pine Island Glacier is too thick;* For me, the contours in Fig. 2 make it virtually impossible to spot thickness differences at the grounding lines or in smaller shelves. Perhaps, as you mention the Amundsen sea sector here in the text, some zoom-in Figure for that particular regions (just like Fig. 4 provides for speed) would be good. Neither is it possible to spot a clear outline of the Antarctic ice sheet. My

suggestion to improve this, would be to have a neutral (e.g. white or blue) colour outside the ice-sheet/shelf area and only plot the grounding line. I, personally, have difficulties with the low contrast between the different shades of yellow and green in this figure. To me there seem to be jumps in ice thickness difference across many places at grounding lines (both with negative and positive signs). For instance, very prominently in the left panel for Amery. I would conclude – assuming that any evolved geometry is at least C0 continuous across the grounding line – that also the difference has to be continuous. If you please could explain where this discrepancy arises from or else point me to my misconception of this figure.

We agree that the color scheme for this and other figures was poorly chosen.  In the revised figure, we use a different color scheme with white for the Southern Ocean (outside the ice sheet boundary), pale yellow for values near zero within the ice sheet, blue for positive, and red for negative.  The differences are, indeed, continuous; the apparent jumps were a result of the color scheme.

We also added a thickness-difference figure zooming in on the Amundsen sector.

Page 11, Fig. 3: In this figure the colour scale actually works for me. I conclude that in Fig. 2 it is not about the colours themselves but the representation. Nevertheless, the black lines in Fig. 3 are not really visible, but in my view on that scale (different in Fig. 4) anyhow obsolete concerning the clear velocity jump between land-based ice and shelves.

We found another color scale we preferred for Fig. 3, with red for slow, blue for fast, and pale yellow for intermediate.  We omitted the black contours, since as you say, shelves are obvious from the velocity jump.

Page 11, Fig. 4: In large parts over Thwaites, the black grounding lines are really hard to spot over the dark green texture – perhaps a different colour (some grey?) would help. I mention that, as this is important since you refer to discrepancies of grounding line positions in this region in the text.

We revised the color scheme as described above for Fig. 3, making the grounding-line contours easier to see.

Page 12, Fig 5: On my screen the underlying Antarctic ice-sheet shape is extremely difficult to spot. Please, enhance this. To me it comes clear that you are representing the nudged total values of δTsector (see earlier comment on Eq. (8)). Would it add value to the graph to include the calibrated values ISMIP 6 values for comparison?

Yes, this is the nudged value of dT_sector.  In the revised figure, we have enhanced the Antarctic background shape and added another plot to show the calibrated ISMIP6 values for comparison.  It was cleaner to make two plots than to put all the numbers in one plot.

Page 14, Fig. 6: Since there are finite values of the ocean thermal forcing underneath land-ice, can you please explain their meanings?

Ocean thermal forcing is masked out for fully grounded cells, but is available in the forcing data in case these cells should later become ungrounded. We added text in the caption to clarify that basal melting due to ocean thermal forcing is applied only in floating cells.

Page 16, Fig. 8: Please, within the caption of this figure, explain the red lines in the pictures, in particular the enclosed one within the Ross ice-shelf for *HadGem2/nonlocal MeanAnt* scenario. I guess they mark shelf areas, and I also guess they represent data for least/most sensitive melt scenario with the same red colour. If so, please distinguish them with lines of different colours or different pattern. Also, please, indicate the meaning of the black line – I guess it is the initial grounding line position.

Sorry, the black and red lines (marking initial and final grounding line positions) were not explained in the caption and were hard to interpret. The revised figure uses an improved color scale, similar to Fig. 2 as described above. We decided to show only the initial shelf boundaries (using black contours), since a second set of contours is hard to read. Regions of grounding-line retreat are easy to identify, since they have the largest decrease in thickness above flotation.

Page 17, line 3: Typo: orphan dot

We fixed the typo.

Page 18, line 1: *Ice sheet retreat in these simulations is modest for the next several decades.* In my view, this statement links to the calving simplification mentioned in the main point. You might address the following question: Could the initial slow response be linked to the fact that you are not able to fully capture effects of potential ice-shelf collapse? To some extent you mention this a few lines below (Page 18, line 17): *In terms of ice sheet physics, these simulations do not include hydrofracture or calving-front retreat, and thus are missing positive feedbacks associated with reduced buttressing of grounded ice by ice shelves (Sun et al., in review).* For me the connection with the timings of SLR appears to be important and is missing.

We addressed this question with the shelf-collapse experiments discussed above. Since the ISMIP6 shelf-collapse data sets have most collapse happening late in this century, this collapse cannot accelerate ice loss earlier in the century. The other set of calving experiments, in which all small shelves collapse in 2100, does trigger an immediate speed-up in ice loss, but this is an extreme scenario, unlikely to occur in the next few decades. Thus, we do not think the lack of shelf collapse is the primary reason for the initial slow response.

We suggest two other reasons for the initial slow retreat:

- For the Ross and Ronne sectors, it takes some time for the ocean to warm sufficiently to induce retreat.

- For the Amundsen sector, the ice is initialized in steady state with a large negative thermal forcing correction, and the ESM anomalies that are applied later are not very large compared to the climatological thermal forcing. Thus, our runs fail to capture ASE ice loss that is already under way. The initialization procedure may be inappropriate in regions that have recently warmed compared to preindustrial forcing.

We have expanded on these points in the revised Sections 4 and 5.

Page 18, line 21: *At 4-km grid resolution, processes such as grounding-line retreat may be under-resolved.* I think it is fair to say that at 4 km resolution grounding line mechanics is underresolved, in particular if combined with the choice of sliding law in this study ().

We agree. The text now reads, "Processes such as grounding-line retreat are under-resolved."

This concern is addressed in part by the sensitivity runs at 2-km resolution, showing increased ice loss at higher resolution but not qualitatively different behavior. In MISMIP3d and MISMIP+ experiments (as described in a manuscript in prep by Leguy et al.), we have explored resolution dependence in detail. We have generally found that numerical errors from running the model behavior at 2–4 km, as compared to 0.5–1 km, are small compared to other sources of uncertainty (e.g., melt rates and basal sliding parameters).

Page 18, line 25: *These uncertainties suggest several lines of research to further improve ice-sheet and sea-level projections:* What I am missing in this list here

Is the aspect of changing the accuracy of the approximation applied in the ISM - in particular as I understood that CISM would have the possibility to solve LMLa (Blatter-Pattyn) equations. Would results improve if deploying higher order ISM approximations? Are they feasible given the time-intensive spinup process?

As explained above, it is not feasible to run a large suite of experiments using the Blatter-Pattyn solver at a resolution of 4 km, or even 8 km, because of the greater cost. But the small DIVA/BP differences at 8-km resolution support the choice of DIVA and suggest that depth-averaging in the Stokes approximation is a relatively small source of uncertainty.

**Reviewer #2:**

This paper provides a detailed report on a high-profile Earth systems model's (Community Earth Systems Model, CESM2) contribution to the Ice Sheet Model Intercomparison Project (ISMIP6) for the Coupled Model Intercomparison Project - Phase 6 (CMIP6). Being an ice-sheet model intercomparison, the contribution from CESM2 is provided by the Community Ice Sheet Model (CISM2), a single component of CESM2. Atmospheric and oceanic forcings for the ice-sheet model are uncoupled, and provided by output of six atmosphere ocean general circulation

models (AOCMs), including CESM2 (when run, called CCSM4.0). The results of the Antarctic portion of ISMIP6 is to be reported in Seroussi et al. (in review), and include results from CISM2.

The novelty of this paper, over Seroussi et al. (in review) is that it:

1. Concentrates on and expands consideration of the sub-shelf melting parameterization, which is the aspect of modeling that Seroussi et al. (in review) report as being *"The largest sources of uncertainty come from the ocean-induced melt rates, the calibration of these melt rates based on oceanic conditions taken outside of ice shelf cavities and the ice sheet dynamic response to these oceanic changes."*

2. Extends simulation to 2500 by maintaining forcing after 2100 based on late 21st century forcing from high emissions scenarios. These longer runs are intended to

   - Identify the regions of Antarctica most vulnerable to retreat

   - Determine the long-term ocean-forced sea-level rise from Antarctica.

3. Unstated by the authors, but important to me is that the paper lays out a number of heuristics and other modeling approaches that were used to bring CISM2 to the level of being a credible participant in ISMIP6. Such information is valuable to other modeling groups. While some of this information does appear in Seroussi et al. (in review), it is in the form of a single, terse paragraph.

In terms of novelty 1, I agree that the ocean-induced melt rates are critical to our projections of Antarctica's response to climate change. However, I see that the entire scheme is based on an unpublished work, Jourdain et al. (in review). I recognize that publication of such a large collaborative project as ISMIP6 is extremely complicated and dependencies such as this do arise. On the other hand, I hope the authors can appreciate that my role as a reviewer is to verify the assumptions made are sound. I can not do this when they are based on unpublished techniques. I am distressed by line 18 of section 2.2, which states that the "*climatology was interpolated to fill gaps and **extrapolated** into ice-shelf cavities*" ( my emphasis on extrapolated). Extrapolation is dangerous business and I'm not willing to accept that everything is OK with that until it's been through peer-review. My reservations are even greater when considering this paper, which frames much of its novelty around the topic of sub-ice shelf melting.

The paper by Jourdain et al. (2020) has since been published in *The Cryosphere*.  We agree that extrapolation introduces large uncertainties, but we think it is a defensible procedure in the absence of high-resolution ocean model simulations.   Since our paper is largely an exploration of the forcing provided by ISMIP6, we thought it was beyond our scope to explore other ways of estimating thermal forcing in ice shelf cavities.  More realistic approaches, including sub-cavity ocean simulations, will be explored in the upcoming MISOMIP2 project, co-led by Nicolas Jourdain.

We added some text in Section 2.2 on the reasons for using an extrapolation (new text in italics):

> The result is a time-varying 3D product that can be vertically interpolated to give the thermal forcing at the base of a dynamic ice shelf at any time and location in the Antarctic domain. *The extrapolation procedure does not account for ocean circulation in cavities, but was seen as a necessary simplification in the absence of output from high-resolution regional ocean models.*

Continuing with my assessment of novelty 1, I also see that the authors have deviated from the prescribed ocean temperature anomalies per sector, in order to assure the thickness in the region of the grounding line is similar to modern thicknesses. OK, but I'm not sure this is a tunable parameter. Jourdain apparently assured that the values of agreed with the estimated mean melt in each sector (no reference provided). No such assurances arise in the author's method, making an unphysical parameter that is accounting for other model shortcomings. Page 12-13 acknowledge what has been done, and suggest that the lack of buttressing may be responsible for the mismatch. I'd liked to have seen more here. If the paper is valuable because it is taking a harder look at ocean-induced melt rates, and their coupling to ice dynamics, then it should investigate why unphysical parameterizations are needed for spin-up. Lack of buttressing? Maybe. What about using spin up targets that are in steady state? Plainly the Pine Island/Thwaites region isn't in steady state, why initialize the model as if it is? I believe these are the richest material coming from these simulations, but they aren't being investigated with enough depth. For example, consider figure 5. Why are many regions, for many simulations in a state that is so negative? The ice near the grounding line is too thin? Surely there is more at play. Also, why are some sectors reaching their constraints in terms of temperature forcing (-2.00). It's as if the model is trying to hold back a strong departure from the steady state. Why take such measures? It seems contrived.

We think that dT_sector can fairly be considered a tunable parameter, given the sparseness of temperature data in ice-shelf cavities. Jourdain et al. (2020) wrote that this parameter accounts "for biases in observational products, ocean property changes from the continental shelf to the ice shelf base (not accounted for in the aforementioned extrapolation), tidal effects and other missing physics". They tuned this parameter to match sector-averaged melt rates, which for most sectors are very uncertain. Our tuning is done with a different target: observed ice thickness instead of melt rates. In most cases, our corrections are of the same order of magnitude as the corrections made by Jourdain et al. (2020); the exceptions are the Aurora and Amundsen sectors. We agree that the Amundsen Sea sector deserves closer consideration; see below.

In Fig. 5, the negative values can be explained by a combination of (a) observational error, (b) missing pinning points, which need to be compensated by reduced melting, (c) excessive melt rates for the given thermal forcing, or (d) model error (e.g., inadequate spatial resolution). All factors are likely at play. For example, the negative corrections are larger for the PIGL calibration than for MeanAnt calibration, which could be taken as support for MeanAnt. Also,

the melt parameterizations are missing ocean feedbacks that could potentially reduce melt rates.  The system is too complex to isolate one primary factor.

For the revised manuscript, we addressed these critiques as follows:

- We made some numerical changes in CISM that reduced the magnitude of some of the larger negative corrections.  In particular, the basal friction parameter is now computed directly at cell vertices (co-located with velocities) instead of being interpolated from cell centers.  We repeated the spin-ups with these changes.  In the revised spin-ups (with standard parameter settings, including power-law sliding), there are no longer any regions with maxed-out negative corrections of -2°.
- We still have large negative corrections (> 1° in magnitude for the PIGL calibration) for the Amundsen sector.  We hypothesize that the recently observed thermal forcing may be significantly greater than pre-industrial values, and thus not appropriate for a spin-up to steady state.  This would provide a physical justification for a negative correction.
- If the Amundsen sector is indeed warmer today than in pre-industrial times, and if the ESMs do not capture the recent warming, then our projections will systematically underestimate ASE retreat.  To address this possibility, we added sensitivity experiments in which the thermal forcing correction is set to zero for the forward projection.  With this correction, we see more rapid Thwaites retreat early in the projection, followed by collapse of the entire Thwaites basin for some parameter settings.  In other words, the negative corrections suggest that we should think more carefully about the spin-up procedure and the current departures from steady state.

Moving on to novelty 2, I just don't accept that these longer runs have value. The idea that ocean coupling, atmospheric coupling, and basal traction do not change on time scales of half a millennia is pure fantasy. Of course, it's also fantastical to imagine that one could do these couplings well, so the authors aren't to be faulted for not providing the couplings. Rather, I fault them for presenting these results at all. They simply are not credible.

We agree that on these long time scales, the ocean coupling, atmospheric coupling, and basal traction will certainly change, with uncertainties growing in time.  For this reason, we have emphasized that the long-term behavior should not be viewed as a prediction, but rather as an exploration of time scales for the important physical processes.  We find that marine ice sheet instability, unaccompanied by fast processes such as cliff collapse, unfolds on a time scale of several centuries.  In some runs, a threshold for massive collapse is reached after 2200, and the collapse is still underway at 2500.  Ending runs sooner (say, at 2200 or 2300) could mask the fact that an entire basin has become dynamically unstable, with irreversible retreat taking several hundred years to play out.

We revised the text in the Introduction as follows:

> Because of modeling and forcing uncertainties that grow larger on time scales beyond a century, our long-term results (e.g., beyond 2100), should not be viewed as predictions. Rather, our intent is to simulate retreat processes, such as MISI, that could unfold over several centuries, and to explore parameter space, taking advantage of the ISMIP6 framework to build on previous multi-century Antarctic simulations...

As sub-glacial hydrology models improve, and as atmosphere and ocean coupling with ice sheets is brought into ESMs, runs on time scales beyond 2100 could become more credible as predictions.

Finally, novelty 3. There is considerable value in simply documenting the complex work that goes into modeling on this scale, at this level of complexity. The paper does that, and does so in a way that is well presented and understandable. I think that the paper is quite honest in that there were no major components of the exercise that were left out, obfuscated, or overstated. A sensible argument for publication is that the cost of publication is low, and the quality of search is high. While the paper is highly specialized, the authors are outstanding professionals in the field and this is a useful report on what they have been doing. I only see one problem with this point of view - there were 15 groups participating in Seroussi et al. (in review). Are all of them to have companion papers detailing the way they handled ISMIP6? That seems really inefficient. I wonder if there couldn't be a single companion paper?

Thank you for commending this part of the paper.  Given the work required to explore these complexities for a single model, it would be infeasible for ISMIP6 participants to undertake such an analysis for all 15 groups in a single paper.  At the same time, it is common for modeling groups participating in CMIP activities (not just ISMIP6) to publish papers focused on their own model implementation and results.  This allows for a deeper exploration of the results than is possible group papers, as well as more information on the model used and how the experimental protocol was implemented.  Not all groups will publish a paper like ours, but they will likely explore other topics in more depth than we were able to do here.

Also, this manuscript goes beyond the ISMIP6 protocols by exploring (1) ice sheet response beyond 2100 and (2) a new melt parameterization (the "slope" scheme) that was not part of the ISMIP6 protocol.  Such explorations can be undertaken much more easily in single-model papers than in group papers.

In conclusion, I don't feel that the paper meets its own criteria for novelty because Jourdain is unpublished, because the need to alter the thermal forcing correction isn't explored in enough depth, and because the longer runs don't provide much value. The paper might provide novelty in that the details are useful, but that could be satisfied more efficiently with a companion paper that included more ISMIP6 participants.

Provided Jourdain is published, I could support publication of this manuscript with what I'd call 'minor' revisions, which I hope would include an expanded discussion of the thermal forcing correction, the need to push the model to steady state, and a consideration of thermal forcing

that produces refreezing. All of these expanded discussions should come at the expense of the long term runs.

These points are addressed above. Briefly, the Jourdain paper has been published; the revised manuscript explores the thermal forcing correction in detail for the Amundsen sector; the longer runs shed light on slow processes that could ultimately prove to be catastrophic; and an ISMIP6 group paper at this level of detail is not feasible.

That said, the option I favor most would be a companion paper that includes more ISMIP6 participants, and that lays out the details of modeling for the experiments in ISMIP6.

As stated above, this paper goes beyond the ISMIP6 protocol, and to our knowledge we are the only group to have investigated these specific aspects of the ocean forcing in detail.

I have a number of comments on individual passages of the paper as well.

Page 1 line 7, 'nudge' seems chatty and imprecise. "Constrain to observed values"? Page 1, line 13: say what the missing physics are.

We revised the text to read "During the spin-up, basal friction parameters and basin-scale thermal forcing corrections are adjusted to optimize agreement with the observed ice thickness."

We clarified the missing physics as follows: "simplified treatments of iceberg calving and basal sliding."

Page 2, line 34: say what is meant by 'committed'.

We added the following text: "By 'committed', we mean SLR due to ice-sheet retreat that has been set in motion and is likely irreversible."

Page 3, line 1: Be more consistent with the purpose of the paper, the abstract makes quantitative claims ("10 cm to nearly 2 m"), I'm not sure this sentence really fits with the approach in much of the rest of the paper.

This is a fair point. We revised this sentence as stated above: "Because of modeling and forcing uncertainties that grow larger on time scales beyond a century, our long-term results (e.g., beyond 2100), should not be viewed as predictions."

Page 2 Lines 5-9 it's as if the authors agree the exercise is pointless, but are rationalizing the CPU allotments. Just drop it.

Our intention here is to be honest and recognize the limitations of our approach. We think it is fair to say that "the timing and magnitude of simulated ice sheet retreat are imprecise", but we

can still "identify responses that are robust across simulations"--for example, the collapse of much of WAIS under projected increases in ocean thermal forcing.

Page 4, line 1-16, this is really a long road to just stating that the power-law sliding law is used. The 'nearly all' statement in line 17 is frustrating too. Where is the relation Coulomb? What's the point of having effective pressure in relations if one just assumes that it is ice overburden. This entire discussion could be collapsed to something much more concise.

We agree that for the initial submission, it was not helpful to discuss a Coulomb relation when the sliding law was a power law in practice.  In the revised submission, we have included simulations with p > 0.  So we have retained a description of the Schoof sliding law but emphasized that a power law is the default.

Page 4, line 30 - I'm often confused by the (unpublished) ISMIP6 protocols. Here you say CISM runs are based on the protocols. Page 7, line 5 says you differ from protocols with regard to dT_sector. That's a fairly big deviation from protocols. So...be consistent in saying what you've done.

The ISMIP6 protocols (Nowicki et al., 2020) have now been published.  We think it is important to state that our runs are based on ISMIP6 protocols--for example, the computation of basal melt rates using ISMIP6 thermal forcing data and melt parameterizations--but also to discuss how and why our runs are different.

Page 5, line 11 - are these mean annual, monthly, daily atmospheric forcings?

They are annual mean forcings.  We revised this to "For projections, ISMIP6 provides data sets of annual mean atmosphere and ocean forcing on a standard 8-km grid."

Page 5, line 35 - This is odd - I guess you force the thickness near the grounding line to be close to observations by altering, but then protect against the large negative values found for many sectors by zeroing out the contribution to melt? So, you have refreezing (I guess) if it aids thickness on initialization, but don't allow refreezing in forward runs? This really isn't clear. Is equation 4 used for spinup, and the modified version used for forward modeling?

The problem with using the parameterization unaltered is that it can lead to large, unphysical melt or freezing rates if an entire sector has negative thermal forcing.  The parameterization was not formulated with this possibility in mind.  We revised the text to make this more clear:  "For the simulations in this paper, the last term in Eq. (4) was modified to max(...).  This change was needed to prevent spurious melt and freezing in sectors where <TF> + dT_sector < 0.

Or, is the modified eq. 4 used for both? If so, then the negative values never have any impact, do they? Eqn 5 also eliminates refreezing.

Here is the big question: Aren't you introducing a big transient when (if) you jump between the refreezing allowed in the initialization and the melt only situation in the forward runs? Is this transient physical (I don't think it is)? Again, this is the novelty of the paper, but here is another case where it's not being treated with enough consideration.

There is no difference between the treatment for spin-up and forward runs, except that dT_sector is held fixed for forward runs.  So there is no such jump.

Page 8, line 22: OK - here we learn that this is not what was done as part of CISM2's contribution to Seroussi et al. (in review). This should have been scoped out in the introduction, as part of the novelty, with a clear statement of what the primary difference is.

We added some text in the Introduction to point out the novelty (new text in italics):

> This paper complements the study of Seroussi et al. (2020} by evaluating the Antarctic response to ocean forcing in a single model, the Community Ice Sheet Model (CISM; Lipscomb et al., 2019).  *A novel feature is the tuning of a single parameter in each of 16 sectors to correct the ocean forcing, thus optimizing model agreement with thickness observations near grounding lines.*

We did not think it was appropriate in the Introduction to explain how this method differs from our previous method.  We left that explanation in Section 3.1.

Page 9, line 17: I'm not aware of an analytical solution to the temperature equation that can accomodate advection of heat. Please provide more detail.

The temperature was initialized to an analytic solution at the start of the spin-up, but then evolves under advection.  The revised text reads, "For each spin-up, the ice sheet is initialized to present-day thickness using the BedMachineAntarctica data set (Morlighem et al., 2019). The internal ice temperature is initially set to an analytic vertical profile and then evolves freely under advection."

---

## Author Response (AR1)

**Reviewer comments and responses**
Nov. 14, 2020

To the Editor:

We have attached the reviewer comments with our responses. This is a revised version of the comments returned several weeks ago, with the addition of page and line references to the annotated manuscript.

We thank both reviewers for their careful analysis and constructive suggestions. Before proceeding with the responses, we thought it would be useful to summarize some substantial revisions of the original manuscript.

Reviewer 1 suggested that we look into the sensitivity of results to the stress-balance approximation (i.e., Blatter-Pattyn vs. DIVA) and the treatment of calving and shelf collapse. Reviewer 2 emphasized the uncertainties introduced by our spin-up procedure, in which a thermal forcing correction (sometimes large and negative) is applied in each basin. Both reviewers pointed out that with the assumption of effective pressure equal to overburden, we were not simulating Coulomb sliding. Also, it is well known that the grounding line is under-resolved at our default grid resolution of 4 km.

In response, we have expanded Section 4, "Results of projection experiments." In addition to our "standard" results using the DIVA solver at 4 km resolution with power-law sliding, we added a subsection (4.2) exploring the following sensitivities:
- Grid resolution; we added experiments at 8-km and 2-km resolution.
- Stress-balance approximation; we ran some experiments with the Blatter-Pattyn solver.
- Ice-shelf collapse; we applied the shelf-collapse maps provided by ISMIP6 and also tested a more extreme scenario in which all small shelves collapse.
- Basal friction law; we ran experiments with $p = 0.5$ and $p = 1$ in the expression for effective pressure (where $p = 0$ corresponds to power-law sliding).
- Threshold behavior in the Amundsen sector.

This section includes many new figures.

In response to Reviewer 2, we also made some numerical and parameter changes that had the effect, in some sectors, of stabilizing grounding lines and reducing negative temperature corrections. We repeated all the standard experiments with these changes included, in addition to the new sensitivity experiments.

For the most part, the sensitivities are modest, and our main conclusions have not changed. We did, however, find some configurations for which the Amundsen basin collapses, in contrast to the standard configuration in which Amundsen retreat is modest. Sect. 4.2.5 explores this threshold behavior.

For forward runs, we changed the nominal date for the spin-up from 1995 to 1950. This allowed us to incorporate ESM anomalies going as far back as 1950. Each forward run is now 550 years, from 1950 to 2500.

The title now begins with "ISMIP6-based projections" instead of "ISMIP6 projections", reflecting that our experiments diverge in several ways from the ISMIP6 protocols.

We removed the Appendix, which described CISM's grounding-line parameterization in detail. This material is now included in a manuscript by Leguy et al. (2020), recently submitted to *The Cryosphere*. We expect that this manuscript will be published soon in *TCD*. We can provide a copy on request.

Our detailed responses are below, with reviewer comments in black font and responses in blue. We have attached two versions of the revised manuscript: one that tracks additions and deletions relative to the original manuscript (with additions in blue), and one clean version. Page and line numbers in our responses to the reviewers refer to the version with tracked changes.

Thank you for your patience and consideration.

**Reviewer #1** (Thomas Zwinger):

**1  General Impression**

This is a manuscript that complements the currently reviewed ISMIP6 intercomparison on the Antarctic ice sheet. Beyond the contribution that was already accounted for in the paper by Seroussi et al. (Seroussi et al., 2020), the authors show new ways of spin-up and present computations beyond the time-frame of ISMIP 6.

The text describes in detail the methods used and the assumptions applied within the contribution from the ice-sheet model CISM to this inter-comparison. From my point of view, the paper to a large extent analyses the impacts of assumptions and approximations on the results, except for two open points which I will raise in the section below. The article is well written and has a clear line to follow. I got the impression of a few inaccuracies in the notation, which in my view is easily to be fixed. I also have some suggestions on how to improve or augment the figures.

Modelling techniques presented here are mainly variations of existing approaches, but the size and the context of the applications make for the scientific novelty of this paper. I also see the value in documenting the techniques leading to these results, which of course cannot be presented in such detail in an inter-comparison paper. In that sense, I see it as a useful contribution to the literature, also to demonstrate what efforts have to be taken in order to realise such an implementation.

I mainly placed a few technical suggestions to (according to my subjective impression) improve the readability of the text, in particular for readers not being that familiar with ISMIP6. Besides this, the only more substantial criticism I would place is the in my view not complete analysis of the impact on the results of some of the assumptions made, in particular the over-simplification of calving and the impact of the (actually not clearly revealed) approximation to the Stokes equation and the resolution applied in the ice-sheet model. If these points are addressed, I would recommend this manuscript for publication in TC.

**2 Main points**

As mentioned above, I would see it necessary to have two in my view not completely clear issues to be elaborated:

1. The influence of the in my view oversimplified calving law on the results, in particular its impact on grounding line migration and retreat

We agree that the no-advance calving law is very simple and that ice-shelf collapse (not explored in our original manuscript) could significantly affect future grounding-line retreat. We are unable to present results with a physically based calving law, because we were not able to obtain a satisfactory Antarctic spin-up with such a law. However, we added two calving sensitivity tests, as described below.

2. The choice of the approximation to the Stokes equation and its consequences on the accuracy and sensitivity of the results

Please see below.

Concerning the first point, i.e. calving, you explain (page 4, line 20): *Instead, we use a no-advance calving mask, removing all ice that flows beyond the observed calving front. The calving front can retreat where there is more surface and basal melting than advective inflow, but more often the calving front remains in place.* Here you might mention the point in time of the observation of the prescribed front (I presume it somehow refers to your initial state in 1995). In the Conclusions, you mention it in just one sentence: *In terms of ice sheet physics, these simulations do not include hydrofracture or calving-front retreat, and thus are missing positive feedbacks associated with reduced buttressing of grounded ice by ice shelves (Sun et al., in review).* In my opinion, the reader would benefit from some clarification on how much impact neglecting the possibility of ice-shelf collapse (as it would be provided by ISMIP 6 forcing) on the resulting sea level has. In my view, the current approach significantly might lead to an underestimation of sea-level rise, in particular at times beyond the year 2050. One ad- ditional experiment could be to, for instance, just remove the shelf in front of Thwaites and see what short-time impact this has for this region – maybe in combination with the upper end combination of AOGM and ocean forcing. Or to apply the shelf-collapse scenario from ISMIP 6. If this is for some technical reason not possible at this stage, please mention that in the paper.

We added two calving sensitivity tests:

1. For each melt parameterization and ESM, apply the ice-shelf collapse masks provided by ISMIP6 for the optional shelf-collapse scenarios (in combination with the ESM thermal forcing anomalies). This option was not available in CISM at the time of the original submission, but is now supported.
2. For each melt parameterization and ESM, force the abrupt collapse of all small ice shelves in year 2100. Here, "small" is defined as all ice shelves outside of sectors 2, 7 and 14, which correspond to the Amery, Ross, and Ronne-Filchner Shelves, respectively. We took this approach because (1) no ESM in our ensemble shows significant collapse in these three basins before 2100, and (2) the effects of large ice-shelf collapse have

recently been explored by Sun et al. (2020). Thwaites is among the shelves removed, so this test incorporates the suggestion above.

We found that the effects of shelf collapse are not negligible, but are smaller than the mass loss associated with ocean thermal forcing anomalies for most ESM and melt configurations. ISMIP6-prescribed shelf collapse typically adds 5 to 15 cm to the SLR due to ocean thermal forcing anomalies alone. The collapse of all small shelves has a larger effect, contributing additional SLR of 15 to 25 cm. Removal of the Thwaites shelf does not lead to much additional retreat in the Amundsen Sea basin, suggesting that the current buttressing effects are small.

Results of these shelf-collapse experiments are described in Sect. 4.2.3.

On the second point, model approximation, you present (Page 3, line 30) that the CISM code includes options for the following approximations to the Stokes equations: *SSA, L1L2, LMLa* – mainly following Hindmarsh' nomenclature (Hindmarsh, 2004). On page 4, line 1, you write: *The simulations for this paper use the depth-integrated solver, which gives a good balance between accuracy and efficiency for continental-scale simulations (Lipscomb et al., 2019).* I consider SSA as well as L1L2 as depth integrated (the latter of course with some corrections from a vertical profile). Could you please clearly identify which of the previously listed models you deploy in your study? I am further confused how a depth integrated model can use 5 vertical layers (as described in the paragraph before), but assume that you use this for some aspects different from depth-integrated parts, such as temperature evolution or velocity corrections. Is this vertical resolution sufficient to resolve the physics?

We used CISM's DIVA solver, which falls in the L1L2 category of Stokes approximations. The solver is two-dimensional in the sense that it solves a global 2D elliptic equation for the mean values of u and v in each column. Given the mean values, it is possible to compute a full velocity profile following Goldberg (2011). The full 3D velocity profile is then used to advect thickness and temperature.

For Antarctica, we found five vertical levels to be sufficient; changes are small when we add more levels. More levels are preferable if the velocity is dominated by vertical shear, but in our regions of interest (near grounding lines), the flow is dominated by basal sliding.

We added some text in the Introduction (p. 2, ll. 7ff) to make these points more clear. The revised text reads:

> CISM runs on a structured rectangular grid with a terrain-following vertical coordinate. Most simulations in this paper were run on a 4-km grid, as for the CISM contributions to ISMIP6 projections (Seroussi et al., 2020). At 4-km resolution, grounding lines are under-resolved, but this is the finest resolution that permits a large suite of whole-Antarctic simulations at reasonable computational cost. To test sensitivity to grid resolution, we repeated some experiments on 8 km and 2 km grids (Sect. 4.2.1). All simulations were run with 5 vertical levels. Increasing the number of vertical levels would not substantially change the results, especially in regions dominated by basal sliding rather than vertical shear (i.e., the regions critical for grounding-line retreat and SLR). Scalars (e.g., ice thickness H and temperature T) are located at grid cell centers,

with horizontal velocity $\mathbf{u} = (u,v)$ computed at vertices. The dynamical core has parallel solvers for a hierarchy of approximations of the Stokes ice-flow equations, including the shallow-shelf approximation (MacAyeal, 1989), a depth-integrated higher-order approximation (Goldberg, 2011), and the 3D Blatter-Pattyn (BP) higher-order approximation (Blatter, 1995; Pattyn, 2003}. The latter two approximations are classified as L1L2 and LMLa, respectively, in the terminology of Hindmarsh (2004).  The simulations for this paper use the depth-integrated solver, known as DIVA (Depth-Integrated Viscosity Approximation), which solves a 2D elliptic equation for the mean velocity, followed by a vertical integration at each vertex to obtain the full 3D velocity. This solver gives a good balance between accuracy and efficiency in idealized settings and whole-ice-sheet continental-scale simulations (Leguy et al., 2020; Lipscomb et al., 2019). In Sect. 4.2.2, we compare DIVA to the more expensive BP solver in selected runs.

Can you include some lines on how much you estimate the influence of the choice of the approximation and the chosen resolutions on the results of your study. For instance, would one expect to reach the same conclusions if using a LMLa model or perhaps the current model with a finer resolution?

This is a good question that motivated some sensitivity experiments, described in the new Sects. 4.2.1 and 4.2.2. CISM's Blatter-Pattyn (LMLa) solver is more than an order of magnitude more expensive than DIVA.  Within our computing budget, we were unable to do long spin-ups with the BP solver at 4 km.  However, we were able to run four spin-ups (nonlocal and nonlocal-slope melt parameterizations, with MeanAnt and PIGL calibrations) using the BP solver at 8 km, and then carried out projection experiments using this solver.  We compared the results to those obtained with the DIVA solver with the same resolution and forcing.

Differences between DIVA and BP are small for the experiments we ran.  Depending on the melt scheme and ESM, the difference in cumulative SLR between BP and DIVA varies from about -50 mm to +20 mm (Fig. 15).  We have run other DIVA–BP comparisons at 4-km resolution (e.g., for Greenland in Lipscomb et al., 2019) or finer (in idealized tests such as MISMIP3d and MISMIP+, as described by Leguy in a paper to be submitted).  We do not think that replacing DIVA with BP would change our main conclusions.

To explore the effect of finer resolution, we ran four additional spin-ups (one for each melt parameterization/calibration) at 2-km resolution. We then ran the usual suite of ESM experiments. To stay within our computing budget, we ran the spin-ups for 10,000 instead of 20,000 model years.  The drift in the control run is slightly larger than for the 4-km runs, but still small compared to the sea-level signal.  Generally, higher resolution leads to greater ice sheet retreat.  Compared to the 4-km runs, the total SLR in 2-km runs is up to 15% larger, or ~150 cm for the high-end scenarios (Fig. 13).  These differences are not large enough to change the main conclusions.

**3  Suggested corrections, additions and typos**

Please, find suggestions for corrections, changes and typos in the order of their occurrence in the text:

Page 1, Author list: Typo: Nicolos → Nicolas

We corrected the typo.

Page 2, line 28: *To study long-term ice sheet evolution, we extend the simulations to 2500, . . .*
Minor issue, but for better readability, please mention that this is a date (it could be iterations or any other time unit).

We revised the text as suggested (p. 3, l. 2).

Page 4, line 6, 9 and 11: Again, minor issue, but in order to stay consistent with the nomenclature in equation (1), I would use the subscript for bedrock in all the following occurrences of the basal velocity, i.e., $u \rightarrow u_b$.

We made the suggested change.

Page 4, line 13: Can you elaborate on the choice of exactly this value for Cc? Is this representing some tested optimum? Or (see next question) does it not really matter what to put there?

This parameter is not well constrained by data, and we are not aware of formal tests that established it as an optimum. The value we chose is the recommended value in Asay-Davis et al. (2016), used by MISMIP+ participants. We revised the text (p. 5, l. 3) to cite Asay-Davis et al. (2016) instead of Pimentel et al. (2010), who used a slightly different value.

Near grounding lines, the value of $C_c$ does matter. So does the value of p, since the Coulomb friction law contains the combination ($C_c * N$), where *N(p)* decreases with increasing p. Thus, it is somewhat arbitrary whether to vary $C_c$ or p in a sensitivity study. We held $C_c$ fixed but varied p (with values of 0.5 and 1.0) in several sensitivity experiments.

Page 4, line 17: *Setting* N *to overburden pressure implies power-law behavior in nearly all of the ice sheet.* To my understanding, this means that power-law.

Sliding then most likely prevails down to the grounding line - or am I missing something? If this is the case, I would question the benefit of explaining sliding law (1) and the Coulomb sliding parameter therein in such a detail. Being aware that in lack of a detailed representation of pinning points, any sliding law that reaches zero resistance at the grounding line inherently causes issues (confirmed also by our experience with Full-Stokes models at similar resolution), I do not criticise the approach as such, but would welcome to include some information on how high the friction parameters at the grounding line actually are (or have to be in order to be stable) and how – also by keeping them constant relative to a moving grounding line – that impacts the prognostic runs.

This is correct. Using the Schoof law with $p = 0$, and thus setting $N$ to overburden pressure, is very similar to using a power law. Our standard experiments now use a power law, which is another CISM option in addition to the Schoof law.

In response to this comment, and another comment from Reviewer 2, we ran a set of sensitivity experiments using the Schoof law (Eq. 1) with p = 0.5 and p = 1. We revised the text in Section

2.1 accordingly, emphasizing that the power law is the default.  Running CISM with $p = 1$ (hence zero resistance at the grounding line) slows down the code significantly, but the velocity solver still converges.  We think the new results are of interest.  In particular, the Thwaites basin is more prone to collapse with $p = 1$ than with $p = 0$, as discussed in the new Sect. 4.2.4.

Near the top of p. 8 in the original text (p. 9, l. 17 in the revised text), we state the maximum value of $C\_p$, which is high enough to effectively turn off sliding. In basins where the grounding line significantly retreats, we do introduce errors by using the tuned $C\_p$ values, either because these values are the wrong magnitude or because they should be changing in time. It is hard to estimate the size of the error.  This is a subject for a future study, perhaps with a physically based model of subglacial hydrology.

The paragraph on biases and limitations of $C\_p$ inversion (p. 9, ll. 21ff) now reads:

> This method works well at keeping most of the grounded ice near the observed thickness. Also, since $C\_p$ is independent of the ice thermal state, we remove low-frequency oscillations associated with slow changes in basal temperature, resulting in a better-defined steady state.  In forward runs, however, $C\_p(x,y)$ is held fixed and cannot evolve in response to changes in basal temperature or hydrology. As a result, we can have unphysical basal velocities when the ice dynamics differs from the spun-up state. Also, the tuning of $C\_p$ can compensate for other errors. For example, if the topographic data set is missing pinning points near the grounding line, the ice will be biased thin, and $C\_p$ can be driven to high values to make up for the lack of buttressing.

Page 5, line 5: *(The simulations described here use only the ocean forcing.)* I do not understand the meaning of this sentence. Does it mean, you do not apply any atmospheric, but only ocean forcing? Or does this only apply in connection to CMIP? Please, elaborate. On a side-note, in my personal opinion, I find it strange to put whole separate sentences in brackets. I would write it without. I mention this, as it occurs a few times in the text.

We apply the same atmospheric forcing (SMB from RACMO) as in the spin-up, but without adding SMB anomalies from the ESMs.  We revised the text as follows (p. 6, ll. 3ff):

> The ISMIP6 projection experiments are run with standalone ISMs, forced by time-varying, annual-mean atmosphere and ocean fields derived from the output of CMIP5 and CMIP6 ESMs. In our study, the projection experiments use ocean forcing that evolves during the simulation, but the atmospheric forcing (specifically, the SMB) surface is held to the values used when spinning up the model.

We also removed parentheses around sentences in several places.

Page 8, line 4: *Eq. 6 is based on the equation for a critically damped harmonic oscillator, where the first term in brackets nudges H toward Hobs, and the second term damps the nudging to prevent overshoots. (The damping is not exactly critical, however, because* dCp/dt *is not exactly proportional to* d2H/dt2.*)* I am slightly confused by this statement, since from my point of view, if dCp/dt would be somehow proportional to d2H/dt2, the Cp on the right-hand-side would introduce some functional relation to H and I would not immediately see the equation of a

damped harmonic oscillator recovered by (6). Could you clarify or insert a reference where this is explained in detail in order to help the reader to follow that up. Minor issue: *Eq. 6* → Eq. (6).

We removed the language about the critically damped harmonic oscillator, because the analogy with classical mechanics is perhaps not close enough to be helpful. The text (p. 9, ll. 15ff) now simply describes the role of the two terms.

Throughout the text, we added parentheses around equation numbers.

Page 8, line 6: *We hold* $C_p$ *within a range between* $10^2$ *and* $10^5$ *Pa m−1/3 y−1, since smaller values can lead to numerical instability, and larger values do not significantly lower the sliding speed.* Can you explain the nature of the instabilities? I wonder if this is somehow linked to the fact that you had to fix the effective pressure to the overburden pressure in the sliding law.

The instabilities appear in the form of CFL violations, i.e., unrealistically fast velocities that can crash the model. We do not think this instability is related to fixing $N$ to overburden pressure; rather, it comes from having close to zero friction in some lightly grounded cells near the grounding line.

We revised the text as follows (p. 9, ll. 17ff):

> We hold C_p within a range between 10^2 and  10^5 Pa m^{-1/3} y^{-1}.  Smaller values can lead to excessive sliding speeds when the basal friction approaches zero.  With C_p at its maximum value, basal sliding is close to zero, and there is little benefit in raising C_p further.

Page 9, line 7 and Eq. (8): Based on results shown in Fig. 5, I conclude that the symbol $\delta T$ stands for the nudged values of the ISMIP 6 protocol given $\delta T_{sector}$. Simply, the missing subscript might also lead to the conclusion that it is the difference to this reference value, i.e., the correction to the correction. I would consistently add the subscript *sector* to $\delta T$ .

We revised the notation as suggested.

Page 9, line 5: *After some experimentation, we set* $m_T = 10$ *m y−1 K−1 and* $\tau_m = 10$ *y.* Do these values link in some way to physics? If not, please explain the procedure behind the term *some experimentation*.

Briefly, we adjusted the values so as to achieve a relatively fast relaxation to equilibrium without oscillations. This was a process of trial and error, with values linked only loosely to physics.

Rather than lengthen the text with an explanation, we decided to remove the phrase "after some experimentation" and simply state the values (p. 10, ll. 16-17).

Page 9, line 6: . . . *, with modest values of* $\delta T$ *(≈ 1 K or less). Modest* with respect to what? I guess to melt-rates.

We were thinking "modest with respect to the background thermal forcing", which typically is 1° or more in regions with significant melt. But since "modest" is a relative term, we rewrote the

text (p. 10, l. 19) to state simply that |δT_sector| < 1°C in most basins. The exceptions are the Aurora and Amundsen sectors, which are discussed later (p. 12, ll. 14ff).

Page 9, line 9: Typo: To me it appears that there is an orphan *A* between two sentences.

We fixed the typo.

Page 9, line 24: *Some biases can likely be attributed to errors in ocean thermal forcing (which is treated simply by the basin-scale melt parameterizations) and seafloor topography (e.g., an absence of pinning points, resulting in grounding- line retreat that can be compensated by spurious ocean cooling).* Could the errors by absence of pinning points also link to the relatively coarse resolution of the applied ice flow-model and the under-resolution of bedrock data in some regions? If so, please mention it.

We agree. We added the following sentence (p. 11, ll. 20-21): "Pinning points could either be missing in the high-resolution (0.5 km) topographic data set (Morlighem et al., 2019), or smoothed away when interpolating to the coarser CISM grid." We hoped that pinning points would be better resolved on the 2-km CISM grid, but this was not generally the case, suggesting that at least some pinning points are missing (or located at too great a depth) in the BedMachine data.

Page 9, line 30 and page 10, Fig. 2: *Thwaites Glacier is too thin and Pine Island Glacier is too thick;* For me, the contours in Fig. 2 make it virtually impossible to spot thickness differences at the grounding lines or in smaller shelves. Perhaps, as you mention the Amundsen sea sector here in the text, some zoom-in Figure for that particular regions (just like Fig. 4 provides for speed) would be good. Neither is it possible to spot a clear outline of the Antarctic ice sheet. My suggestion to improve this, would be to have a neutral (e.g. white or blue) colour outside the ice-sheet/shelf area and only plot the grounding line. I, personally, have difficulties with the low contrast between the different shades of yellow and green in this figure. To me there seem to be jumps in ice thickness difference across many places at grounding lines (both with negative and positive signs). For instance, very prominently in the left panel for Amery. I would conclude – assuming that any evolved geometry is at least C0 continuous across the grounding line – that also the difference has to be continuous. If you please could explain where this discrepancy arises from or else point me to my misconception of this figure.

We agree that the color schemes for this and other figures was poorly chosen. In the revised Fig. 2, we use a different color scheme with white for the Southern Ocean (outside the ice sheet boundary), pale yellow for values near zero within the ice sheet, blue for positive, and red for negative. The differences are, indeed, continuous; the apparent jumps were a result of the color scheme.

We also added a thickness-difference figure (Fig. 3) that zooms in on the Amundsen sector.

Page 11, Fig. 3: In this figure the colour scale actually works for me. I conclude that in Fig. 2 it is not about the colours themselves but the representation. Nevertheless, the black lines in Fig. 3 are not really visible, but in my view on that scale (different in Fig. 4) anyhow obsolete concerning the clear velocity jump between land-based ice and shelves.

We found another color scale we preferred for this figure (now Fig. 4), with pale yellow for slow speeds, red for intermediate, and purple/blue for fast. We omitted the black lines, given that shelves are obvious from the velocity jump.

Page 11, Fig. 4: In large parts over Thwaites, the black grounding lines are really hard to spot over the dark green texture – perhaps a different colour (some grey?) would help. I mention that, as this is important since you refer to discrepancies of grounding line positions in this region in the text.

We revised the color scheme as described above for Fig. 4, and we changed the grounding lines to light gray to make them easier to see.

Page 12, Fig 5: On my screen the underlying Antarctic ice-sheet shape is extremely difficult to spot. Please, enhance this. To me it comes clear that you are representing the nudged total values of δTsector (see earlier comment on Eq. (8)). Would it add value to the graph to include the calibrated values ISMIP 6 values for comparison?

Yes, this is the nudged value of δT_sector. In the revised figure (now Fig. 6, p. 15), we enhanced the Antarctic background shape and added another plot to show the calibrated ISMIP6 values for comparison. It was cleaner to make two plots than to put all the numbers in one plot.

Page 14, Fig. 6: Since there are finite values of the ocean thermal forcing underneath land-ice, can you please explain their meanings?

Ocean thermal forcing is masked out for fully grounded cells, but is available in the forcing data in case these cells should later become ungrounded. We added text in the caption (now Fig. 9, p. 19) to clarify that basal melting due to ocean thermal forcing is applied only in floating cells.

Page 16, Fig. 8: Please, within the caption of this figure, explain the red lines in the pictures, in particular the enclosed one within the Ross ice-shelf for *HadGem2/nonlocal MeanAnt* scenario. I guess they mark shelf areas, and I also guess they represent data for least/most sensitive melt scenario with the same red colour. If so, please distinguish them with lines of different colours or different pattern. Also, please, indicate the meaning of the black line – I guess it is the initial grounding line position.

Indeed, the black and red lines (marking initial and final grounding line positions) were not explained in the caption and were hard to interpret. The revised figure (Fig.12, l. 22) uses an improved color scale, similar to Fig. 2 as described above. We decided to show only the final shelf boundaries (using black contours), since a second set of contours is hard to read. Regions of grounding-line retreat are easy to identify, since they have the largest decrease in ice thickness.

Page 17, line 3: Typo: orphan dot

We fixed the typo.

Page 18, line 1: *Ice sheet retreat in these simulations is modest for the next several decades.* In my view, this statement links to the calving simplification mentioned in the main point. You

might address the following question: Could the initial slow response be linked to the fact that you are not able to fully capture effects of potential ice-shelf collapse? To some extent you mention this a few lines below (Page 18, line 17): *In terms of ice sheet physics, these simulations do not include hydrofracture or calving-front retreat, and thus are missing positive feedbacks associated with reduced buttressing of grounded ice by ice shelves (Sun et al., in review).* For me the connection with the timings of SLR appears to be important and is missing.

We addressed this question with the shelf-collapse experiments discussed above (Sect. 4.2.3). Since the ISMIP6 shelf-collapse data sets have most collapse happening late in this century, this collapse cannot accelerate ice loss earlier in the century. The other set of calving experiments, in which all small shelves collapse in 2100, does trigger an immediate speed-up in ice loss, but this is an extreme scenario, unlikely to occur in the next few decades. Thus, we do not think the lack of shelf collapse is the primary reason for the initial slow response.

Page 18, line 21: *At 4-km grid resolution, processes such as grounding-line retreat may be under-resolved.* I think it is fair to say that at 4 km resolution grounding line mechanics is underresolved, in particular if combined with the choice of sliding law in this study ().

We agree. The text now reads (p. 38, l. 18), "At 4-km grid resolution, processes near the grounding line are under-resolved."

This concern is addressed in part by the sensitivity runs at 2-km resolution, showing increased ice loss at higher resolution but not qualitatively different behavior. In MISMIP3d and MISMIP+ experiments, as described in the recently submitted manuscript by Leguy et al. (2020), we have explored resolution dependence in detail. We have generally found that numerical errors from running the model behavior at 2–4 km, as compared to 0.5–1 km, are small compared to other sources of uncertainty (e.g., melt rates and basal sliding parameters).

Page 18, line 25: *These uncertainties suggest several lines of research to further improve ice-sheet and sea-level projections:* What I am missing in this list here

Is the aspect of changing the accuracy of the approximation applied in the ISM - in particular as I understood that CISM would have the possibility to solve LMLa (Blatter-Pattyn) equations. Would results improve if deploying higher order ISM approximations? Are they feasible given the time-intensive spinup process?

As explained above, it is not feasible to run a large suite of experiments using the BP solver at a resolution of 4 km, or even 8 km, because of the greater cost. But the small DIVA–BP differences at 8-km resolution (Sect. 4.2.2) support the choice of DIVA and suggest that depth-averaging in the Stokes approximation is a relatively small source of uncertainty.

**Reviewer #2:**

This paper provides a detailed report on a high-profile Earth systems model's (Community Earth Systems Model, CESM2) contribution to the Ice Sheet Model Intercomparison Project (ISMIP6) for the Coupled Model Intercomparison Project - Phase 6 (CMIP6). Being an ice-sheet model intercomparison, the contribution from CESM2 is provided by the Community Ice Sheet Model

(CISM2), a single component of CESM2. Atmospheric and oceanic forcings for the ice-sheet model are uncoupled, and provided by output of six atmosphere ocean general circulation models (AOCMs), including CESM2 (when run, called CCSM4.0). The results of the Antarctic portion of ISMIP6 is to be reported in Seroussi et al. (in review), and include results from CISM2.

The novelty of this paper, over Seroussi et al. (in review) is that it:

1. Concentrates on and expands consideration of the sub-shelf melting parameterization, which is the aspect of modeling that Seroussi et al. (in review) report as being *"The largest sources of uncertainty come from the ocean-induced melt rates, the calibration of these melt rates based on oceanic conditions taken outside of ice shelf cavities and the ice sheet dynamic response to these oceanic changes."*

2. Extends simulation to 2500 by maintaining forcing after 2100 based on late 21st century forcing from high emissions scenarios. These longer runs are intended to

   - Identify the regions of Antarctica most vulnerable to retreat

   - Determine the long-term ocean-forced sea-level rise from Antarctica.

3. Unstated by the authors, but important to me is that the paper lays out a number of heuristics and other modeling approaches that were used to bring CISM2 to the level of being a credible participant in ISMIP6. Such information is valuable to other modeling groups. While some of this information does appear in Seroussi et al. (in review), it is in the form of a single, terse paragraph.

In terms of novelty 1, I agree that the ocean-induced melt rates are critical to our projections of Antarctica's response to climate change. However, I see that the entire scheme is based on an unpublished work, Jourdain et al. (in review). I recognize that publication of such a large collaborative project as ISMIP6 is extremely complicated and dependencies such as this do arise. On the other hand, I hope the authors can appreciate that my role as a reviewer is to verify the assumptions made are sound. I can not do this when they are based on unpublished techniques. I am distressed by line 18 of section 2.2, which states that the "*climatology was interpolated to fill gaps and* **extrapolated** *into ice-shelf cavities"* ( my emphasis on extrapolated). Extrapolation is dangerous business and I'm not willing to accept that everything is OK with that until it's been through peer-review. My reservations are even greater when considering this paper, which frames much of its novelty around the topic of sub-ice shelf melting.

The paper by Jourdain et al. (2020) has since been published in *The Cryosphere*. We agree that extrapolation introduces large uncertainties, but we think it is a defensible procedure in the absence of high-resolution ocean model simulations. Since our paper is largely an exploration of the forcing provided by ISMIP6, we thought it was beyond our scope to explore other ways of estimating thermal forcing in ice shelf cavities. More realistic approaches, including sub-cavity ocean simulations, will be explored in the upcoming MISOMIP2 project, co-led by Nicolas Jourdain.

We added some text in Section 2.2 (p. 6, ll. 22ff) on the reasons for using an extrapolation (new text in italics):

> The result is a time-varying 3D product that can be vertically interpolated to give the thermal forcing at the base of a dynamic ice shelf at any time and location in the Antarctic domain. *The extrapolation procedure does not directly resolve ocean circulation in cavities, but was seen as a necessary simplification in the absence of output from high-resolution regional ocean models.*

Continuing with my assessment of novelty 1, I also see that the authors have deviated from the prescribed ocean temperature anomalies per sector, $\delta T\_sector$, in order to assure the thickness in the region of the grounding line is similar to modern thicknesses. OK, but I'm not sure this is a tunable parameter. Jourdain apparently assured that the values of $\delta T\_sector$ agreed with the estimated mean melt in each sector (no reference provided). No such assurances arise in the author's method, making $\delta T\_sector$ an unphysical parameter that is accounting for other model shortcomings. Page 12-13 acknowledge what has been done, and suggest that the lack of buttressing may be responsible for the mismatch. I'd liked to have seen more here. If the paper is valuable because it is taking a harder look at ocean-induced melt rates, and their coupling to ice dynamics, then it should investigate why unphysical parameterizations are needed for spin-up. Lack of buttressing? Maybe. What about using spin up targets that are in steady state? Plainly the Pine Island/Thwaites region isn't in steady state, why initialize the model as if it is? I believe these $\delta T\_sector$ are the richest material coming from these simulations, but they aren't being investigated with enough depth. For example, consider figure 5. Why are many regions, for many simulations in a state that is so negative? The ice near the grounding line is too thin? Surely there is more at play. Also, why are some sectors reaching their constraints in terms of temperature forcing (-2.00). It's as if the model is trying to hold back a strong departure from the steady state. Why take such measures? It seems contrived.

We think it is justified to treat $\delta T\_sector$ as a tunable parameter, given the sparseness of temperature data in ice-shelf cavities. Jourdain et al. (2020) wrote that this parameter accounts "for biases in observational products, ocean property changes from the continental shelf to the ice shelf base (not accounted for in the aforementioned extrapolation), tidal effects and other missing physics." They tuned this parameter to match sector-averaged melt rates, which for most sectors are very uncertain. Our tuning is done with a different target: observed ice thickness instead of melt rates. In most cases, our corrections are of the same order of magnitude as the corrections made by Jourdain et al. (2020); the exceptions are the Aurora and Amundsen sectors. We agree that the Amundsen Sea sector deserves closer consideration; see below.

The negative values in Fig. 5 (now Fig. 6, p. 15) can be explained by a combination of observational error, missing pinning points (compensated by reduced melting), excessive melt rates for the given thermal forcing, or model error (e.g., inadequate spatial resolution). All factors could be at play. For example, the negative corrections are larger for the PIGL calibration than for MeanAnt calibration, which could be taken as support for MeanAnt (or some intermediate calibration). Also, the melt parameterizations are missing ocean feedbacks that could reduce melt rates. The system is too complex to isolate one factor.

For the revised manuscript, we addressed these critiques as follows:

- We made some numerical changes in CISM that reduced the magnitude of some of the larger negative corrections. In particular, the basal friction parameter is now computed directly at cell vertices (co-located with velocities) instead of being interpolated from cell centers. We repeated the spin-ups with these changes. In the revised spin-ups (with standard parameter settings, including power-law basal friction), there are no longer any regions with maxed-out negative corrections of -2°.
- We still have large negative corrections (> 1° in magnitude for the PIGL calibration) for the Amundsen sector. We hypothesize that the climatological thermal forcing (based on 1995-2018 observations) may be significantly greater than pre-industrial values, and thus not appropriate for a spin-up to steady state. This would provide a physical justification for a negative correction.
- If the Amundsen sector is indeed warmer today than in pre-industrial times, and if the ESMs do not capture the recent warming, then our projections will systematically underestimate Amundsen retreat. To address this possibility, we added sensitivity experiments (Sect. 4.2.5) in which the thermal forcing correction is set to zero for the forward projection. With this correction, we see collapse of the Thwaites and Pine Island basins for some parameter settings (Figs. 21-23).

Moving on to novelty 2, I just don't accept that these longer runs have value. The idea that ocean coupling, atmospheric coupling, and basal traction do not change on time scales of half a millennia is pure fantasy. Of course, it's also fantastical to imagine that one could do these couplings well, so the authors aren't to be faulted for not providing the couplings. Rather, I fault them for presenting these results at all. They simply are not credible.

We agree that on these long time scales, the ocean coupling, atmospheric coupling, and basal traction will certainly change, with uncertainties growing in time. For this reason, we have emphasized that the long-term behavior should not be viewed as a prediction, but rather as an exploration of time scales for the important physical processes. We find that marine ice sheet instability, unaccompanied by fast processes such as cliff collapse, unfolds on a time scale of several centuries. In some runs, a threshold for massive collapse is reached after 2200, and the collapse is ongoing at 2500. Ending runs sooner (say, at 2200 or 2300) could mask the fact that an entire basin has become dynamically unstable, with irreversible retreat taking several centuries to play out.

We revised the text in the Introduction as follows (p. 3, ll. 10-14):

> Because of modeling and forcing uncertainties that grow larger on time scales beyond a century, our long-term results (e.g., beyond 2100), should not be viewed as predictions. Rather, we aim to simulate retreat processes, including MISI, that could unfold over several centuries, and to explore parameter space, using the ISMIP6 framework to build on previous multi-century Antarctic simulations.

As sub-glacial hydrology models improve, and as atmosphere and ocean coupling with ice sheets is brought into ESMs, runs on time scales beyond 2100 could become more credible as predictions.

Finally, novelty 3. There is considerable value in simply documenting the complex work that goes into modeling on this scale, at this level of complexity. The paper does that, and does so in a way that is well presented and understandable. I think that the paper is quite honest in that there were no major components of the exercise that were left out, obfuscated, or overstated. A sensible argument for publication is that the cost of publication is low, and the quality of search is high. While the paper is highly specialized, the authors are outstanding professionals in the field and this is a useful report on what they have been doing. I only see one problem with this point of view - there were 15 groups participating in Seroussi et al. (in review). Are all of them to have companion papers detailing the way they handled ISMIP6? That seems really inefficient. I wonder if there couldn't be a single companion paper?

Thank you for commending this part of the paper. Given the work required to explore these complexities for a single model, it would be infeasible for ISMIP6 participants to undertake such an analysis for all 15 groups in a single paper. At the same time, it is common for modeling groups participating in CMIP activities (not just ISMIP6) to publish papers focused on their own model implementation and results. This allows for a deeper exploration of the results than is possible with group papers, as well as more information on the model used and how the experimental protocol was implemented. Not all groups will publish a paper like ours, but they will explore other topics in more depth than we were able to do here.

Also, this manuscript goes beyond the ISMIP6 protocols by exploring (1) ice sheet response beyond 2100 and (2) a new melt parameterization (the nonlocal-slope scheme) that was not part of the ISMIP6 protocol. Such explorations can be done much more easily in single-model papers than in group papers.

In conclusion, I don't feel that the paper meets its own criteria for novelty because Jourdain is unpublished, because the need to alter the thermal forcing correction isn't explored in enough depth, and because the longer runs don't provide much value. The paper might provide novelty in that the details are useful, but that could be satisfied more efficiently with a companion paper that included more ISMIP6 participants.

Provided Jourdain is published, I could support publication of this manuscript with what I'd call 'minor' revisions, which I hope would include an expanded discussion of the thermal forcing correction, the need to push the model to steady state, and a consideration of thermal forcing that produces refreezing. All of these expanded discussions should come at the expense of the long term runs.

These points are addressed above. Briefly, the Jourdain paper has been published; the revised manuscript explores the thermal forcing correction in detail for the Amundsen sector; the longer runs shed light on slow processes that could ultimately prove to be catastrophic; and an ISMIP6 group paper at this level of detail is not feasible.

That said, the option I favor most would be a companion paper that includes more ISMIP6 participants, and that lays out the details of modeling for the experiments in ISMIP6.

As stated above, this paper goes beyond the ISMIP6 protocol, and to our knowledge we are the only group to have investigated these specific aspects of the ocean forcing in detail.

I have a number of comments on individual passages of the paper as well.

Page 1 line 7, 'nudge' seems chatty and imprecise. "Constrain to observed values"? Page 1, line 13: say what the missing physics are.

We revised the Abstract (p. 1, ll. 6-8) to read "During the spin-up, basal friction parameters and basin-scale thermal forcing corrections are adjusted to optimize agreement with the observed ice thickness."

We clarified the missing physics as follows (p. 1, l. 17): "simplified treatments of calving and basal sliding."

Page 2, line 34: say what is meant by 'committed'.

We added the following text (p. 3, ll. 7-8): "By 'committed', we mean SLR due to ice-sheet retreat that has been set in motion and is likely irreversible."

Page 3, line 1: Be more consistent with the purpose of the paper, the abstract makes quantitative claims ("10 cm to nearly 2 m"), I'm not sure this sentence really fits with the approach in much of the rest of the paper.

We revised this sentence (p. 3, ll. 10-11) as stated above: "Because of modeling and forcing uncertainties that grow larger on time scales beyond a century, our long-term results (e.g., beyond 2100), should not be viewed as predictions."

Page 2 Lines 5-9 it's as if the authors agree the exercise is pointless, but are rationalizing the CPU allotments. Just drop it.

Our intention here is to be honest and recognize the limitations of our approach. We think it is fair to say that "the timing and magnitude of simulated ice sheet retreat are imprecise", but we can still "identify responses that are robust across simulations", such as the collapse of much of the WAIS under projected increases in ocean thermal forcing.

Page 4, line 1-16, this is really a long road to just stating that the power-law sliding law is used. The 'nearly all' statement in line 17 is frustrating too. Where is the relation Coulomb? What's the point of having effective pressure in relations if one just assumes that it is ice overburden. This entire discussion could be collapsed to something much more concise.

We agree that for the initial submission, it was not helpful to discuss a Coulomb relation when the sliding law was a power law in practice. In the revised submission, we have included simulations (Sect. 4.2.4) that use the Schoof sliding law with p > 0. In Sect. 2.1 (p. 4, ll. 24ff), we still describe the Schoof sliding law (Eq. 2), while emphasizing that a power law (Eq. 1) is the default.

Page 4, line 30 - I'm often confused by the (unpublished) ISMIP6 protocols. Here you say CISM runs are based on the protocols. Page 7, line 5 says you differ from protocols with regard to

δT_sector. That's a fairly big deviation from protocols. So...be consistent in saying what you've done.

The ISMIP6 protocols (Nowicki et al., 2020) have now been published. We think it is important to state that our runs are based on ISMIP6 protocols – for example, the computation of basal melt rates using ISMIP6 thermal forcing data and melt parameterizations – but also to discuss how and why our runs are different.

Page 5, line 11 - are these mean annual, monthly, daily atmospheric forcings?

They are annual mean forcings. We revised this text (p. 6, ll. 13-14) to "For projections, ISMIP6 provides data sets of annual-mean atmosphere and ocean forcing on standard grids."

Page 5, line 35 - This is odd - I guess you force the thickness near the grounding line to be close to observations by altering δT_sector, but then protect against the large negative values found for many sectors by zeroing out the contribution to melt? So, you have refreezing (I guess) if it aids thickness on initialization, but don't allow refreezing in forward runs? This really isn't clear. Is equation 4 used for spinup, and the modified version used for forward modeling?

The problem with using the parameterization unaltered is that it can lead to large, unphysical melting or freezing rates if an entire sector has negative thermal forcing. For example, negative forcing in the basin combined with negative local forcing would yield a large positive melt rate. The parameterization was not formulated with this possibility in mind. We revised the text (p. 7, ll. 2-3) to make this more clear: "For the simulations in this paper, the last term in Eq. (4) was modified to max(...) to prevent spurious melt and freezing in sectors where <TF> + dT_sector < 0.

Or, is the modified eq. 4 used for both? If so, then the negative values of δT_sector never have any impact, do they? Eqn 5 also eliminates refreezing.

Here is the big question: Aren't you introducing a big transient when (if) you jump between the refreezing allowed in the initialization and the melt only situation in the forward runs? Is this transient physical (I don't think it is)? Again, this is the novelty of the paper, but here is another case where it's not being treated with enough consideration.

There is no difference between the treatment for spin-up and forward runs, except that δT_sector is held fixed for forward runs. So there is no such jump.

Page 8, line 22: OK - here we learn that this is not what was done as part of CISM2's contribution to Seroussi et al. (in review). This should have been scoped out in the introduction, as part of the novelty, with a clear statement of what the primary difference is.

We added some text in the Introduction (p. 2, ll. 30-32) to point out the novelty (new text in italics):

This paper complements the study of Seroussi et al. (2020} by evaluating the Antarctic response to ocean forcing in a single model, the Community Ice Sheet Model (CISM;

Lipscomb et al., 2019). *A novel feature is the tuning of a single parameter in each of 16 sectors to adjust the ocean forcing, thus optimizing model agreement with thickness observations near grounding lines.*

We did not think it was appropriate in the Introduction to explain how this method differs from our previous method.  We left this explanation in Section 3.1.

Page 9, line 17: I'm not aware of an analytical solution to the temperature equation that can accomodate advection of heat. Please provide more detail.

The temperature was initialized to an analytic solution at the start of the spin-up, but then evolves under advection.  The revised text (p. 10, ll. 25-26) reads, "For each spin-up, the ice sheet is initialized to present-day thickness using the BedMachineAntarctica data set (Morlighem et al., 2019).  The internal ice temperature is initially set to an analytic vertical profile and then evolves freely under advection."

Page 9, line 19, specify the extent of the ice that is considered for each type of 'nudge'.

The details are given earlier in Section 3.1.  We made this sentence more specific (p. 10, ll. 28-30):  "As described above, $C_p$ is adjusted in grounded cells to better match the observed local ice thickness, and $\delta T\_sector$ is adjusted in each of 16 sectors to match the observed thickness of ice near the grounding line."

Figure 5 - for context provide the $\delta T\_sector$ from Jourdain.

This figure (now Fig. 6, p.15) was modified to include a second panel with the values of $\delta T\_sector$ from Jourdain et al. (2020).

Page 12, line 6-7. This is important to me, but the treatment is not satisfactory.

We have expanded the treatment, pointing out that the reduced melt rates could be associated with prevailing conditions in the mid 20[th] century or before.  The text now reads (p. 14, ll. 1ff):

> In both sectors, significant cooling is needed to curtail the grounding-line retreat that occurs under climatological thermal forcing.  As a result, the spun-up melt rates in these regions are lower than observed.  Total sub-shelf melting for Antarctica at the end of the spin-up ranges from 625 to 744 Gt y$^{-1}$, about half the values estimated by Depoorter et al. (2013) and Rignot et al. (2013).  Negative values of $\delta T\_sector$ might be compensating for other errors, such as biases in the climatology or the failure of the melt scheme to deliver ocean heat to the right places.  Another possibility, considered in Section 4.2.5, is that for some sectors, the thermal forcing derived from the 1995–2018 climatology exceeds the forcing that was typical in the mid 20[th] century and before.  In this case, $\delta T\_sector$ would be correcting for the recent warming, to generate melt rates closer to preindustrial values.

Page 17, line 5 not really based on the protocols, right? "Loosely based"?

We think the current language is appropriate.  As discussed in Sections 2 and 3.1, the ocean forcing follows ISMIP6 protocols in (1) the use of ISMIP6 climatological thermal forcing, (2) the use of thermal forcing anomalies derived from the ESMs selected by ISMIP6, and (3) the use of parameterizations from Jourdain et al. (2020) to convert thermal forcing to basal melting.  The main differences are the different tuning for $\delta T\_sector$ (based on observed ice thickness instead of melt rates) and the repeat cycling through the ESM forcing after 2100.

Page 18, line 9-11 - this is frustratingly vague.

Amundsen sensitivity and forcing are now discussed extensively in Section 4.2.5.  Here, we revised the text as follows (p. 38, ll. 3-8):

[revised manuscript text omitted]

---

## Referee Report (RR1)

**Review of: ISMIP6-based projections of ocean-forced Antarctic Ice Sheet evolution using the Community Ice Sheet Model**

Thomas Zwinger

December 7, 2020

**1   General Impression**

The second revision of the manuscript now includes a wider sensitivity study of the ISMIP6 CISM setup, including also different approximations to the governing equations. In my view, the main points of my criticism from the first round have been fully addressed. The authors have taken great efforts to accomplish a wide range of simulations with variations in parameters, approximations to the Stokes equation and physical models for calving and sliding. Representation of data in graphics has significantly improved - I in particular like the choice of colour scales. I see one issue about the conclusions concerning the Amundsen sea sector stability open to be elaborated, but else recommend the publication of this manuscript.

**2   Main point**

I kind of pick up my first point from the last review, with a particular focus on a **possible underestimation of the contribution of SLR from Amundsen sea sector**. In your response letter you state: *Removal of the Thwaites shelf does not lead to much additional retreat in the Amundsen Sea basin, suggesting that the current buttressing effects are small.* I presume that this statement applies solely to the previously applied power-law. In the main text you added a statement: *For example,if the topographic dataset is missing pinning points near the grounding line,the ice will be biased thin, and $C_p$ can be driven to high values to make up for the lack of buttressing.* I subscribe to this statement, but I think one should also include the relatively coarse interpolation of the bedrock that can easily miss out on potential pinning points. In view of this, I am not surprised that in your model a vanishing shelf in front of Thwaites is of less severe consequence as – in my opinion – it would be in real world. Apparently, when using the Coulomb law, which is less able to compensate for missing pinning points and in my view a physically more justifiable choice at the grounding line

in particular with coarse resolution ([Gladstone et al. 2017]), you seem to get the tendency for an ice-sheet collapse in this region. It is my opinion that the latter version hence is the more realistic scenario and that scenarios showing stability of the Amundsen sea sector are merely a consequence of a combination of under-resolution and the choice of in my view not optimal power law as the standard for sliding. I see that you reported all these findings at different places in the text, but it is somewhat spread all over the manuscript. I would like to see some statement in the conclusions (as this the highest impact part of the paper) on the in my view not enough highlighted importance of the choice of the correct sliding law, which I think is key to get realistic SLR estimations. Else, I would like to get some clear arguments (beyond numerical stability) why the power law in your view is the right standard choice.

**3 Suggested corrections, additions and typos**

Please, find suggestions for corrections, changes and typos spotted in the order of their occurrence in the text. Line numbers refer to the change-tracked version of the manuscript that was attached to the reply letter:

- Page 9, line 10: *...$C_p$ is initialized to 20,000 Pa $m^{-1/3}$ $y^{-1}$* . What motivated the significant reduction of the initial value of 50,000 Pa m$^{-1/3}$ y$^{-1}$ in the previous version?

- Page 10, line 29: *...is adjusted in each of the 16 sectors to match the observed ...*

- Page 11, line 1: *We compensate by increasing the target thickness for this sector by the equivalent of 1000 Gt of ice, ....* Can you please elaborate how exactly this was implemented? How do you distribute this amount of ice underneath the shelf?

- Page 26, Fig. 7: Can you improve the explanation what comes to the black line that sometimes goes through the middle of the ice shelf (upper left panel; non-local MeanAnt) and is missing on the calving front at all shelves in (lower left panel; non-local PIGL). From reading further in the text I understand that all the shelf that is not confined within black lines is gone in 2500. For convience for the readers, I suggest you put this piece of information into the caption of the figure. Same applies Fig. 8 on next page.

- Page 22, Fig. 12: Can you help me (and also the readers) interpreting the closed black loop inside the Ross ice-shelf in the lowest left panel (nonlocal MeanAnt CESM2)?

- Page 23, Fig. 22: *CISM gives accurate results in steady-state and perturbation experiments at resolutions of 2 to 4 km.* Accurate with respect to what?

- Page 23, Fig. 27: ...*resulting in slightly more drift during the projection runs.* Is it possible to elaborate on the trend (lower or higher sea level) of this drift?

- page 24, line 2: ...*for which the SLR contributions at 2, 4, and 8 km are 1047 mm, 1300 mm,and 1473 mm, respectively.* In this line you report the lowest SLR (= lowest ice loss) for the 2 km grid but in the beginning of the next paragraph – which seems also to be confirmed by the graphs in Fig. 13 – claim that the finest 2 km grid has the largest ice loss (quote): *Thus, refining the grid to 2 km leads to greater ice loss.* For me this is contradicting. Please let me know if I am missing something.

- page 35, Fig. 24: Subjectively, I cannot read a lot from this graph because of a with respect to the lines over-dominating texture colour scale. This single new figure is in contrast to the in my view good choice of colour scales in the other figures. Additionally, if in any way possible with reasonable effort, it would be good to get the years annotated to the different lines.

**References**

[Gladstone et al. 2017] Gladstone, R. M., Warner, R. C., Galton-Fenzi, B. K., Gagliardini, O., Zwinger, T., and Greve, R.: Marine ice sheet model performance depends on basal sliding physics and sub-shelf melting, The Cryosphere, 11, 319–329, https://doi.org/10.5194/tc-11-319-2017, 2017.

---

## Author Response (AR2)

To the Editor:

We have attached the latest reviewer comments with our responses.  We corrected the manuscript as requested, and have attached two versions of the new manuscript.  One version (at the end of this document) shows text changes in blue font, and the other is a clean copy.

We reiterate our thanks to Thomas Zwinger for exceptionally thorough and thoughtful comments that have greatly improved the paper.  Our responses are below.  Page and line references are to the corrected version.

**Reviewer #1** (Thomas Zwinger):

**General impression**:  The second revision of the manuscript now includes a wider sensitivity study of the ISMIP6 CISM setup, including also different approximations to the governing equations. In my view, the main points of my criticism from the first round have been fully addressed. The authors have taken great efforts to accomplish a wide range of simulations with variations in parameters, approximations to the Stokes equation and physical models for calving and sliding. Representation of data in graphics has significantly improved - I in particular like the choice of colour scales. I see one issue about the conclusions concerning the Amundsen sea sector stability open to be elaborated, but else recommend the publication of this manuscript.

Thank you for these positive comments.  The issue with the conclusions is addressed below.

**Main point**: I kind of pick up my first point from the last review, with a particular focus on a possible underestimation of the contribution of SLR from Amundsen sea sector. In your response letter you state:  "Removal of the Thwaites shelf does not lead to much additional retreat in the Amundsen Sea basin, suggesting that the current buttressing effects are small."  I presume that this statement applies solely to the previously applied power-law. In the main text you added a statement: "For example, if the topographic dataset is missing pinning points near the grounding line, the ice will be biased thin, and $C_p$ can be driven to high values to make up for the lack of buttressing." I subscribe to this statement, but I think one should also include the relatively coarse interpolation of the bedrock that can easily miss out on potential pinning points. In view of this, I am not surprised that in your model a vanishing shelf in front of Thwaites is of less severe consequence as – in my opinion – it would be in real world. Apparently, when using the Coulomb law, which is less able to compensate for missing pinning points and in my view a physically more justifiable choice at the grounding line in particular with coarse resolution (Gladstone et al. 2017), you seem to get the tendency for an ice-sheet collapse in this region. It is my opinion that the latter version hence is the more realistic scenario and that scenarios showing stability of the Amundsen sea sector are merely a consequence of a combination of under-resolution and the choice of in my view not optimal power law as the standard for sliding. I see that you reported all these findings at different places in the text, but it is somewhat spread all over the manuscript. I would like to see some statement in the conclusions (as this is the highest

impact part of the paper) on the in my view not enough highlighted importance of the choice of the correct sliding law, which I think is key to get realistic SLR estimations. Else, I would like to get some clear arguments (beyond numerical stability) why the power law in your view is the right standard choice.

We agree that Amundsen stability depends in large part on the use of power-law rather than Coulomb sliding. (Another key factor, as emphasized in the revised draft, is the treatment of ocean thermal forcing anomalies.) We have not taken a position on whether a power law or Coulomb law is optimal, in part because we think a different law (or a different value of the parameter $p$ used for effective pressure) may be optimal in different parts of the ice sheet.

We modified the conclusion to emphasize the importance of the basal friction law, both for the Amundsen sector and overall. In the third paragraph, which summarizes the sensitivity experiments, we added this line (p. 36, l. 26): "Thus, the sensitivity of SLR to the basal friction law can be comparable to the sensitivity to the basal melt parameterization and calibration."

Later in the conclusion, when discussing caveats relating to Amundsen sensitivity, we added this sentence (p. 37, l. 29): "If Amundsen glaciers are best described by a Coulomb basal friction law, then the use of power-law friction would also understate the sensitivity."

We also modified the Abstract to state that the friction law is more important than the other factors tested in sensitivity experiments. The sentence summarizing these experiments now reads (p. 1, l. 13), "Further experiments show relatively high sensitivity to the basal friction law, moderate sensitivity to grid resolution and the prescribed collapse of small ice shelves, and low sensitivity to the stress-balance approximation."

In the passage quoted above from Sect. 3.1 (p. 9, l. 15), we changed "topographic data set" to "prescribed topography." The new wording is neutral as to whether missing pinning points are absent in the high-resolution BedMachine data or are smoothed out by grid coarsening.

**Suggested corrections, additions and typos**

Please, find suggestions for corrections, changes and typos spotted in the order of their occurrence in the text. Line numbers refer to the change-tracked version of the manuscript that was attached to the reply letter:

- Page 9, line 10: . . . $C_p$ is initialized to 20,000 Pa m$^{-1/3}$ y$^{-1}$ . What motivated the significant reduction of the initial value of 50,000 Pa m$^{-1/3}$ y$^{-1}$ in the previous version?

  The optimized value of $C_p$ is not sensitive to the initial value, so this choice does not significantly affect the spun-up results or forward simulations. A lower initial value results in faster convergence to the final value in regions with small $C_p$ and fast sliding. However, choosing a value that is too low can lead to instability early in the simulation. After making some minor numerical changes that improved stability, we found that spin-ups were stable with initial values as low as 20,000, so we used this value in the second round of simulations. Since this is a fairly minor technical detail, we have not modified the text.

- Page 10, line 29: ...is adjusted in each of *the* 16 sectors to match the observed ...

  Done.

- Page 11, line 1: We compensate by increasing the target thickness for this sector by the equivalent of 1000 Gt of ice, .... Can you please elaborate how exactly this was implemented? How do you distribute this amount of ice underneath the shelf?

  There is no extra ice, per se, that needs to be distributed. The extra 1000 Gt is added to the target value H_obs, which contributes to the adjustment of δT_sector via Eq. (10). In all spin-ups, Pine Island Glacier advances beyond the present-day grounding line, which lies in an unstable position. Since ice in the PIG region is then thicker than observed, Eq. 10 will compensate by increasing the thermal forcing and driving GL retreat elsewhere in the sector. Increasing the thickness target reduces or removes the need for this compensating (and possibly spurious) retreat.

  To clarify the reasoning behind this adjustment, we added the following text (p. 10, l. 24): "With this adjustment, the advance of the Pine Island grounding line from the present location to a more stable position downstream (which takes place in all spin-ups) does not have to be compensated by retreat elsewhere in the sector."

- Page 26, Fig. 7: Can you improve the explanation what comes to the black line that sometimes goes through the middle of the ice shelf (upper left panel; non-local MeanAnt) and is missing on the calving front at all shelves in (lower left panel; non-local PIGL). From reading further in the text I understand that all the shelf that is not confined within black lines is gone in 2500. For convenience for the readers, I suggest you put this piece of information into the caption of the figure. Same applies Fig. 8 on next page.

  Your understanding is correct, and we agree with the suggestion. We added the following text in the caption of Fig. 7 (p. 16): "In the nonlocal-PIGL run (lower left), black lines at the present-day calving fronts of several large ice shelves are absent, indicating shelf collapse. In the nonlocal-MeanAnt run (upper left), the Ross Ice Shelf partly collapses."

  Similarly, we added the following to the Fig. 8 caption (p. 17): "In the two nonlocal runs (upper and lower left), black lines are absent at the present-day calving fronts of several ice shelves, indicating shelf collapse."

- Page 22, Fig. 12: Can you help me (and also the readers) interpreting the closed black loop inside the Ross ice-shelf in the lowest left panel (nonlocal MeanAnt CESM2)?

  Yes, this is an odd feature. Part of the shelf interior collapses, without calving-front retreat. This might be an unrealistic artifact of the nonlocal melt scheme, combined with the CESM2 thermal forcing anomaly and the simple no-advance calving scheme.

  In the text discussion of Fig. 12 (p. 22, l. 20), we added the following sentence: "As seen in Figs. 7 and 8, the nonlocal-slope melt scheme drives substantial grounding-line retreat

without calving-front retreat, while the nonlocal scheme can trigger Ross Ice Shelf collapse, with less grounding-line retreat." We also added a paragraph break.

In the Fig. 12 caption (p. 21), we added the following: "In the nonlocal HadGEM2 run (upper left), retreat of the black line from the Ross calving front indicates shelf collapse. In the nonlocal CESM2 run (lower left), the enclosed black contour shows that part of the Ross Ice Shelf interior has collapsed, without calving-front retreat."

- Page 23, Fig. 22: CISM gives accurate results in steady-state and perturbation experiments at resolutions of 2 to 4 km. Accurate with respect to what?

  We meant that the results at these coarser resolutions are close to the well-converged results at higher resolutions of 1 km or less. We modified the text as follows (p. 23, l. 4): "CISM results at resolutions of 2 to 4 km are close to the converged results at 0.5 to 1.0 km."

- Page 23, Fig. 27: ...resulting in slightly more drift during the projection runs. Is it possible to elaborate on the trend (lower or higher sea level) of this drift?

  Yes, the drift in ice mass is positive (i.e., toward lower sea level), with a value of 1 to 5 Gt/y, compared to < 1 Gt/y at 20ky in the standard spin-ups. We added this text (p. 23, l. 11): "The drift in ice mass at the end of the 2-km spin-ups is 1 to 5 Gt/y (toward greater mass and lower sea level), compared to a drift of < 1~Gt/y at the end of the 20-ky standard spin-ups."

- page 24, line 2: . . . for which the SLR contributions at 2, 4, and 8 km are 1047 mm, 1300 mm,and 1473 mm, respectively. In this line you report the lowest SLR (= lowest ice loss) for the 2 km grid but in the beginning of the next paragraph – which seems also to be confirmed by the graphs in Fig. 13 – claim that the finest 2 km grid has the largest ice loss (quote): Thus, refining the grid to 2 km leads to greater ice loss. For me this is contradicting. Please let me know if I am missing something.

  Thank you for spotting the contradiction. We meant to say "for which the SLR contributions at **8, 4, and 2 km** are 1047 mm, 1300 mm, and 1473 mm, respectively." We fixed the error (p. 23, l. 20).

- page 35, Fig. 24: Subjectively, I cannot read a lot from this graph because of a with respect to the lines over-dominating texture colour scale. This single new figure is in contrast to the in my view good choice of colour scales in the other figures. Additionally, if in any way possible with reasonable effort, it would be good to get the years annotated to the different lines.

  We agree that the color scheme for topography was too noisy. In the revised figure (p. 35), the bed topography is shown on a simpler gray scale, with white for higher elevation and black for lower. The grounding lines are now shown in colors (mauve, cyan, orange, red, green) that stand out from the background, and the caption is modified accordingly. With these changes, we do not think it is necessary to annotate the years.

[revised manuscript text omitted]